# Electrochemical ohmic memristors for continual learning

**Shaochuan Chen** [1,2,10] ✉, **Zhen Yang**[3,10], **Heinrich Hartmann**[4], **Astrid Besmehn**[4], **Yuchao Yang** [3,5,6,7] ✉ **& Ilia Valov** [8,9] ✉

Developing versatile and reliable memristive devices is crucial for advancing future memory and computing architectures. The years of intensive research have still not reached and demonstrated their full horizon of capabilities, and new concepts are essential for successfully using the complete spectra of memristive functionalities for industrial applications. Here, we introduce two-terminal ohmic memristor, characterized by a different type of switching defined as filament conductivity change mechanism (FCM). The operation is based entirely on localized electrochemical redox reactions, resulting in essential advantages such as ultra-stable binary and analog switching, broad voltage stability window, high temperature stability, high switching ratio and good endurance. The multifunctional properties enabled by the FCM can be effectively used to overcome the catastrophic forgetting problem in conventional deep neural networks. Our findings represent an important milestone in resistive switching fundamentals and provide an effective approach for designing memristive system, expanding the horizon of functionalities and neuroscience applications.

The memristor was proposed by Leon Chua in the 1970s as the fourth fundamental circuit element complemented by the capacitor, resistor, and inductor[1]. It is a two-terminal resistor that exhibits memory function, that is, the resistance of a memristor is a function of the charge flown through the system, controlled by the application of external electrical bias. Despite resistance switching has been observed in electronic devices back to the nineteenth century[2], it was until 2008 that Strukov et al. proposed the link between the theoretical model and a physical device[3]. Considerable progress has been made towards utilizing the unique memristive properties not only for different types of non-volatile memories but as well as sensors, hardware security, artificial intelligence, and as well artificial neurons and synapses,

constituting a building unit for new type of hardware, overcoming the physical limitations of the modern nanoelectronics and considerably reducing the energy consumption[4–12].

Memristive devices can operate on different physical principles, and among all the redox-based ones (ReRAM) are considered as one of the most promising. Here, depending on the operation mechanisms we distinguish electrochemical metallization (ECM) and valence-change mechanism (VCM) devices. The devices share identical structure but differ in the nature of the used materials, electrochemical processes, and thus, in mechanism. For all types of devices applies one general rule—the switching mechanism practically determines the functionalities and therefore the applications.

[1]Institute of Materials in Electrical Engineering 2 (IWE2), RWTH Aachen University, Aachen, Germany. [2]International Center for Young Scientists (ICYS), National Institute for Materials Science (NIMS), Tsukuba, Japan. [3]Beijing Advanced Innovation Center for Integrated Circuits, School of Integrated Circuits, Peking University, Beijing, China. [4]Central Institute for Engineering, Electronics and Analytics (ZEA-3), Forschungszentrum Jülich, Jülich, Germany. [5]Guangdong Provincial Key Laboratory of In-Memory Computing Chips, School of Electronic and Computer Engineering, Peking University, Shenzhen, China. [6]Center for Brain Inspired Chips, Institute for Artificial Intelligence, Peking University, Beijing, China. [7]Center for Brain Inspired Intelligence, Chinese Institute for Brain Research (CIBR), Beijing, China. [8]Peter Grünberg Institute 7 and JARA-FIT, Forschungszentrum Jülich, Jülich, Germany. [9]Institute of Electrochemistry and Energy Systems, Bulgarian Academy of Sciences, Sofia, Bulgaria. [10]These authors contributed equally: Shaochuan Chen, Zhen Yang. ✉e-mail: CHEN.Shaochuan@nims.go.jp; yuchaoyang@pku.edu.cn; i.valov@fz-juelich.de

In ECM, an essential role plays cations, that are electrochemically dissolved and transported through the switching film, forming a conductive metallic nanofilament. The filament stability and dynamics are determined by redox reactions, ionic and electronic transport at/through the interfaces, and within the volume of the switching film. To improve and modulate the performance of ECM devices various metals, such as Ag, Cu, Ni, and their alloys are employed as electrochemically active electrodes[13–15]. The high resistive state (HRS) and low resistive state (LRS) are determined solely by the formation/dissolution cycles of this filament[16,17].

VCM devices are more complex in mechanism—one of the electrodes must form a Schottky contact with the oxide, required to be a high-work-function metal such as Pt or Pd[13,18]. This electrode is selected inert but should have certain catalytic activity towards the main electrochemical reactions which include oxidation/reduction of oxygen and/or the active metal component and as well parallel reactions of moisture/protons[14]. HRS and LRS transition relies on the formation and modulation of the Schottky barrier between this electrode and the filament tip, driven by the movement of oxygen ions/vacancies and/or cations and electrons within this disc[18,19]. The other electrode should have low-work function and high oxygen affinity such as Ta, Hf, Zr, and Ti, forming an ohmic contact with the oxide, serving additionally as an oxygen buffer/reservoir. In VCM, the filament is only partially dissolved upon the RESET process and remains in reduced length during the entire device life and operation, leaving a small spacing (disc) of several nanometers to the electrode.

Despite significant progress has been reported and the efforts made to achieve high-performance and reliable electrical properties, for both ECM and VCM-type memristive systems, the commercialization is progressing slowly mainly due to stochasticity and operation failures. Intrinsic limitations can be pointed for instance the sensitivity of small volumes towards charge enrichment/depletion. In macroscopic systems, electrolytes are considered infinite sources of ions. Taking or adding some is not influencing the system as a whole. In contrast, for nanoscale volumes every charge added or extracted may significantly influence the conductivity, the transference numbers (ratio of conductivities of different charges), and as well cause a transition from field accelerated kinetics to low field conditions and vice versa. In addition, Joule heating and mechanical effects influence the systems[20–22]. This complex behavior makes the nanoscale memristive systems difficult to study and control. These limitations cause severe device non-idealities in state retention, endurance, and uniformity, making the device operation rather complicated. Therefore, a continued fundamental research in this field is essential in order to improve the control over the nanoscale processes, reduce the level of complexity of the systems, exploring new materials and switching mechanisms.

In this study, we report the discovery of a principally different electrochemical memristive mechanism, that we term as filament conductivity change mechanism (FCM) by creating dual ohmic contact metal/oxide interfaces, instead of one Schottky contact and one ohmic contact, as used in conventional VCM devices[18]. The two-terminal ohmic memristive devices based on OE/Ta$_2$O$_5$/OE junction with various ohmic electrode (OE) materials were fabricated and characterized. Based on electrical, transmission electron microscopy (TEM), and spectroscopy characterization, we show that the switching mechanism in our memristive systems relies entirely on redox reactions and ion transport. Moreover, the design of the OE/oxide/OE structure reduces the physicochemical complexity of the system by avoiding the presence of Schottky barrier height modulation. We also discuss in detail and explain the essential role of the interplay between the Schottky barrier, and the electrochemical redox barrier influences the processes and stability of memristive systems and explain the significant improvement of the devices in terms of complex performance and reliability. The kinetics studies reveal rich physicochemical dynamics,

implying the presence of migration and redox reactions of both anion and cation species. This phenomenon is corroborated by the cross-sectional TEM, by which nanoscale conduction channels were detected in the oxide layer, bridging the top and bottom electrodes. The various oxidation states of the conduction channel at the active switching region allow the programming of the memristive device in a binary and/or analog manner. Combining the controlled, ultra-stable binary and analog switching functionalities, we further conduct neural networks simulation demonstrating the ohmic memristive device can be implemented in artificial neural networks model and achieve high pattern recognition accuracies on multiple image datasets.

## Results and discussions
### Materials design and electrical properties
The ohmic memristor is composed of a thin Ta$_2$O$_5$ layer (~8 nm), sandwiched by two OEs with low-work functions (Supplementary Table 1). Figure 1a shows a schema of the two-terminal OE/Ta$_2$O$_5$/OE junction. We used Ta as the bottom electrode, combined with various top electrodes: Hf, Ta, and Zr, which are commonly used as OEs (creating nearly ohmic contact with the oxide) in conventional VCM devices[18,23]. To compare the electrical/electrochemical properties, we fabricated reference VCM devices with Pt/Ta$_2$O$_5$/Ta layered structure, where a Schottky contact is present at the top electrode/oxide interface. For the ohmic memristors, in the first stage, no capping layers were used for the top (Hf, Ta, or Zr) and bottom (Ta) electrodes. The Hf(Ta, Zr)/Ta$_2$O$_5$/Ta junctions were directly fabricated on a 430 nm thermally oxidized SiO$_2$/Si wafer. The purpose was to eliminate any influence of capping materials on the electrical characteristics of the device[24]. We conducted I−V sweep measurements on the fabricated devices by applying an electrical bias to the top electrode. Figure 1b shows the corresponding I−V characteristics, with typical counter-eight-wise (c8w) bipolar resistive switching[13,18]. The bipolar switching is determined by the chemical difference of Hf/Zr and Ta electrodes (ensuring an electromotive force of at least 800 mV) and is supported by the different water and oxygen partial pressures at the top and bottom electrodes ($p$H$_2$O and $p$O$_2$), and as well by the different thickness and/or stoichiometry of the suboxides formed at the interfaces during the device preparation. The latter factors are essential for the Ta/Ta$_2$O$_5$ interfaces[25]. After exceeding a negative threshold voltage, the current increases abruptly (SET transition) shifting the cell from a HRS to a LRS. A positive voltage sweep induces the RESET transition, switching the device from LRS to HRS. The material of the top electrode does not cause significant influence on the switching behavior (Fig. 1b). This differs from conventional VCM devices whose electrical characteristics are highly dependent on the class of top electrode since the oxygen affinity and free energy of oxide formation of top electrode decides the total amount of created oxygen vacancies in the metal oxide[26].

We noted the electrical characteristics of the ohmic memristive devices are comparable to that obtained from conventional Pt/Ta$_2$O$_5$/Ta VCM device (Supplementary Fig. 1b). However, due to the high oxygen affinity of Hf, Ta, Zr electrodes[19,27], the ohmic memristive devices without capping layers are sensitive to surface oxidation when measured at ambient conditions, leading to electrode passivation[28,29] and eventually resulting in deterioration in electrical properties, leading to limited electrical stability in endurance and state retention with time (Supplementary Fig. 2).

To improve the chemical and electrical stability, we have looked for electrode materials, able to conduct sufficiently good electrons, to have the ability to exchange oxygen ions and forming no Schottky barrier. We utilized IrO$_2$/Pt as suitable capping layers for the OEs to prevent their passivation, however, keeping the performance. The conductive IrO$_2$ buffer layers could act as internal series resistors and can stabilize the electrical characteristics by improving the electrical contact and limiting the total current[30,31].

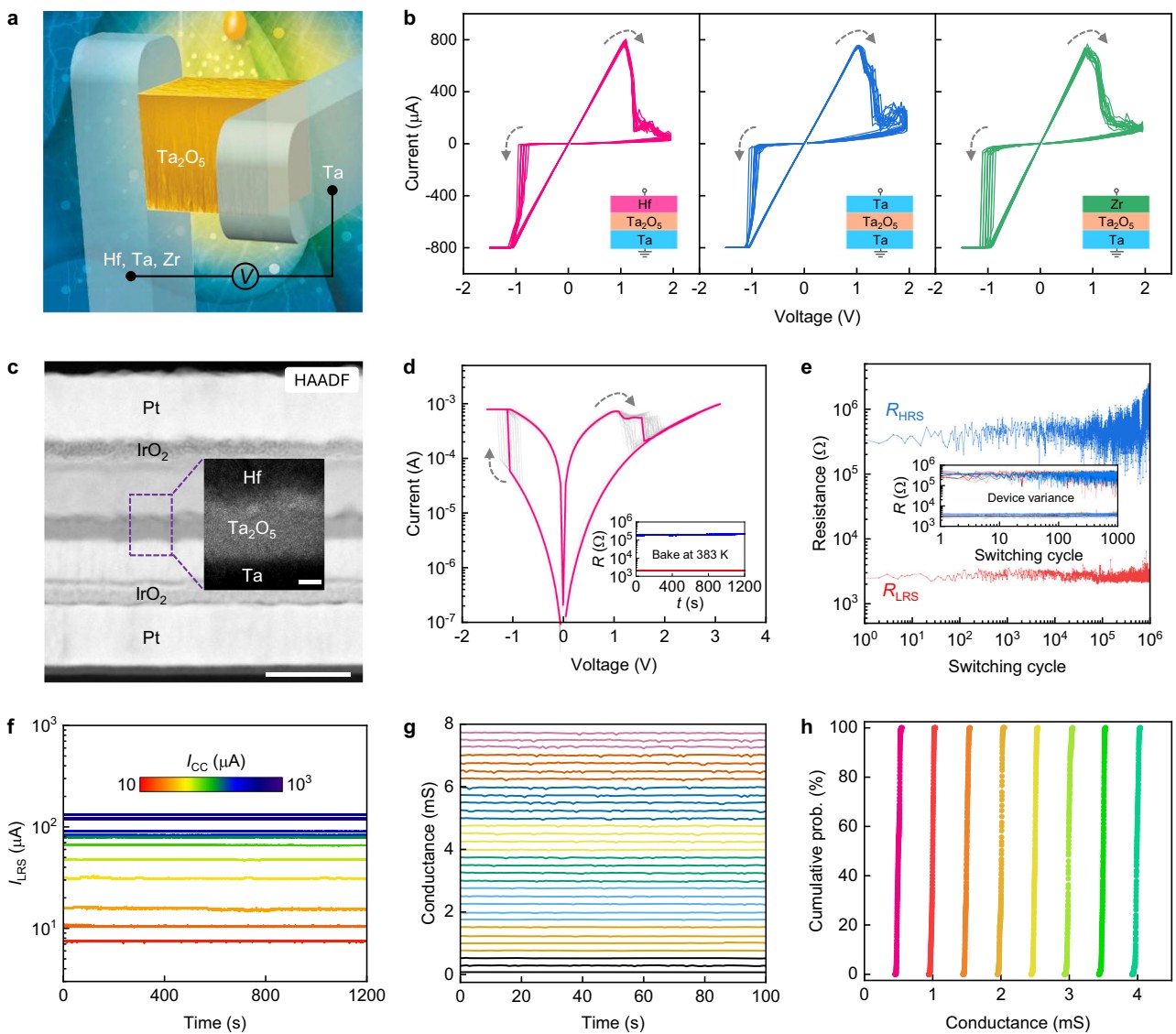

**Fig. 1 | Device and electrical properties. a** Schematic illustration of the two-terminal metal-insulator-metal memristive device. The Ta$_2$O$_5$ resistive switching layer is sandwiched by top and bottom electrodes. Optical microscopy image of the device structure is shown in Supplementary Fig. 14a. **b** I−V characteristics of the ohmic memristive devices using Ta as the bottom electrode and Hf, Ta, Zr as top electrode, respectively. **c** High-angle annular dark-field TEM image showing the layered information of the optimized ohmic memristive device. Scale bar, 50 nm. The inset shows the zoomed-in HRTEM image. Scale bar, 4 nm. EDS elemental mapping of each layer and EDS line-scan profiling are shown in Supplementary Fig. 14d. **d** Consecutive 100 resistive switching cycles of the optimized Hf/Ta$_2$O$_5$/Ta device. HRS and LRS resistance show no degradation under a constant −200 mV

electrical stress at 383 K for over 1200 s. **e** Endurance result showing one million pulse resistive switching cycles. The inset shows the endurance results obtained from ten individual devices. HRS and LRS resistance were collected from −0.2 V/ 6 ms read pulse after each RESET and SET pulse, respectively. For all devices, the applied electrical potentials are identical: $V_{SET}$: −1.2 V/30 ms, $V_{RESET}$: 3.2 V/2 ms. Statistical analysis on cycle-to-cycle and device-to-device variation is shown in Supplementary Fig. 6. **f** Electrical stability of various LRS currents after the device was programmed using different current compliance in SET operation. **g** Stability of 32 (5-bit) distinguishable states which were obtained using the write-verify program (Supplementary Fig. 11). **h** Read noise analysis for selected 8 distinguished states, each state was read 1000 times for 10 devices.

Figure 1c shows the cross-sectional high-angle annular dark-field (HAADF) TEM image of the optimized ohmic memristive device. The inset presents a high-resolution transmission electron microscopy (HRTEM) image, showing the amorphous phase Ta$_2$O$_5$ film and clean electrode/oxide interfaces. The forming and subsequent I−V curves of the optimized devices are depicted in Supplementary Fig. 3. By comparing Fig. 1b and Supplementary Fig. 3c, it can be seen that the use of capping layers does not alter the resistive switching behavior. Moreover, the optimized ohmic memristive devices exhibit significantly improved electrical stability and performance (Fig. 1d–h). They can be characterized by a much broader voltage stability window, showing lower forming voltages, high-temperature stability, no current overshooting, low variability, and the ability to switch in both binary and

analog manner. The I−V characteristics exhibit high cycling uniformity (Supplementary Fig. 3d). $R_{HRS}$ and $R_{LRS}$ show no distinguishable drift for over 1200 s under the constant electrical stress (−200 mV) at 383 K (see inset of Fig. 1d), demonstrating high-temperature stability. At room temperature, the HRS and LRS resistance can be retained for over 2 days (Supplementary Fig. 4). The HRS and LRS resistance ratio (200–1000) was maintained for over one million pulse switching cycles (Fig. 1e, Supplementary Fig. 5). Moreover, the devices show minimal spatial variation (see inset of Fig. 1e, Supplementary Figs. 6 and 7) under identical pulse programming conditions. These electrical properties are superior to that of conventional Ta/Ta$_2$O$_5$/Pt and Hf/Ta$_2$O$_5$/Pt VCM devices prepared and tested at the same conditions, as shown in Supplementary Fig. 8, which typically suffer from

RESET failure problem in endurance measurement. Overall, by using two OEs to minimize the interfacial Schottky barrier height, the ohmic memristive devices exhibit more reliable forming process (Supplementary Fig. 3a, b), high cycle-to-cycle (Fig. 1d) and device-to-device uniformity (Supplementary Figs. 6 and 7), good endurance with high HRS and LRS resistance ratio (Supplementary Fig. 9). The ohmic memristive device can be also programmed in an analog manner[32]. In I–V sweep measurement, by varying the current compliance, multilevel LRS currents with stable retention stability (biased at 200 mV) can be obtained, as shown in Fig. 1f and Supplementary Fig. 10. For the analog switching capability, we further developed a pulse write-verify scheme (Supplementary Fig. 11) to program the device into the target conductance within given errors. The typical examples of device conductance variation under the negative feedback programming scheme were shown in Supplementary Fig. 12. Figure 1g shows obtained 32 (5-bit) distinguished and evenly distributed LRS states with conductance levels range from μS to mS, and can be retained after writing (see Supplementary Fig. 13). The device-to-device variation and read noise for the analog switching are shown in Fig. 1h. We confirmed that by using the capping layers, the passivation of the OEs has been suppressed, as the energy-dispersive X-ray spectroscopy (EDS) and X-ray photoelectron spectroscopy (XPS) analysis reveal the presence of the metallic phase of OEs (Supplementary Figs. 14 and 15).

## Interfacial redox reactions and charge transport

The robust device characteristics of ohmic memristors are not relying on Schottky barrier height modulation, but rather on conductivity changes induced electrochemically. Therefore, we realized the importance of analyzing in detail the relation and interplay between Schottky and redox barriers at the electrode/filament interface, determining the physics of the filament formation. We have qualitatively considered the physical and electrochemical properties of the metal/oxide interface in both Schottky (Fig. 2a, b) and ohmic systems (Fig. 2c, d), and studied their dynamic responses by cyclic voltammetry (CV) measurements (Fig. 2e, f). The discussion and measurements provide essential conclusions about the charge/mass transfer processes, responsible for the resistive switching and are indicative for the ratio between electronic and Faraday currents.

Considering the transport of mass and charge at the metal/oxide interface two main equations describe the physical processes, namely the equation for thermionic emission and the Buttler-Volmer equation, giving the Faraday currents. The equation for overcoming the Schottky barrier by electrons is given by:

$$j_{TE} = j_S \left[ \exp\left(\frac{e\Delta\varphi}{kT}\right) - 1 \right] \tag{1}$$

where $j_S = \left| A^* T^2 \left[ \exp\left(-\frac{e\phi_B}{kT}\right) \right] \right|$ is the saturation current density, $A^*$ is the effective Richardson constant, $k$ is the Boltzmann constant, T is the absolute temperature, $e$ is the elementary charge, $e\phi_B$ is the barrier height, $k$ is the Boltzmann constant, $\Delta\varphi$ is the applied voltage. Under the reverse (negative) bias $\left[ \exp\left(\frac{e\Delta\varphi}{kT}\right) \right] \ll 1$, the current density, referred to as saturation current density, depends on the effective barrier height.

The equation for charge transfer limited electrode reactions given by the Butler–Volmer relation[33]:

$$j_{ION} = j_0 \left[ \exp\left(\frac{\alpha_a z e \Delta\varphi}{kT}\right) - \exp\left(-\frac{\alpha_c z e \Delta\varphi}{kT}\right) \right] \tag{2}$$

where $j_0 = zekc \exp\left(-\frac{\Delta G_a}{kT}\right)$ is the exchange current density, $z$ is the number of exchanged electrons, $k$ is the rate constant, $c$ is the

concentration of ions at equilibrium, $\Delta G_a$ is the free energy of activation, $\alpha_a$ is the anodic transfer coefficient, $\alpha_c = 1 - \alpha_a$ is the cathodic transfer coefficient, $\Delta\varphi$ is the electron-transfer overpotential.

The total steady-state mass and charge transport through this interface in both type devices is represented as a sum of Schottky-emission and redox current densities $j = j_{TE} + j_{ION}$. Despite this was mathematically included in simulation models[34], the physical interpretation and discussion on the interplay between Schottky and redox barriers have been not considered and analyzed yet. We found this is of crucial importance for understanding the complex physicochemical behavior of redox-based memristive devices with essential implications on materials design and performance.

To account for the complex electrochemical reactions and charge transport across the interfaces and to formally describe the interplay/impact of the different energy barriers, we comparatively analyzed the situations accounting for both thermionic emission and redox reactions contribution at the metal/oxide interface[33,35,36] (see Supplementary Note 1). Four boundary cases were formulated:

(i) System with high Schottky barrier under forward (positive) bias:

$$j = A^* T^2 \left[ \exp\left(\frac{\Delta\varphi - \phi_B}{kT} e\right) \right] + 2ekc \left[ \exp\left(\frac{e\Delta\varphi - \Delta G_a}{kT}\right) \right] \tag{3}$$

(ii) System with high Schottky barrier under reverse (negative) bias:

$$j = -2ekc \exp\left(\frac{-e\Delta\varphi - \Delta G_a}{kT}\right) \tag{4}$$

(iii) System with low Schottky barrier under forward (positive) bias:

$$j = A^* T^2 \left[ \exp\left(\frac{\Delta\varphi - \phi_B}{kT} e\right) - \exp\left(-\frac{e\phi_B}{kT}\right) \right] + 2ekc \left[ \exp\left(\frac{e\Delta\varphi - \Delta G_a}{kT}\right) \right] \tag{5}$$

(iv) System with low Schottky barrier under reverse (negative) bias:

$$j = A^* T^2 \left[ \exp\left(\frac{\Delta\varphi - \phi_B}{kT} e\right) - \exp\left(-\frac{e\phi_B}{kT}\right) \right] - 2ekc \left[ \exp\left(\frac{-e\Delta\varphi - \Delta G_a}{kT}\right) \right] \tag{6}$$

The corresponding energy diagrams for each boundary condition are provided in Fig. 2a–d.

In case (i) of forward bias, both electronic and Faraday processes proceed in parallel, where electronic currents are dominating. This lowers the Faraday current efficiency and suppresses the oxidation/passivation rate of the OE. High Schottky barrier at metal/oxide interface at reverse bias significantly suppress electronic currents but cannot block (case (ii)) the redox process(es) at the electrode surface as also shown in the CV measurement (inset of Fig. 2e), making the Faraday efficiency of the electrochemical reaction very high. This will lead to the formation of much stronger filaments and as well will cause also faster passivation (blocking effect) of the OE (Fig. 2e and Supplementary Fig. 16). For example, applying a pulse of 1 V of 1 mA, for 1 μs in an oxide memristive system will cause the evolution 0.56 μm³ O₂ (resulting in forming the corresponding number of oxygen vacancies and electrons). In practice, this could cause the necessity of using high RESET voltages. High Schottky barriers are also not beneficial for the forming process, as for a virgin devices high forming voltages will be necessary to apply, (because the large voltage drop across the

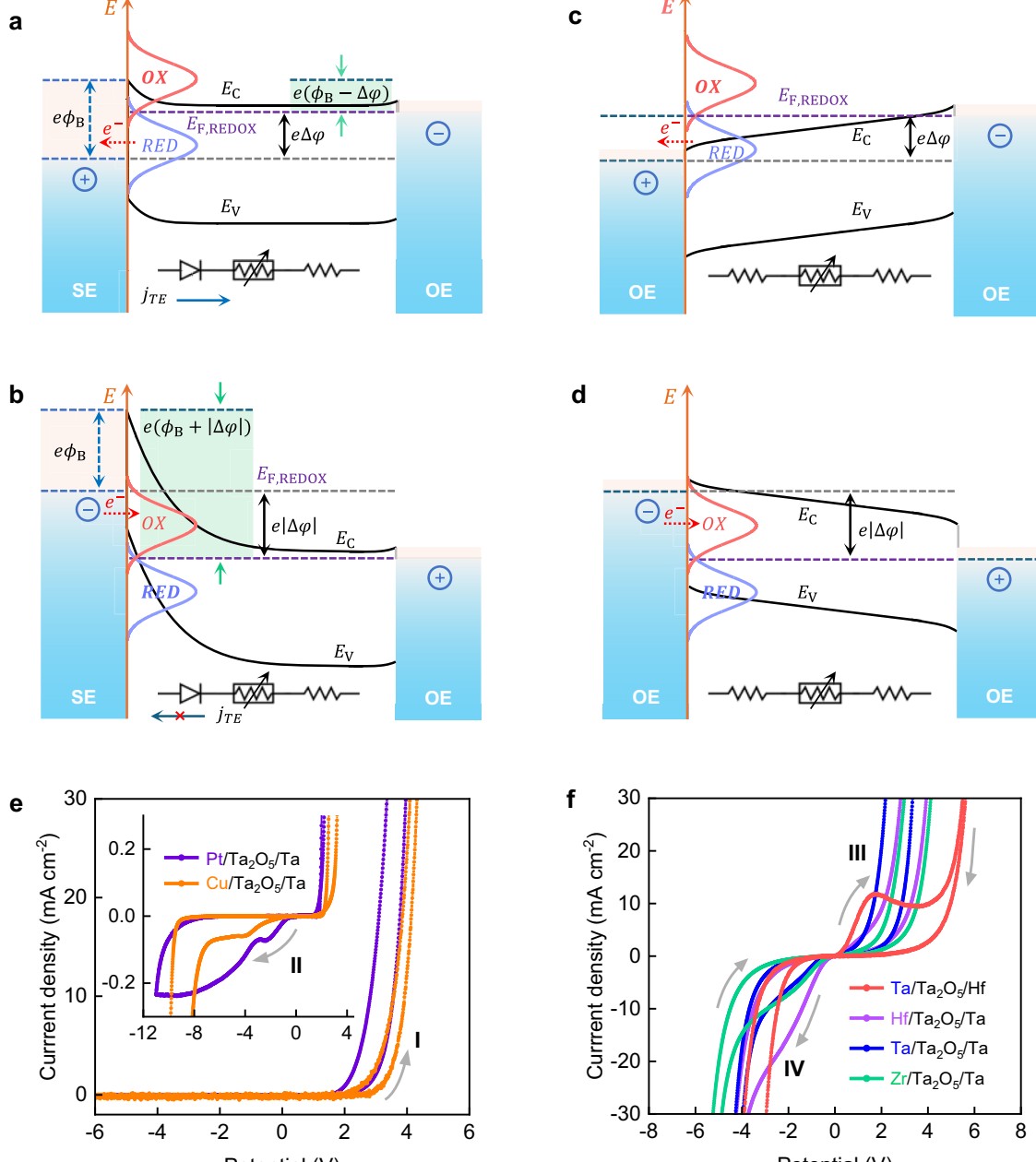

**Fig. 2 | Cyclic voltammograms and charge transfer at interfaces. a** Schematic of energy diagram of the OE/oxide/OE system with one Schottky interface and one ohmic-like interface. The Schottky interface is positively biased, providing high electronic current. **b** Schematic of energy diagram of Schottky system when the Schottky interface is under reverse bias. This results in a high energy barrier and blocks the electron movement through the SE/oxide interface. **c** Schematic of energy diagram of the OE/oxide/OE system with dual ohmic-like interfaces under positively biased and reverse biased (**d**). In these cases, due to the low Schottky energy barrier and lower level of electrode passivation, higher electronic currents can flow through the metal/oxide interface, regardless of the applied voltage polarity. For all systems, when the top electrode (Schottky electrode SE, or ohmic electrode) is anodically polarized (**a, c**), the Fermi level of the redox system ($E_{F, REDOX}$) is shifted, electrons are transferred from the occupied states (*RED*) to metal. Oxidation currents are generated during the anode process. Under the reverse

biased condition, electrons are transferred from metal to unoccupied states (*OX*), reduction process occurred at the cathodic interface. **e** Cyclic voltammograms measured in Schottky system-based memristive devices (Pt,Cu/Ta$_2$O$_5$/Ta). The current densities show diode-like behavior. High currents are observed in positively biased condition (region I, energy diagram shown in **a**), and low currents in reverse bias condition (region II, energy diagram shown in **b**). The inset shows the current-voltage characteristics under high-revolution CV sweeps, indicating the presence of ionic currents originated from electrochemical redox reactions (region II). **f** Cyclic voltammograms in ohmic memristive systems. The high current densities shown at both positive (region III, energy diagram shown in **c**) and negative biased region (region IV, energy diagram shown in **d**) suggest that the Schottky-like barriers at metal/oxide interfaces are highly reduced. The pronounced current densities peaks imply strong redox processes at OE/oxide interfaces (see also Supplementary Figs. 17 and 18).

Schottky interface) which could lead to irreversible breakdown of the oxide layer by generating high concentration oxygen vacancies.

For low Schottky barrier height (cases (iii) and (iv)), the situation is similar for both voltage polarities with high electronic currents, parallel redox reactions and ionic currents, and lower level of passivation of the OEs (Fig. 2f, Supplementary Figs. 17 and 18). The blocking effect in reverse bias region is low and the electronic currents are dominating the interface exchange.

Thus, high Schottky barriers appear not advantageous for the stability of the cells due to the enhanced rate of passivation of the OEs.

These qualitative models agree with the experimental CV characteristics shown in Fig. 2e, f and confirm the conclusions made by the analysis of the energy barriers ratio at the metal/oxide interface. Figure 2e shows cyclic voltammograms in a conventional VCM system using unformed 30-nm $Ta_2O_5$ switching film and high-work-function metals i.e., Pt and Cu, respectively, forming a Schottky-contact interface with the oxide[37]. At the same time, Pt is electrochemically inert i.e., no redox reaction is expected, whereas Cu can be oxidized and incorporated into the oxide. The CV characteristics for both devices are shown in Fig. 2e and Supplementary Fig. 16. At positive voltages, after overcoming the energy barrier, the current density increases exponentially with the applied voltage (Supplementary Fig. 16). The electronic current is dominating, covering the ionic currents. In contrast, on the negative voltage branch, the current density is profoundly reduced due to the high energy barrier, where electron transport across the metal−oxide interface is dominated by the Schottky-emission process[35,36].

Analyzing more carefully within the diode (blocking) region we found current density peaks in both devices under high-revolution CV sweeps, see insets in Fig. 2e, and Supplementary Fig. 16b, d. In the Cu/ $Ta_2O_5$/Ta devices, the reverse current density further increases with increasing reverse bias (Supplementary Fig. 16c, d), whereas in Pt/ $Ta_2O_5$/Ta devices, the current density goes to saturation (Supplementary Fig. 16a, b). We attributed the current density peaks to reduction reactions of metal ions and $H^+$ or $H_2O$[38].

The current densities of these reactions, however, decrease with the increasing number of CV sweep cycles (Supplementary Fig. 16). We relate this effect to the parallel anodic oxidation of the Ta bottom electrode, forming a thicker oxide layer that suppresses the total current[27]. Not surprisingly, the subsequent CVs show reduced current density as higher electric potentials are increasing the oxide film thickness.

The electrochemical behavior in ohmic memristive systems is principally different. Figure 2f depicts the CVs of differently polarized (30-nm thick) $Ta_2O_5$ ohmic cells: Ta vs. Hf, Hf vs. Ta, Ta vs. Ta, and Zr vs. Ta. The XPS depth spectra confirmed the top and bottom OEs are metallic (Supplementary Fig. 15). We observed high current densities in both forward and reversed biased conditions (see also Supplementary Figs. 17 and 18), indicating low Schottky barrier height at the top and bottom electrode interfaces. Hence, higher concentration charge carriers can flow across the metal/oxide interface. The ionization of the OE is clearly evidenced in the Ta vs. Hf system, where distinct oxidation current density peaks were observed (Supplementary Fig. 17) when Ta electrode was positively biased. This corresponds to redox processes at Ta/$Ta_2O_5$ interfaces. At a higher voltage region, the current density peaks in Hf vs. Ta, Ta vs. Ta, and Zr vs. Ta system were also observed (Supplementary Fig. 18b, d, f), suggesting that redox reaction of Hf, Ta, and Zr OEs were also taking place. The interfacial redox interaction (ionization of OE and partial reduction of metal oxide) results in significant decrease of interfacial barrier height, leading to ohmic-like electrical contact[39]. Moreover, in contrast to the CVs in Schottky VCM systems (Pt/$Ta_2O_5$/Ta, see Fig. 2e), the CVs in ohmic memristive devices show much higher current densities in the low voltage (−2 to 2 V) region (Fig. 2f).

Due to the complexity of the system (multiple redox processes), the clear assignment of the redox peaks is challenging. Possible electrode half-cell reactions are summarized in Supplementary Table 2. Moreover, the electrode reactions lead to the incorporation of additional ionic species that can be considered as mobile donors/acceptors. Thus, exponential increase in the concentration of ions/vacancies can lead to same exponential increase of the electronic charge carriers and even changing the interfacial energy barriers, making analytical description of the systems extremely complicated.

## Resistive switching mechanism and kinetics

The mechanism of the resistive switching in the ohmic memristive devices is based solely on electrochemical reactions of partial oxidation and reduction of the filament and no Schottky barrier height modulation is involved. The switching is filamentary type as observed in the TEM images (Fig. 3). Depending on the applied voltage polarity, two different SET mechanisms were observed, showing the importance of the electrochemical properties[32] of the electrodes. The crucial factors in determining the mechanism are the difference in the energy barriers at both interfaces, the standard redox potentials of the half-cell reactions (see Supplementary Table 2) and the type and transference number of the mobile species.

Applying voltage bias to the Ta electrode results in pronounced oxidation and reduction currents as shown in Fig. 2f, and Supplementary Fig. 17 indicating the electrochemical ionization reactions. We conducted cross-sectional TEM and confirmed the formation of Ta-rich conduction channel in the oxide layer. Figure 3a shows the HAADF TEM image of the ohmic memristive device at LRS. As indicated by the dashed oval, a nanoscale channel was detected within the thin $Ta_2O_5$ film, connecting the Hf and Ta electrodes. The nanoscale conduction channel is responsible for the robust c8w resistive switching observed in the ohmic memristive devices (see Figs. 1d, e and 3e). The EDS elemental mappings show the composition of the conduction channel is rich in tantalum (Fig. 3b), and deficient in oxygen (Fig. 3c). This is further evidenced by the EDS line-scan analysis, as shown in Fig. 3f. No incorporation of Hf into $Ta_2O_5$ film was detected inside or outside the conduction channel (see Fig. 3d). It should be noted that the conduction channel has highest diameter (~3 nm) at Hf interface, and the diameter gradually decreases when approaching Ta electrode. This is in line with the classical reduction electrode reaction, in which the Ta cations first nucleate at Hf electrode (cathode) interface, followed by the growth process towards Ta (anode). The negative standard electrode potentials of $Ta^{3+}$/Ta (−0.6 V) and $Ta^{5+}$/Ta (−0.75 V) and negative free energy of oxide formation ($Ta_2O_5$, −760.5 kJ $mol^{-1}$) indicate the Ta nucleus has a high tendency to get oxidized, creating Ta-rich (O-deficient) conduction channel. The read currents at LRS and HRS show linear dependence with applied voltage, indicating that the electron transport in the oxide is dominating conduction mechanism Moreover, the HRS and LRS currents increase with increasing temperature, suggesting the conduction behavior is non-metallic (Supplementary Fig. 19). Additional TEM images and quantitative analysis also reveal other, partially formed Ta-rich conduction channels at Hf interface (Supplementary Fig. 20a).

The transient electrical characteristics of the optimized Hf/$Ta_2O_5$/ Ta device were monitored in pulse measurement mode, varying the SET voltage amplitude ($V_{SET}$). The resulting SET time ($t_{SET}$) is defined as the time difference between the half of the SET pulse rising edge and the half of the SET current rising edge (Supplementary Fig. 21). Figure 4a depicts the semilogarithmic plot of SET time as a function of applied SET voltage. Three regions can be distinguished as indicated at the linear fit: region I from $V_{SET}$ 0.35 to 0.8 V, with a slope of −50 mV; region II from $V_{SET}$ 0.8 to 1.2 V, with a slope of −155 mV per decade; region III where $V_{SET}$ is above 1.2 V, and the slope is −1100 mV/decade. The results fit well with the model presented previously for ECM devices, distinguishing three electrochemical processes: nucleation of metal ions at low voltage regime, electron-transfer reaction at intermediate voltage regime, and ionic transport for high voltage regime. The nucleation time of the metal ions at the counter electrode is given as[40]:

$$t = t_0 \left[ \exp\left( \frac{(N_c + \alpha_c)ze\Delta\varphi_c}{kT} \right) \right] \quad (7)$$

where $t_0$ is a pre-dominant factor depending on the cation concentration and number of active sites for nucleation, $N_c$ is the nucleus

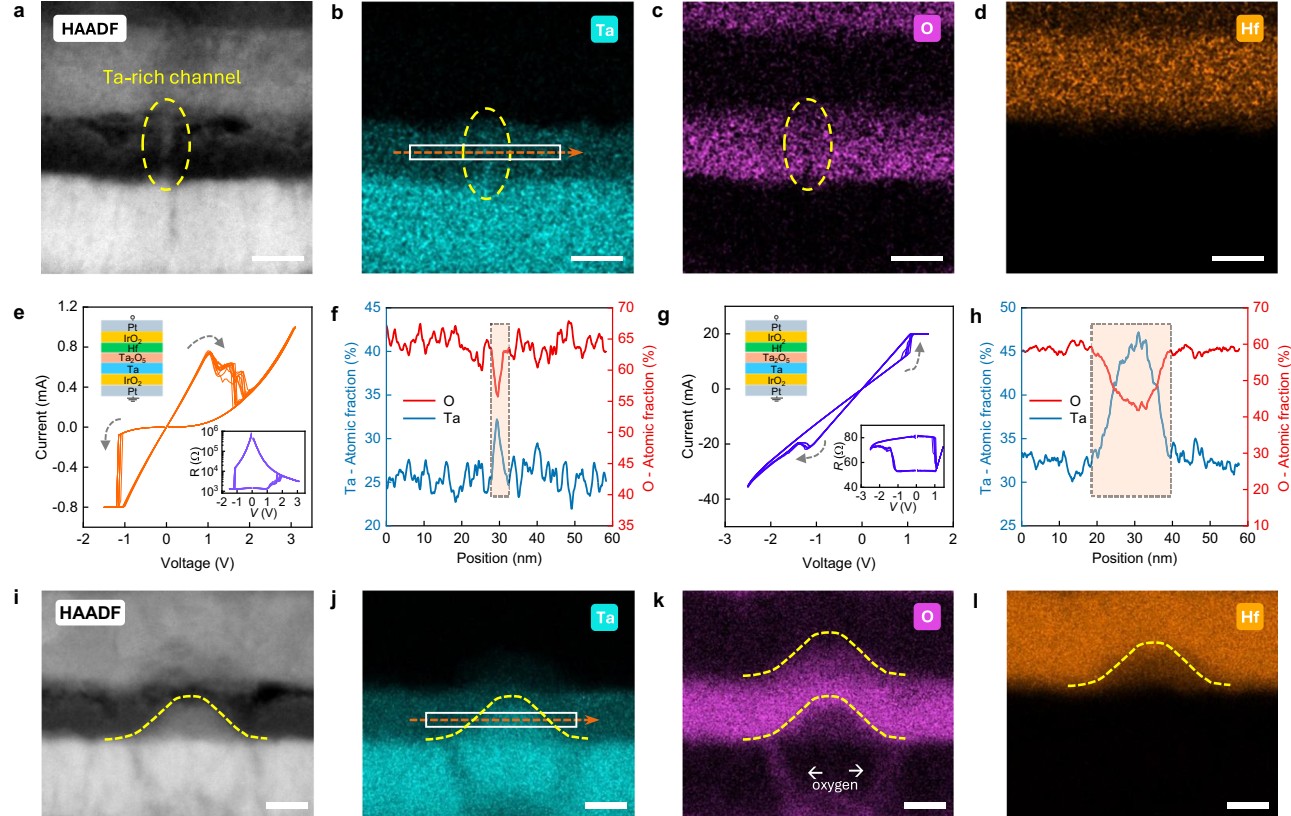

**Fig. 3 | Observation of conducting channel by cross-sectional transmission electron microscopy. a** HAADF TEM image of the nanoscale Ta-rich conduction channel. **b–d** Corresponding EDS elemental mapping of Ta, O, and Hf, respectively. **e** c8w *I–V* curves obtained from the ohmic memristive device when Hf metal electrode were negatively biased (Ta electrode positively biased) for SET operation. The inset shows the resistance as a function of applied voltage. **f** EDS line-scan analysis of the conduction channel region (oval area in **b**). **g** *I–V* characteristics of the ohmic memristive device when operating at high current and voltage ranges. This leads to another SET mode resulting in 8w switching polarity (see also

Supplementary Fig. 3f). Both HRS and LRS are highly conductive. The inset shows the corresponding *R-V* plot. **h** EDS line-scan analysis (see oval area in **j**) of the conduction channel region, showing that a larger Ta-rich filament is formed in ohmic memristive devices when operated in 8w polarity (as shown in **g**). **i** HAADF image showing Ta-rich cluster observed in 8w switching polarity. **j–l** EDS elemental mapping of Ta, O, and Hf, respectively. Scale bars, 10 nm. It can be clearly observed that the O migrates from the bottom electrode to the anode (**k**), generating oxygen-deficient and tantalum-rich filament at the bottom electrode interface.

number, $\Delta\varphi_c$ is the cathodic potential ($\varphi_c$<0), $\alpha_c$ is the cathodic transfer coefficient. Equation 7 suggests an exponential dependence of the nucleation time on applied potential, that is, nucleation time decreases exponentially with increasing (absolute) cathodic potential. This agrees with the linear fitting in the region I (Fig. 4a). We employed the slope 50 mV to Eq. 7 to calculate $(N_c + \alpha_c)$, and got $(N_c + \alpha_c) = 0.24$. Thus, the number of atoms in the critical nucleus $N_c$ is derived to be 0, indicating each metal cation that is reduced at the cathode can further grow as a new phase. Nevertheless, we cannot clearly distinguish from the Tafel plot whether we have nucleation limitations with $N_c$ taking statistically 0/1, or electron-transfer limits at the metal/oxide interface. The value for the transfer coefficient alpha is around 0.2, comparable to that determined for other ECM systems. From the applied electrical polarity, we attributed the nucleus to Ta, which was generated through the reduction of Ta cation under the cathodic potential. Using the same method, we got $(N_c + \alpha_c)$ - 0.1 in region II. Here we assume the charge transfer control is dominating as under voltage pulse, we observed quantized conductance (Fig. 4b) that rather suggests that charge transfer (that also can be influenced by Joule heating) is the limiting step. From the cyclic voltammograms and SET kinetics, we are convinced that the ionization of the Ta electrode, nucleation of the Ta atom, and the formation of Ta-rich conduction channel are crucial in conductance modulation in the fabricated ohmic memristive devices (Fig. 4c). The results also indicate Ta has similar electrochemical mentalization dynamics with Cu that can be ionized and incorporated

into the oxide layer. This is further evidenced by the electrical characteristics observed in Cu/Ta$_2$O$_5$/Ta devices (Supplementary Fig. 22).

Based on the information from the cyclic voltammograms, *I–V* sweeps, and cross-sectional TEM characterization, we concluded that the SET/RESET transitions in the reliable counter-eight-wise (c8w) switching originate in the conductivity change of the filament. This change of the conductivity and therefore, of the resistance is determined by two factors as shown in Fig. 4c—the oxidation state of the metal ions and related TaO$_x$ stoichiometry, and the length. The oxidation state of the TaO$_x$ is changed by the redox reaction, which is a function of the applied voltage. Ta-ions in the filament of lower oxidation state (e.g., 4+ and/or 3+ and/or 2+ and/or 1+) are oxidized to higher oxidation state e.g., Ta$^{5+}$, accompanied by additional incorporation of oxygen ions and the oxide transits to its stoichiometric form. The oxidation will affect not only the first layers of the tip but also expand, reducing the effective length of the (TaO$_x$) filament, being however, not completely dissolved (transformed to Ta$_2$O$_5$) during this process. The shorter the length the higher the resistance. The change in oxidation state/stoichiometry and length is determined by the redox reaction rate and mobility of the ions, where these two processes occur most probably in parallel. The differences in FCM mechanism, compared to ECM and VCM switching processes are summarized in Supplementary Table 3.

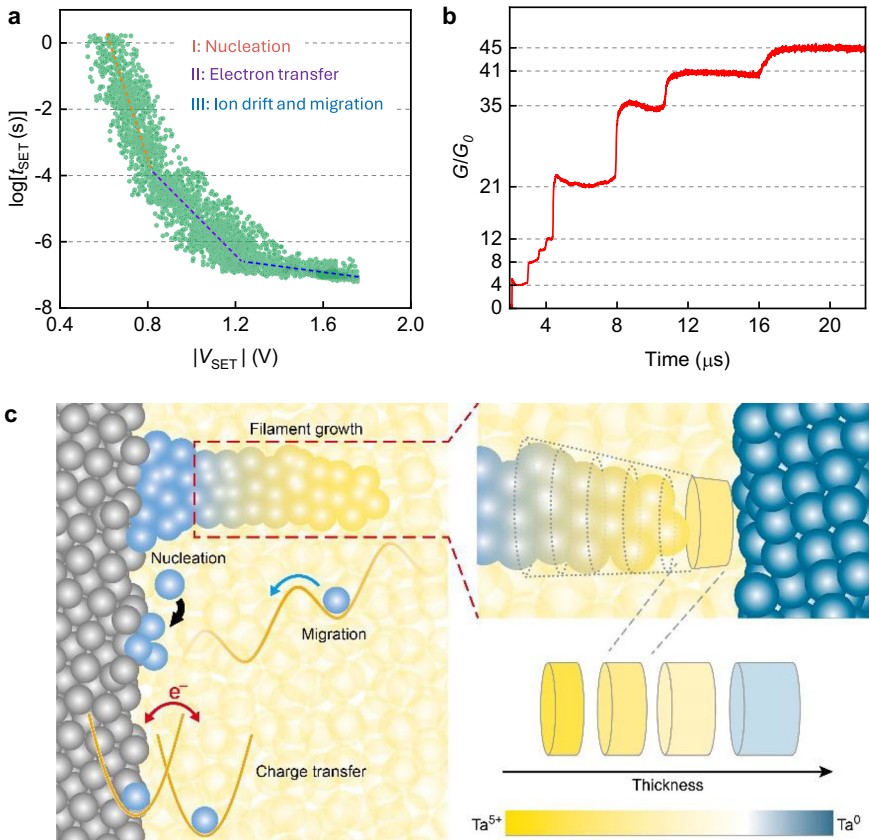

**Fig. 4 | Switching kinetics. a** Switching time ($t_{SET}$) versus applied SET voltages. The fittings of experimental data indicate three dominating factors. **b** Quantized conductance change in the ohmic memristive device. **c** Illustration of the underlying electrochemical redox processes, energy diagram for memristive switching. The various lengths and oxidation states of the conduction channel which can be achieved by the electrical programming enable the multi-bit switching capability, as shown in Fig. 1f–h.

Superior analog switching capability (Fig. 1f–h) of the ohmic memristive devices can be obtained depending on the *I–V* sweeping and pulse programming conditions. This is enabled by the ability of the ohmic memristor to form conduction channels of different lengths, and with different valences in the oxide (Fig. 4c, right panel). These two factors in combination allow to achieve various conductance levels in the device.

The second electrochemical mechanism was observed when the devices were operated at high current and voltage ranges (Supplementary Fig. 3e, f) in already-formed devices. This results in the transition from c8w switching to 8w switching polarity with highly conductive HRS and LRS (Fig. 3g). As we were not able to identify Hf conduction channels from the TEM measurement, we concluded that no Hf-ions are injected into the switching film. Instead, a large oxygen-deficient cluster was found in $Ta_2O_5$ (Fig. 3h–l). At the same time a decrease in Hf concentration (Fig. 3l) and an increase of oxygen concentration (Fig. 3k) at the Hf electrode interface was observed, implying the migration of oxygen anions ($O^{2-}$) towards Hf and the formation of substoichiometric hafnium oxide. Thus, in this voltage polarity the conduction channel seems to be predominantly formed due to scavenging of oxygen ions from the thin $Ta_2O_5$ (~8 nm) layer into Hf electrode, generating a large oxygen-deficient $TaO_x$ cluster in the oxide layer as shown in Fig. 3h–l and Supplementary Fig. 20b (more TEM images, EDS elemental mapping and line scans are shown in Supplementary Figs. 23 and 24). The formed Ta-rich cluster is highly conductive, explained by the high oxygen scavenging capability and negative defect formation energy ($-1.5$ eV)[23,41] of the Hf metal electrode. We also observed the migration of oxygen ions from the $IrO_2$ layer toward the upper layers (Fig. 3k). In contrast to Hf, Ta has positive formation energy of oxygen vacancies (0.1 eV) when in contact with $Ta_2O_5$[41], making it difficult to create oxygen vacancy defects in the oxide layer. Despite resistive switching using this voltage polarity is possible, the devices suffer from high currents (Fig. 4g) and limited stability, which can also be justified by the larger size/volume of the oxygen-deficient filament (Fig. 3h–l, Supplementary Figs. 23 and 24).

The difference in the two mechanisms and respectively SET modes can be explained by the energy diagram shown in Fig. 2c, d, accounting for the standard redox potentials and dominating mobile species. $Ta/Ta_2O_5$ interface has a less negative standard electrode potential, compared to $Hf/Ta_2O_5$. Applying a positive voltage to Ta (negative to Hf) allows for electronic currents through both interfaces and does not enhance the oxidation reaction of Ta, being ionized and incorporated into the oxide layer. Here the ions responsible for the filament formation appeared to be predominantly Ta-ions. In contrast, when positive bias is applied to Hf (negative to Ta) the much more negative standard electrode potential of $Hf/HfO_2$ half-cell reaction (Supplementary Table 2), is a reason for enhanced electrochemical oxidation of Hf (as evidenced by Hf EDS spectra, see Fig. 3f). Here oxygen ions appear to be predominantly responsible for the switching as TEM images have not detected Hf-ions within $Ta_2O_5$. This is also a reason for the different shapes and sizes of the filaments and also demonstrates that Hf-ions are not incorporated into the filament, and have no influence on its defect chemistry and resistance. Moreover, in contrast to eight-wise (8w) switching (Fig. 4g) that relies on high currents operation, the c8w operation (Figs. 1d and 4e) provides more

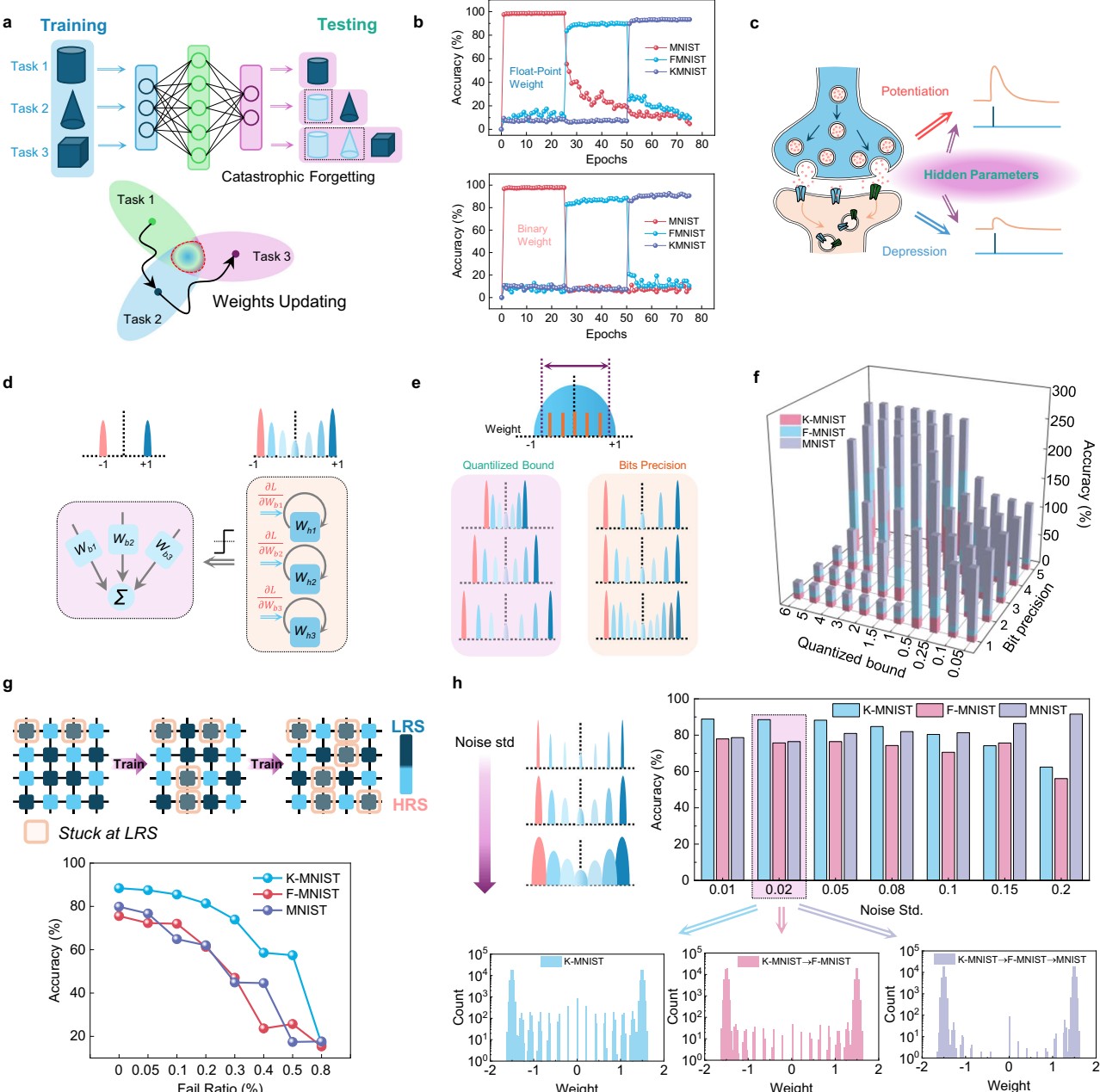

**Fig. 5 | Overcome catastrophic forgetting in deep neural networks.**
**a** Schematics of catastrophic forgetting problems when neural networks are trained by different tasks sequentially. **b** The testing accuracies for three different tasks during the sequential training of the neural networks, whose weights were set as float point or binary values respectively. **c** Schematics of the hidden states within synapses, which can influence the plasticity of each synapse. **d** Diagrams of the neural networks implementation inspired by the higher order synaptic plasticity. **e**, **f** Final testing accuracies of all the trained tasks concerning the quantized bounds and bit precision. **g** Neural network performances for different devices' fail ratios during the continual training process. **h** The impacts on the continual learning capacity of the neural network model under the different multilevel programming errors.

reliable and stable switching characteristics. The positive oxygen vacancy defects formation energy of Ta also suppresses the generation of defects which may lead to device failure[23].

**Neural network application**
On the basis of well-defined resistive switching mechanisms and electrical characteristics, the ohmic memristors were programmed in a versatile way between two (binary) or more states (analog) in an adequate condition, as shown in Fig. 1d–h. The binary and analog switching capability in the ohmic memristive device enables energy-efficient neuromorphic computing hardware acceleration which

requires different-precision memory modules. In the human brain, some neuroscience research[42–44] suggested there existed higher order plasticity variables that can control synaptic weight changes, which were also termed as "metaplasticity". The multiple degrees of plasticity enable advanced cognitive abilities, one of which includes learning new tasks continually without forgetting old contents. However, for most of the modern deep neural networks, it is hard to keep "memory" for the learned tasks when trained by a new task, one of the most important reasons is that the networks' weights were optimized into another subspace to accomplish this new task without protected within previous subspaces (Fig. 5a)[45]. Only when the networks' weights

converge to the intersection of these tasks, can the network finally learn all the trained tasks simultaneously.

Recently, synaptic metaplasticity in binarized neural networks has been introduced to overcome catastrophic forgetting problems[46]. In this neural network model, there were two groups of different-precision weights, including floating-point hidden weights and binary-value inference weights. To test if the model can learn different tasks continually, three nonoverlapping datasets, involving the Modified National Institute of Standards and Technology (MNIST) database, Fashion MNIST (F-MNIST), and Kuzushiji MNIST (K-MNIST) (Supplementary Fig. 25)[47–49], were fed into the neural network sequentially. The superior characteristics and functionalities measured on individual ohmic memristors were implemented in a neural network model simulation. Firstly, we studied the neural network performances when their weights were just used floating-point values or binary values (Supplementary Fig. 26, details can be found in the Supplementary Materials). The results presented in Fig. 5b show that the catastrophic forgetting problems still cannot be avoided. We then incorporated the two different switching modes using the unique properties of our ohmic memristors, to emulate the modulation effects between synaptic plasticity and hidden parameters behind the biology synapses (Fig. 5c), in which the hidden parameters can decide the degree of the synaptic weights' change. The forward and backward procedures for the calculation process were based on different-precision weights (Supplementary Fig. 27), in which the backward updating steps included a nonlinear function $f_{\mathrm{meta}}$ that will give a certain punishment when the signs of the hidden weight and the gradient are the same, which is aiming to reduce the proportion of weight sign changes in the binarized neural network and keeps the network performances stable on the trained tasks. In conventional deep neural networks, in addition to reloading old tasks, which will cause much more extra storage resources and time wasting for retraining, the pre-trained weights between two neurons are hard to keep "memory" for previous training experience after being trained by the new tasks. Therefore, we adopted hidden weights behind inference weights to control their updated directions, the analog-value weights represent the importance factors of the corresponding inference weights about the old tasks, in which the larger the absolute value of the hidden weights is, the more important the associated inference weights are with respect to the trained tasks. Through the updated punishment under the nonlinear function when the backward gradients will cause the sign changes of the inference weights, the important weights can be protected. As illustrated in Fig. 5d, through different operation ranges to control the conductive filaments formation, the ohmic memristor can simultaneously work in the binary-switching mode or the multilevel transition, which can separately act as inference weights and hidden weights.

In other words, the hidden weights store the memory for all the trained tasks, which require floating-point storage format in the algorithms and should be retained without memory loss, however, it will cause much more latency and memory sources when this model is employed in conventional computing architecture. By adopting the energy-efficient computing-in-memory (CIM) architecture[50,51], with both hidden weights and inference weights stored by the ohmic memristor, it can enable more compact and energy-saving continual learning (Supplementary Fig. 28a).

In the hardware mapping of the hidden weights, we quantized them into limited states to avoid floating-point storage wasting, when considering that the hidden weights are also the variables of the meta function $f_{\mathrm{meta}}$, the quantized bounds and bit precision both have impacts on the neural network performances (Supplementary Fig. 29). Figure 5e, f illustrated the final testing accuracies of all the trained tasks about the quantized bounds ranging from 0.05 to 6 and bit precisions with the highest precision set at 5-bit, and the digital baseline

performances of the neural network can be found in Supplementary Fig. 28b, in which all the weights are implemented at the software platform. At the largest fixed quantized bound, high bit precision can guarantee better continual learning results with lower precision loss of the hidden floating-point weights. For the quantized bit precision, it can cover most of the studied quantized bounds from 3-bit to 5-bit, which all can be easily achieved by our ohmic memristors, more information about the network performances with different quantized bounds and bit precisions can be found in Supplementary Figs. 30 and 31. When the quantized bounds are limited within small areas, such as smaller than 0.25 in Fig. 5f, followed by the meta function, the value of $f_{\mathrm{meta}}$ almost keeps at 1, which nearly loses the regulation capacity for the analog weight updating and continual learning ability, so it is necessary to the set the quantized bound in a proper range.

The ohmic memristors have excellent endurance properties, and the low failure ratio can help neural networks keep better continual learning capacity. The results are shown in Fig. 5g, and the simulation details can be found in Supplementary Note 2 or Supplementary Fig. 32. As the failure ratio increases from 0.05% to 0.8%, the network implemented by the memristors almost loses continual learning capacity, in which the number the programmable devices is decreasing. On the other hand, when we increase the training epochs for learning each task under the 0.1% fail ratio, as shown in Supplementary Fig. 33, the final network performances will also degrade for the less programmable devices. So, as for this kind of continual learning application required for frequent training, the excellent stability of ohmic memristors is more suitable for it. The comparison of neural networks performance between conventional VCM and ohmic memristors is shown in Supplementary Fig. 34.

Given the realistic array implementation, the effects of line resistance and cell shape design were explored in Supplementary Figs. 35 and 36, owing to the relatively small neural network structure, the hardware design can tolerate a large range of line resistances. When considering the programming errors about the target conductance, we conducted experiments exploring the relationship between network performances and writing noises ranging from 0.01 to 0.2. As illustrated in Fig. 5h, the model can keep learning continually when writing noise standard variation is less than 0.15. For picked write noise standard variation at 0.02, the evolution of the weight distributions in the hidden layer in the training process indicated the model can not only have consolidated weights for learned tasks but can also have unconstrained weights for learning new tasks. Other final weight distributions of the hidden layer for the different programming errors were shown in Supplementary Fig. 37, the obvious overlapping between different states illustrated the robustness of the model to tolerate some noises in the hidden weights to a greater extent, which can save much cost in writing and verifying the analog hidden weights.

In summary, we have reported a new type of ohmic memristive device based on an oxide film sandwiched by two electrodes with low-work functions. The use of tantalum as an active electrode, and the minimization of Schottky barrier heights at metal/oxide interfaces provide improved electrical stability in device performance such as cycling uniformity, multilevel switching, retention, and endurance. We analyze in detail the interfacial energy barriers and demonstrate that for reliable switching one does not necessarily need Schottky barrier and barrier height modulation, but solely an asymmetry in the electrochemical interfaces and redox reactions. The mixed-precision neural network computing based on configurable switching modes between binary and multilevel can help overcome the catastrophic forgetting problem, and the robustness of network performances got verified by discussing myriad potential impacting factors. Our findings provide a new methodology for the design paradigm of memristive devices and will further advance the development of electronics for computation-in-memory applications.

## Methods

### Devices fabrication

The samples fabrication starts from ultrasonic cleaning of the thermally oxidized $SiO_2$ (430 nm)/Si wafers (one-inch diameter) in acetone, isopropanol, and deionized water for 10 min, respectively. Afterwards, a sequence of optical lithography, layer deposition, and lift-off processes were conducted to structure the geometry of the crossbar devices. The active area of the fabricated devices ranges from 4 to 2500 $\mu m^2$. The electrical results reported in the main text and Supplementary information were collected from $50 \times 50$ $\mu m^2$ size devices. For TEM characterization, the devices have a junction size of $2 \times 2$ $\mu m^2$. For the uncapped ohmic memristive devices (no capping layers were used for the top and bottom electrodes), the Ta electrodes were deposited directly on $SiO_2$/Si wafers and were used as the bottom electrodes. The $Ta_2O_5$ layer was subsequently deposited, followed by lithography and top electrode deposition, eventually leading to devices with Hf(Ta, Zr)/$Ta_2O_5$/Ta stacks (Fig. 1b). Note that in the absence of capping layers, as the top electrodes suffer from strong passivation when exposed at ambient condition (see Supplementary Fig. 2), we measured the devices immediately after the deposition of top electrodes and the lift-off process. To improve the electrical stability, $IrO_2$/Pt layers were deposited in-situ to protect the OEs from passivation, leading to Pt/$IrO_2$/Hf/$Ta_2O_5$/Ta/$IrO_2$/Pt layer stacks. Pt/TiN/Cu/$Ta_2O_5$/Ta/$IrO_2$/Pt devices (see Supplementary Fig. 22) were also fabricated using the same fabrication flows. For the comparison of resistive switching characteristics, conventional VCM devices with Pt/$Ta_2O_5$/Ta(Hf) structure were fabricated, where Ta and Hf electrodes were used as top electrodes, respectively. For all memristive devices, the thickness of the Pt, Ta, Hf, and Zr electrodes is 30 nm, the thickness of the $Ta_2O_5$ resistive switching layer is ~8 nm, and the thickness of the $IrO_2$ layer is 15 nm. The CV measurements were performed on 30-nm thick $Ta_2O_5$ before the forming process. Thin film deposition was realized by the magnetron sputtering technique. The Pt, Ta, Hf, and Zr electrodes were deposited by radio frequency (RF) magnetron sputtering. The Cu electrode was deposited by electron-beam evaporation, followed by TiN (DC magnetron sputtering) and Pt (DC magnetron sputtering) capping layers deposition. The $Ta_2O_5$ resistive switching layer was reactively sputtered by RF magnetron sputtering using Ta metal target in mixed Ar (60%) and $O_2$ (40%) atmosphere. The $IrO_2$ buffer layer was reactively sputtered by RF magnetron sputtering using an Ir metal target in a mixed Ar (90%) and $O_2$ (10%) atmosphere. The purity of the Ta, Zr, Hf, Cu, Pt, and Ir metal targets is higher than 99.95%. Details about layer deposition conditions are shown in Supplementary Table 4.

### Electrical measurements

Cascade SUMMIT 9600 probe station was used for electrical characterization. The CV measurements were carried out using Keithley 6430 Sub-FemtoAmp Remote SourceMeter with a triaxial cable connection. This system allows providing triangular voltage sweep with sweeping rate ranges from 1 to 3000 mV s⁻¹. For cyclic voltammograms, 130 mV s⁻¹ were used. Potentiodynamic $I$–$V$ sweeps were performed using Keithley 2636 A SourceMeter with triaxial cable connections. In total 230 ohmic memristive devices have been measured. Pulse measurements were conducted using Keithley 4225 ultrafast pulse measure units with Keithley 4200 semiconductor parameter analyzer and Agilent B1500A semiconductor parameter analyzer. Wavetek 395 Arbitrary Waveform Generator was also used for applying voltage pulses to the devices. In this case, the current signals were recorded in real-time by monitoring the voltage drop across the input channel (50 $\Omega$ shunt resistor) of the Tektronix DPO7254C storage oscilloscope. Unless otherwise specified, the voltages were always applied to the top electrode of the device. The devices were measured in ambient condition at humidity of ~35%.

### X-ray photoelectron spectroscopy

The XPS experiments were performed with Phi5000 VersaProbe II (ULVAC-Phi Inc.) system using a monochromatic aluminum K-alpha ($E_\lambda = 1.486$ keV) X-ray source.

### TEM characterization

To characterize the chemical change of the switching matrix, cross-sectional TEM characterization was conducted with Talos F200X TEM. The energy-dispersive X-ray spectroscopy was also employed to reveal the specified elements distribution and variation by Super-X with SDD detector (Thermofisher). Before the cross-sectional TEM characterization, the devices were programmed into LRS. Then the devices were cut by a focus ion beam followed by lift-out processing (Thermofisher Helios G4 UX DualBeam). Subsequently, the obtained lamellae were replaced on the TEM grid for inspection.

### Metaplasticity-inspired continual learning

By incorporating the "metaplasticity" concept in neuroscience into a binary neural network (BNN) during the training process, the dynamic changes of neural network weights can be modulated, which aims to overcome catastrophic forgetting problems[46]. In the specific hardware acceleration of this algorithm, the CIM architecture and mixed-precision memristive synapses are both the major points. Based on the well-defined ohmic memristor that uses low-work-function OEs, the configurable and stable binary and analog switching could be adopted as inference weights and hidden weights in the framework of metaplasticity-inspired continual learning.

As shown in Supplementary Fig. 28, the main body of this algorithm is the BNN, whose weight values and neuron activations are limited at 1 or −1, the differences between that with a conventional BNN are reflected in the training process whose updated gradients need to be multiplied by a nonlinear function. The nonlinear function $f_{meta}$, which is also referred to as meta function, is relevant to the hidden weight values and one hyperparameter $m$.

To test the continual learning capacity of the network based on the devices' characteristics, the chosen datasets include MNIST, F-MNIST, and K-MNIST, which are in the same image size of $28 \times 28$ and the same 10 categories, so the network structure needs no change during the training sequences.

In the comparison with the conventional neural networks, which used only float-point weights or binary weights, the network structure was set as 784-4000-4000-10, 10 fully-connected format, in which the numbers of neurons in the two hidden layers were both set as 4000, and the numbers of neurons in the input layer and output layer were fixed at 784 and 10. The large neural network can better confirm the necessity of metaplasticity in overcoming the catastrophic forgetting problem. When only float-point weights were used in the neural network, there would be no conversion between the analog weights and binary weights and inference weights were analog values; similarly, for only using the binary weights, the training and inference were both based on the binary value.

For all the simulation experiments of continual learning, fully-connected neural network structure was kept as 784-500-200-10 (Supplementary Fig. 28), in which the numbers of neurons in the two hidden layers were set as 500 and 200, respectively, and the numbers of neurons in the input layer and output layer were fixed at 784 and 10, respectively. In the above experiments, the noise levels added at the hidden weights were all in the standard variation of 0.05. For the research into the impacts of the network performances under the different multilevel programming errors, the quantized bound and bit precision were set 1.5 and 3, while the hyperparameter $m$ was still kept at 3. When considering the differential conductance distribution errors (Supplementary Fig. 6), for all the experiments, the weight values transfer from hidden weights to binary inference weights were all

added extra Gaussian noise at an average standard variation level of 0.1457.

## Endurance requirement for continual learning

For continual learning applications, the weights need to be updated continually along with the network optimization, hence the high endurance of devices is crucial for the online training. The endurance is associated with the resistance-switching window, therefore the endurance for the binary-switching devices is the bottleneck in this application. Generally, the maximum number of the binary resistance switching can be derived as:

$$N_{\max} = \sum_{i=1}^{T} \frac{S_{total,i}}{S_{batch,i}} \times E_{train,i}$$

Where $S_{total,i}$ is the total number of samples in the $i$th trained task, $S_{batch,i}$ is the batch number of samples in the $i$th trained task, $E_{train,i}$ denotes the training epochs for the $i$th trained task, hence the endurance of the device should be more than the maximum binary-switching number $N_{\max}$ during training all the tasks from the first to the $T$th. By taking the experimental values in this work into the equation and assuming that the parameters of each dataset are consistent, the endurance of more than $10^6$ can be trained for more than 83 tasks, which can meet most needs in the edge platform, and the maximum number of trained tasks can be further increased by adjusting related parameters.

## Data availability

All data that support the findings of this study are available within the paper and Supplementary Information file, or are available from the corresponding authors upon request. Source data are provided with this paper.

## Code availability

The codes that support the findings of this study are available in Zenodo with the identifier https://doi.org/10.5281/zenodo.14849157.

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

## Acknowledgements

We thank D. Erdoglija and P. Grewe (IWE2, RWTH Aachen) for their assistance with sample preparation and T. Pössinger for the figure graphics. We thank J. Mohr (IWE2, RWTH Aachen) for fruitful discussions on electrical characterization. This work has been funded in part by the project (EMPIR 20FUN06 MEMQuD) that has received funding from the EMPIR program co-financed by the Participating States and from the European Union's Horizon 2020 research and innovation program. S.C. acknowledges ICYS research fellowship from NIMS. Y.Y. and Z.Y. acknowledge support from the National Natural Science Foundation of China (61925401, 92064004, 92164302), the 111 Project (B18001), and the Tencent Foundation through the XPLORER PRIZE. The work was supported by the Guangdong Provincial Key Laboratory of In-Memory Computing Chips (2024B1212020002).

## Author contributions

S.C. and I.V. generated the idea, designed the experiments, and analyzed the data. I.V. and Y.Y. supervised this project. S.C. fabricated the devices. S.C. conducted the electrical characterization with the contribution from Z.Y. H.H. and A.B. conducted XPS analysis. Y.Y. and Z.Y. conducted the TEM characterization and designed the artificial neural network. S.C. and I.V. wrote the manuscript. All authors discussed the results and contributed to the revision of the manuscript.

## Funding

## Competing interests

The authors declare no competing interests.
