## [Transparent Peer Review file · Nature Communications]

Electrochemical ohmic memristors for continual learning

Corresponding Author: Professor Ilia Valov

Version 0:

Reviewer comments:

Reviewer #1

(Remarks to the Author)

In this manuscript, the authors present resistive switching devices with two ohmic contacts and claim that they demonstrate a new switching mechanism. Indeed, the majority of resistive switching devices use at least one non-ohmic electrode and there are reports where two ohmic electrodes do not lead to resistive switching. The electrical resistive switching data look very promising, the manuscript is mostly well-written, and the breadth of investigations is commendable. Yet I have a hard time placing the manuscript on the map and I am not convinced of the switching mechanism.

- Firstly, the presented IV curves look very similar to, e.g., Fig. 2.3 in <https://www.sciencedirect.com/science/article/pii/B9780081027820000022>. In both cases, the LRS is pretty much linear, which can be explained simply by the conductive filament in this state. The authors' claim to this configuration in the LRS seems sound and the TEM supports the metallic filament explanation. However, the HRS is not analysed further. If the filament is ruptured in the HRS, what would the authors call the 'blocking' part in this state? It is probably a part of oxide or interface without the metallic filament. Whether at the interface or slightly apart from it, there will be an energy barrier for electrons between the filament and the filament-free oxide part. The junction between a metallic part (filament) and an insulating part is effectively something like a Schottky barrier, so I don't really understand the authors' claim to a Schottky-barrier-free new mechanism. The SI itself indicates there is an expected band offset of about 1 eV between the ohmic electrodes and the tantalum oxide film. One simple way to analyse the HRS further is fitting an electronic conduction model to it, possibly Schottky emission ... In broader terms, I am not convinced whether there is a new mechanism at play here, or whether it is 'just' a filamentary switching mechanism with low work function electrodes. The latter is surely a great achievement in itself, but would significantly alter the claim of the manuscript.
- Related to this, have the authors checked whether their devices can be operated simply as unipolar devices? In addition to the comment above, the two halves of the IV curve look very much like unipolar switching curves, cp., e.g., Fig. 4 in <https://www.tandfonline.com/doi/full/10.1080/00018732.2022.2084006> It would be helpful to have this in the SI, as it would support the later mechanism discussion if the devices do not work as unipolar.
- Also related, on p. 10, "The robust device characteristics suggest that the occurrence of resistive switching in the ohmic memristors does not rely on Schottky barrier height modulation." This is not a self-evident conclusion. Very robust resistive switching has been attributed to Schottky barrier control, e.g., recently <https://www.science.org/doi/full/10.1126/sciadv.adg1946>
- If the devices are filamentary switching with low work function electrodes rather than a truly different switching mechanism, the authors should present more data on device to device variability than SI Fig. S4. The main challenge of filamentary switching is reproducibility, so, reported improvement on such devices should demonstrate how they are better than existing filamentary devices.
- Also related to the mechanism, the analysis and discussion around Fig. 2 and SI Fig. S12 and S13 do not convince me of the 'new' mechanism either, and they do not really fit neatly with the electrical data. It seems like there is the electrical resistive switching data, supported by the TEM analysis, and as a second part there are the CV studies, but the connection is missing.
- Partly, the authors may be able to remedy this by clarifying their explanations. Partly, there are more fundamental concerns around Fig. 2a and SI Fig. S12:
 - o The insets of the CV in Fig. 2a and SI Fig. S12 are confusing. They cannot be just zoom-ins of the main panels, as there are features visible or different, which are missing in the main figures. For example, in Fig. 2a, in the main panel, the current

is just zero at all negative voltages, but in the inset it is clearly different. Are they maybe forward and backward sweeps instead of zoom-ins? Please clarify in the text and figure captions. (Also, rather “zoom-out”, as the voltage axis is larger in the insets?)

o Please also explain the difference between IV measurements and cyclic voltammetry.

o It is not entirely clear, which currents the authors ascribe to the electronic and which to the ionic processes. Is the ionic current only the little bump? What about the strong current increase after the little bump in the first cycles? Surely these are not ascribed to ionic processes, but electronic conduction? Firstly, the currents are much too high for ionic currents.

Secondly, the strong increase in the reverse direction is also visible in the Pt devices. With eq. 1, however, the electronic current in the reverse direction would be zero. This is in line with the previous comment that eq. 1 is not appropriate for the reverse direction.

o Eq. 1 is for a forward-biased metal-semiconductor diode. The authors already cite the book by Sze, which discusses this in detail in ch. 3. While for a forward-biased metal-insulator junction as in the presented devices, this is appropriate, it is not really for the reverse direction, which is also discussed in the book by Sze in ch. 4. (Essentially, charge injection in a reverse-biased Schottky contact will lead to image force lowering of the barrier and result in a dependence J proportional to $\exp(\sqrt{V})$). The rapid current increase in the reverse direction would support this.) With this, many of the authors' assumptions in the derivations of the expressions around Fig. 2 and SI Fig. S12 are not valid anymore.

o If the ionic current is only the little bump, the overall conclusion may still be valid, but the explanation needs to be updated.

o In SI Fig. S12, for the first cycle, the authors should add arrows to indicate the orientation of the considerable hysteresis. It does not seem self-evident whether the current actually saturates or whether the hysteresis is ‘the other way around’.

o If in the analysis of the ionic currents, Fig 2a and SI Fig. S12, the strong current decrease of the second and third cycle, with respect to the first, is due to Ta oxidation of the bottom electrode, surely this would also happen in the actual devices with thinner oxide layers? In that case they would not be as stable and good performance as demonstrated in Fig. 1, so this explanation of Ta oxidation seems inconsistent. The authors argue that indeed, the same happens in the Ta/Hf electrode devices, so why does this not affect the device performance in Fig. 1? It can't be due to the capping, because the authors argue that it is metal oxidation between the electrode and the oxide film.

• I have a hard time following the network simulations. It may simply be outside of my expertise, so that other reviewers may be more suitable to evaluate this. For reference, I would claim that I have a solid understanding of the fundamentals of resistive switching networks and their training and operation, but I am not an expert in the various advanced modes of operation. From this perspective, there is too little explanation in the manuscript to describe the complexity of the simulations and their results. It may be informative to experts on neural networks, but I am afraid the findings will be lost to a broader audience and that is a shame, given the hard work that the authors put in. As a suggestion, maybe the authors could present the main findings in the main manuscript in a more simplified and stringent way, but then include a more detailed discussion in the SI. This could be a way to make the results accessible to a broader audience. Either way, Fig. 5 is very busy and some of the schematics do not really help my understanding.

• Possibly based on my limited understanding, is the specific switching mechanism of the presented devices a necessary property to provide the simulated results? Or can the same functionality be achieved by devices with other switching mechanisms, e.g., any other multi-level resistive switching?

Some minor comments:

• The pulses used for endurance measurements were fairly long (milliseconds). The manuscript and the presented devices would be strengthened if the authors included measurements with shorter pulse widths, e.g., trying the limits of switching speed. The equipment listed in the methods part is certainly capable of such measurements.

• On p. 9, the statement of the metallic phase by EDS does not seem convincing. In SI Fig. S10, there is oxygen detected in the hafnium layer. So there could still be oxidation. Maybe the authors meant to reference XPS in SI Fig. S11?

• Introduction, technically, the memristor was introduced as an element linking charge and flux, not resistance and charge.

• In the introduction, the authors write that their system is reduced in complexity, but follow up immediately with pointing out rich physicochemical dynamics, this is not quite consistent.

• Already at the end of the introduction, it would be helpful to point out whether the network implementations are experimental or simulations.

• I recommend not putting the SI tables at the end, but include them with the rest of the document in order of appearance in the main text. This will make it easier to follow the SI along with the main manuscript.

• The 3D illustration of the device structure in Fig. 1a is a bit misleading, as it does not represent the actual physical layout. Yet it might be understood to imply that there is some 3D device structure.

• I recommend adding numbered arrows to all IV curves (linear, log, main manuscript, SI) for increased figure legibility.

• The figure caption of SI Fig. S4 is not very explanatory. Some more information would clarify.

• On p. 8, I don't understand why the authors compare their devices with SI Fig. S6, which is measured with larger voltages. SI Fig. S1 already showed comparison devices with Pt top electrodes, why not compare with this instead?

• In SI Fig. S7, shouldn't the decision “Loop > Max_loop” have Y and N exchanged? I.e., end the script when max_loop is exceeded?

In summary, I think the authors have presented a promising manuscript, but in its current state there seem to be two main parts, which are not linked very smoothly. These are the very promising electrical resistive switching data, backed by a very good TEM-based analysis, and separate from this the CV-based analysis, which needs clarification. The network simulations seem too complex to include with the very short provided text.

If the authors can explain more convincingly that indeed, they found a new kind of switching mechanism, and can integrate the two parts better, I think a revised manuscript can be suitable for Nature Communications. If the mechanism, however, is ‘only’ filamentary with low work function electrodes, a different journal would be more suitable. Please note that by ‘only’, I do not mean to diminish the authors' work. I think the measurements and results are very impressive and promising, and

'only' is just meant to differentiate between a possible new mechanism and a different iteration of similar existing mechanisms. The results should definitely be published. But at this point they still need refinement of what the main findings actually are.

Reviewer #2

(Remarks to the Author)

Reviewer #3

(Remarks to the Author)

The paper reports a new structure design of memristors, with low-work-function electrodes to form a two-terminal Ohmic memristor. By mitigating or eradicating the Schottky contact, the new device shows improvement in reliability and stability. Detailed experimental and theoretical analysis of its working principle and dynamics are given. Possible application of the proposed device in ANN training for continuous learning is also demonstrated. The manuscript is well-structured and written. The topic of exploring a new switching mechanism for improving device performance is interesting and important, and worth further exploring for the community. Still, there are some concerns that need to be addressed to improve the manuscript.

First, the paper lacks a more straightforward and quantitative statistical comparison between their new devices and the traditional VCM system (e.g., their fabricated Pt/Ta₂O₅/Ta; Hf/Ta₂O₅/Ta memristors). It is suggested to mention whether Ta is the bottom electrode or top electrode in those conventional VCM for comparison. This comparison should cover aspects such as forming voltage, retention, read precision, cycle-to-cycle stability, device-to-device uniformity, and endurance. While they did mention that the conventional VCM can fail in LRS during the RESET process, it is recommended to provide detailed statistics, such as the number of cycles before failure and the number of cycles/devices tested on Ohmic memristors. It is worth noting that the write endurance of conventional VCM is reported to be at least 10⁶. Without quantitative data and a direct comparison, claiming the new device has better performance is less convincing.

2. Second, it is not very clear what 'unique properties' of the ohmic memristor enable continual learning. The authors mentioned "The binary and analog switching capability in the ohmic memristive device," but conventional VCM can do the same. It is suggested to provide a comparison of network performance from different types of devices to substantiate the claim, regardless of this application or any other application that is better suited for this device.

3. In Figure 1g & h, which illustrates the device's precision and uniformity, it is challenging to compare the device's performance with other VCM devices due to the conductance ranging across several mS. It would be more informative to discuss if their unique device structure enables this mS-level conductance range. For instance, this could be caused by the removal of the Schottky contact or the large cross-size. This raises another concern: are devices with high mS-level conductance favorable in applications? Such high conductance would lead to high current and a more significant IR drop issue in an array. Therefore, it is recommended that the authors address this concern in their discussion.

4. In the "Methods" section, they mention "the active area of the fabricated devices ranges from 4 μm to 2500 μm". But they don't specify the exact lateral size for the device performance data they report earlier.

5. The discussion in at the end of page 4 ("Despite significant progress has been reported and the efforts made to achieve high-performance and reliable electrical properties... making the device operation rather complicated." is confusing. As from my understanding, the devices proposed by the authors also works in "small volumes of charge enrichment/depletion", which will also suffer from related "non-idealities in state retention, endurance, and uniformity". Please clarify the logic here over the main topic of this paper.

6. There seem to be some drawing errors in Supplementary Fig. 4 (device 9), where the bar chart looks narrow while the fitted curve looks much wider.

Version 2:

Reviewer comments:

Reviewer #1

(Remarks to the Author)

Thank you to the authors for their efforts in replying to my comments. I appreciate and respect greatly the work which they invested. They have addressed several of my comments satisfactorily. However, the main point of the manuscript, the explanation of a new mechanism, is still not convincing to me. After clarification in the review comments, based on the provided explanations and experimental evidence, the mechanism looks like ECM filamentary switching to me with low work function electrodes. The authors even write this themselves in the discussion, where they use established ECM/VCM models to analyze their data. (For example l. 392: "The results fit well with the model presented previously for ECM devices", l. 407 they conclude that the calculated values are "comparable to that determined for other ECM systems", or in Fig. 4, where they use the plug/disc model.)

The manuscript can still be suitable for Nature Communications, even if there is not a discovery of a new mechanism.

My high-level recommendations are twofold: 1) To drop the focus on a new mechanism and just describe the mechanism as derived from the experimental support. The achievement of low work function (or combination of ECM with oxygen instead of metal) can still be highlighted! 2) Irrespective of the mechanism being new or not, to make the explanation clearer. Some more detailed points about this:

1) As discussed by the authors in the mechanism discussion sections, and illustrated by Fig. 4, it looks like the mechanism is essentially the tip of the filament near the electrode being oxidized and reduced during switching, which leads to a longer or shorter filament and thus different resistance. This process happens by the movement of oxygen (vacancies) to and from the changing area (the tip of the filament) and this is common understanding for changing filaments in resistive switching. I understand that the authors want to highlight that the resistance is not due to a VCM Schottky mechanism where the height of a barrier changes at the tip of the filament. But the change of the filament still is very similar to an ECM mechanism, just oxygen moves instead of metal. The authors even write themselves that the switching kinetics can be explained with existing ECM models and they use the disc model (side comment, the disc is introduced in the text without explanation at first) which was also developed for ECM/VCM devices. So this looks something like an ECM mechanism where oxygen moves instead of metal ions. It seems to make sense to explain it like that, and can even be highlighted, but it is not a completely new mechanism.

There also remains part of my initial comment about the HRS. In the reviewer replies the authors write that the filament does not dissolve completely, but in the manuscript, Fig. 4, the disc model discusses and illustrates the opposite, that the filament dissolves completely. With the added figure R1.2 (which would be good to show in the SI!), the authors argue that HRS is not controlled by Schottky or Frenkel-Poole, because the plotted areas are not linear vs $E^{1/2}$. But the HRS currents are not linear ohmic either, plus they increase as temperature increases (more than LRS, Suppl. Fig. 19b). So they do not seem to be determined purely by the filament resistance, as the review reply states. So what is the conduction in HRS? (The part between 200 and 400 (V/cm) $^{1/2}$ in R1.2a actually looks more linear than Suppl. Fig. 19b.)

Maybe as the other alternative, the authors mean by new mechanism that this happens with a low work function electrode. This is a very good achievement and should be highlighted! But it does not constitute a new mechanism, as it is still electrochemical as observed before, resulting in a change in length of filament. (The authors point out that the resistance of the remainder of the filament could affect the resistance further, but there does not seem sufficient evidence to conclude this. More in comment 3.)

2) About clarification, the explanation in the author replies to the reviewer comments are much clearer about the mechanism than the manuscript itself. R2.2., for example, is a very clear statement, whereas in the manuscript, the explanation is not as clear and straightforward. For example, on p. 17 it is not clear whether the discussion is about the electroforming step or the switching. And the key aspects of the mechanism are not mentioned very explicitly. I recommend making the manuscript discussion clearer and more explicit on the central points.

With more clarity from the review replies, there are still open questions, especially about the CV part:

3) It is mentioned a few times in the review replies and manuscript that it is not only the length of the filament, but also its resistance along the entire filament. However, there does not seem to be sufficient experimental evidence to support this as a novel finding. It can be concluded from previous publications where the oxygen concentration was measured directly (e.g. <https://onlinelibrary.wiley.com/doi/full/10.1002/adma.201700212>), but the experiments in the present manuscript do not reveal this conclusion directly. With the added understanding from the review replies, I think the CV measurements are meant to provide evidence for the switching mechanisms, but as the authors write, they only reveal information about the interface reactions, and thus only about switching close at the interface with the electrode, but not about the rest of the filament.

4) In my initial assessment, I commented that the CV discussion seemed detached from the rest of the manuscript and its purpose was not clear, so there were no comments to make really. From the review comments, I understand that the CV is meant to support the switching mechanism. This is still not explained very clearly in the main text. (It says “We have studied in more details and comparatively analyzed the electrochemical behavior in both Schottky and Ohmic systems.” – but not why this matters for the manuscript. The CV discussion needs a clear motivation before delving into the details.) With this motivation now clarified, either way, there are some open questions in the CV discussion. (Minor comment in Fig. 2d caption, region II is in panel a.)

a. Why does the current in CV saturate for the Pt devices, but not for the Cu devices? What is the difference and why does it matter for the manuscript? Is it just the different electronic barrier height before the electronic current takes over?

b. Centrally, about the H⁺ and H₂O conclusion, l. 247, which peaks does this refer to in Fig. 2a? The first small peak before saturation in the negative voltage region? It seems like it, because this peak occurs both for Pt and for Cu, and the authors write that the Pt should not undergo any reaction, whereas the Cu should. So the common feature in both measurements would not be due the different reactivity of the metals. If the little peak is indeed H⁺ or H₂O and not metal/electrode reaction, however, this implies that the shoulders in Fig. 2b could also be from the H⁺ or H₂O reaction, and not from metal/electrode reaction, and then there is no support for the electrode oxidation conclusion from CV. (Either way, the H⁺ or H₂O mention needs a reference.)

c. The authors write that the current decrease in subsequent CV cycles for Pt and Cu is due to oxidation of the opposing Ta electrode. But there is a control missing for this conclusion with an inert opposing electrode. Could the authors make a device with Pt electrodes on either side? If the decrease in CV current is from oxidation of opposite electrode, then with Pt on both sides, the current should not decrease.

d. The authors write that the eq. 3-6 agree well with the experimental characteristics, but do not show it, like a fit to the data.

Is this instead meant simply as qualitative statement that in different voltage regimes, the electronic current dominates and in others, where the electronic current is suppressed, the ionic contribution becomes visible? I think it should be stated like this if this is the case. Also, it is not really true for eq. 6, as in Fig. 2b, the electronic current is very pronounced both in forward and reverse, but in eq. 6, the electronic contribution is constant and small. This goes back to my initial comment about the electronic current in a reverse Schottky barrier.

e. The discussion after these equations, is it hypothesis as to what the effect might be? There is no experimental evidence that any of the Faradaic reactions indeed are more efficient or less efficient in the different situations. Or is this just a description of what is visible in Fig. 2? If yes, this would go with comment d that the equations give a qualitative description rather than direct confirmation. It should be stated clearly whether something is hypothesis or description of observation.

f. Finally, in terms of contributing to the mechanism explanation, the CV explores effects at the interface to electrodes, but does not provide support for the changing resistance of the remainder of the filament. As mentioned earlier, the experimental support is only there for interface effects.

g. Another note on clarification, p. 17, l. 344, "Switching at positive voltages (to Ta bottom electrode)" – this is confusing. At the start of the manuscript, it was written that bias was applied to the top electrode. There was no mention of changing this during the CV discussion, but this is very important, as the CV measurements were carried out with very asymmetric devices (Pt and Cu at top), so where was the bias applied? The schematics indicate bias application still at the top, but in l. 344 now suddenly it says at the bottom

5) Finally a side note, Q4 from the previous comments maybe my initial comment was not clear. It was not my intention to compare the authors' work with this reference. Just the sentence "The robust device characteristics suggest that the occurrence of resistive switching in the ohmic memristors does not rely on Schottky barrier height modulation." On p. 11, l. 220, is not correct. Just because the device characteristics are robust, it does not mean that they do not rely on Schottky height barrier modulation. It is not a logically valid conclusion.

Apologies for the load of text. I do not mean to drag out the review process. But I want to engage in a proper scientific discussion which the manuscript deserves. With the recommended clarifications, I believe the manuscript can still be suitable for Nature Communications. It does not need a 'revolutionary thing' like a new mechanism. The demonstration of excellent resistive switching with low work function electrodes is a strong enough finding! But it is very important to present the data and conclusions accurately and clearly.

A few minor comments:

6) For the sputtering processes, the gas pressure, gas flow, substrate temperature, substrate bias, and target power are missing.

7) As the simulations are a central part of the manuscript, I believe there should be a Code Availability statement. Ideally, the code would be available in a repository.

8) The difference between Fig. 3 f and h is not pointed out in the caption.

9) I tried to understand better the network simulations. The key things that made it difficult are:

a. The structure of the simulated network is not explained very clearly, the information is scattered in different methods and figures parts. I think a clearer dedicated description of the network architecture will help, for example by expanding on Suppl. Fig. 28 and 29 and writing clearly in the main text that the structure is explained there.

b. It would be helpful to have the propagation and update equations in the supplementary information. For example, the text says in l. 510: "in which the hidden parameters can decide the degree of the synaptic weights' change." That sounds like the hidden value is a weight parameter in some equation. But what is the equation?

c. L517, "the weights between two neurons are hard to keep 'memory' for previous training experience", there seems to be something wrong with this sentence.

d. Different effects of noise and failure rate are discussed, but without assessment, how 'good' it is. Given that the experimental resistance values were mapped to the network, is there something like a digital baseline, with which to compare? Or how do the simulations compare with other networks based on experimental values?

Reviewer #2

(Remarks to the Author)

The authors have provided adequate explanations regarding the issues I previously raised. My concerns about the interpretation of the mechanism proposed by the authors, compared to the conventional filament-type switching mechanism, have not been fully resolved. However, there are no logical inconsistencies, and the manuscript offers readers a meaningful and interesting perspective. Therefore, I recommend publishing the current version of the manuscript in Nature Communications.

Reviewer #3

(Remarks to the Author)

The authors have addressed most of my comments. Regarding the 'unique properties' for continual learning, the authors rightly pointed out that the application demands low programming error. While there have been other efforts to reduce programming errors through testing methods and algorithms, the material effort is orthogonal. The authors also provided endurance performance comparisons with other device technologies, but it would be even better if the authors could discuss whether the current endurance performance can support continual learning applications.

Version 3:

Reviewer comments:

Reviewer #1

(Remarks to the Author)

Thank you to the authors for their continued effort and patience to improve the manuscript. The new changes and explanations are very good and especially the naming of their observation and comparison with ECM and VCM is an elegant solution. With this comparison and the additional explanations, the experimental observations can be explained consistently.

Apologies for the things I had overlooked in the previous version - this is a very comprehensive work! – or where my formulations weren't clear (e.g. between Q3 and Q6 – I meant complete rupture of the tip of the filament, not complete dissolution of the complete filament, obviously not clear enough wording).

My main concerns have been addressed and the integration of the CV experiments is much clearer now, as well as the explicit explanation of the observed switching. As part of this, the linear IV in HRS in SI Fig. 19 looks convincing. For easier comparison, I suggest adding the measured voltage range in the figure caption for SI Fig. 19c, so it is clear that the displayed voltage range is comparable to a&b.

There is one inconsistency left, which I think should be addressed.

Q&R16 there remains the inconsistency between the equation (6) and the measured CV curve in Fig. 2b plus the explanation beginning in l. 331. Fig. 2b and the explanation clearly state that there is similar dominant exponential electronic current in forward and reverse bias, but eq. (6) does not include this electronic contribution. It only includes the constant contribution ($-A \cdot T^2$) if the argument of the Schottky exponential function is zero. The $-A \cdot T^2$ term provides a constant contribution, which does not reflect the measurement and explanation of Fig. 2b.

With some further thought, I think that for the OE/oxide/OE in Fig. 2b, the starting point of adding eq. (1) and (2) is not appropriate anymore in the same way as for the case of high Schottky contact. It should be ok as a simplification for the high Schottky contact, where the electronic current is mostly blocked in the reverse bias direction, and the forward direction is then controlled by the forward-biased Schottky barrier, if the counter-electrode is ohmic.

But for the OE/oxide/OE, the main point of the manuscript is that the electronic resistance is not controlled by a Schottky barrier, isn't it? So then in the expression $j = j_{TE} + j_{ION}$, for the OE/oxide/OE case, the j_{TE} should not be a Schottky expression, right? Otherwise this would be inconsistent with the mechanism explanation. This then also extends to eq. (5): Here, while the exponential term from the Schottky expression qualitatively describes the current, the Schottky expression does not seem appropriate physically, as the explanation of the mechanism is that the current is not controlled by a Schottky barrier.

Since the injection from the OE into the oxide is near-ohmic, maybe an expression in the space-charge-limited conduction framework can provide an appropriate expression for the electronic contribution. I know SCLC is often misused in resistive switching work, but in this case, some of the boundary conditions should actually be fulfilled. See e.g. <https://journals.aps.org/prapplied/abstract/10.1103/PhysRevApplied.9.044017> for an example of a similar band structure configuration as for the present manuscript. (I know it's for a completely different materials system, but the theoretical treatment might be transferable.) Of course, the explanation doesn't have to be based on SCLC, it is just an idea for a possible starting point.

For the explanation of the measured added ionic and electronic current, in all contact cases, please can the authors also add a very brief explanation, why the ionic contribution appears as peaks, i.e. reduces after this peak? From the provided expression for the ionic contribution, I think there should not be a peak, but a constant increase with voltage? Are some of the parameters in eq. (2) possibly voltage-dependent? Just one sentence would clarify this.

On the same discussion, while the authors disagreed to my comment Q17, they also confirmed that the provided equations are qualitative descriptions of the observed measurements, rather than exact analytic expressions which. While a small detail, and while qualitative explanations are often appropriate, for scientific accuracy, I think it is important to distinguish between qualitative description and quantitative/analytical analysis. (This is obviously a minor comment.)

Finally, I recommend special care in proof-reading the hopefully final version of the manuscript, as there persist some small grammar mistakes here and there. Again, this is obviously a minor comment.

Reviewer #3

(Remarks to the Author)

The authors have addressed my comments, and I agree that this is ready for publication in Nature Communications.

Version 4:

Reviewer comments:

Reviewer #1

(Remarks to the Author)

Thanks again to the authors for their efforts to improve the manuscript. The smaller comments have been answered sufficiently. However, I am sorry to say that my main comments Q3-5 have not been answered. The authors provided a discussion about ion exchange, but this does not address my comments.

In a CV measurement, an electronic signal is measured. Just from the measured signal, it is not possible to say whether this is an ionic or electronic response. (Frequency- or temperature-dependent measurements could possibly distinguish.) However, in the absence of an ion reservoir (like liquid metal electrodes in the early days of ionic measurements), and thus at least partially ion-blocking electrodes, and at the same time low electronic barriers, as in the given case of near-ohmic electrodes, the electronic contribution is typically much larger than the ionic contribution. Without proof to the opposite, it thus stands to assume that the exponential current increase in Fig. 2f (and similar figures) between -3 and -5 V and between 2 and 4 V is electronic and not ionic. So the explanation that this exponential current increase is due to ionic exchange is not convincing and not supported by evidence. The author reply "Equation 6 says that the electronic current due to Schottky emission at the interface is low and constant and that the ionic reaction current is supposed to increase exponentially." is not proof for this either, as it is only an interpretation of the results. In fact, this is exactly what my main comments Q3-6 are about, namely the concern that this interpretation is not correct and the authors have not convincingly argued otherwise, as detailed just above. In fact, in R7 the authors write that the reason why the ionic contribution causes a peak, which decreases afterwards, is that "at certain moment (depends on the sweep rate), the concentration of reactants depletes, as the reaction rate is fast, and the supply of new reactants (ions) is too slow. Of this reason, the current decreases, thus, forming a peak. Both the concentration depletion at the interface and slow transport of reactants leads to the formation of current density peaks." This supports further the interpretation that the exponential current increase after the peaks is not ionic anymore. In the previous manuscript version, the authors had written that there is similar exponential electronic current increase in the forward and negative bias, but in the new version, this was just replaced by writing that there are similar ionic contributions. So the authors just changed their interpretation from before. To clarify my point, I do not assume that the electronic contribution would arise from changing dopant concentrations due to changing ion concentrations. It simply arises because the low, near-ohmic, electronic barriers between the oxide and the electrodes make it very easy to inject electrons from the electrodes into the oxide. This is not a doping effect, but a carrier injection effect. (Hence my suggestion in Q7 that maybe SCLC could play a role here.)

In brief, the exponential current increase in Fig. 2f and similar figures looks to be electronic, not ionic, and equations 5 & 6 are not adequate to describe this, as it ascribes the exponential increase to an ionic contribution and does not include the exponential electronic contribution. My previous comments Q3-5 provide further details.

To my understanding, the authors either need to provide proof that the exponential increase is indeed ionic, or they need to adapt the equations for the interpretation of the results.

As a side comment arising from this discussion, the CV measurements are actually not always explained clearly. What were the sweep rates in the figures? The experimental part provides a large range of sweep rates. But which ones were used in the figures? And were different sweep rates compared to investigate reactivity?

Version 5:

Reviewer comments:

Reviewer #1

(Remarks to the Author)

Thanks to the authors for these final corrections. I am glad we agree on the science of the discussed measurements and equations and think it inspires confidence in the results. The updated discussion reflects the complexity of the system well and I think it will be instructive to a wide audience. Thanks to the authors for their patience and the engaging review discussions! I think the article can be published in Nature Communications now and will have good impact. Congratulations on these demonstrations and good luck for all future work!

Response letter to the reviewers

We would like to thank the reviewers for their time reading our manuscript. We sincerely appreciate the positive assessment and constructive comments. Based on reviewers' comments, we have carefully revised our manuscript and supplementary information by providing more experimental and simulation details, adding new results with electrical measurements, adding new simulation data, amending/modifying some claims. All changes are highlighted in yellow color in the main text and supplementary information.

The main changes we have made are:

- 1: Provide more comprehensive comparisons between ohmic memristors with conventional VCM
- 2: Add new data on device endurance and device-to-device variability
- 3: Add comparison on neural networks performance
- 4: Provide more details on neural networks simulation
- 5: Add new data showing retention characteristics of conductance levels at μS range
- 6: Justify the switching mechanism

Please, find enclosed the point-by-point responses to the referees' comments. We hope that the reviewers will find the responses and the corresponding revisions satisfactory. If any additional information is required, we will be more than happy to provide it as soon as possible.

Kind regards,

Shaochuan Chen, Yuchao Yang and Ilia Valov

Point-by-point response to the reviewers' comments

Reviewer #1:

Q1: In this manuscript, the authors present resistive switching devices with two ohmic contacts and claim that they demonstrate a new switching mechanism. Indeed, the majority of resistive switching devices use at least one non-ohmic electrode and there are reports where two ohmic electrodes do not lead to resistive switching. The electrical resistive switching data look very promising, the manuscript is mostly well-written, and the breadth of investigations is commendable. Yet I have a hard time placing the manuscript on the map and I am not convinced of the switching mechanism.

R1: We thank the reviewer for reading and commenting on our manuscript. According to his/her comments we have performed new experiments, analyses and improved the discussion. We have tried our best to explain more clearly the new mechanism and the inherent relation between the different parts of the manuscript.

As first argument, we would point out that up to now in the literature all types of VCM devices are reported to operate essentially and solely due to the presence of a Schottky barrier/electrode. *In all these papers, the resistive switching (and ON and OFF states) has been attributed solely to the modulation of the Schottky barrier* at the interface between the Schottky electrode and the filament tip due shift of oxygen vacancies and electrons. In same time, all ECM devices are proposed to work entirely on the formation and *complete* dissolution of (purely) metallic filaments. In our devices, we report a mechanism, that is neither VCM, nor ECM. It is based on electrochemical modulation of a metal-ion rich filament (reduced oxide) leading to oxidation/passivation of parts of this filament and decrease of its thickness, where in same time the filament is not completely dissolved. The mechanism we propose in the manuscript is new and not reported previously. We also highlight the great importance of the interplay between Schottky and redox barriers at the interface of memristors and discuss in detail this impact. Our findings represent an important step ahead in understanding and controlling the functionalities of memristive devices in general.

In this manuscript, we experimentally verify and discuss that *Schottky barrier is not necessary/essential to have resistive switching in the VCM systems*. The switching mechanism is based on electrochemical passivation, and it is neither VCM, nor ECM mechanism.

We experimentally verified this new mechanism by electrical measurements and chemical analysis. In our revised manuscript we'll do our best to explain better our results.

Q2.1: Firstly, the presented IV curves look very similar to, e.g., Fig. 2.3 in <https://www.sciencedirect.com/science/article/pii/B9780081027820000022>. In both cases, the LRS is pretty much linear, which can be explained simply by the conductive filament in this state. The authors' claim to this configuration in the LRS seems sound and the TEM supports the metallic filament explanation.

R2.1: We thank the reviewer for the comment. In our devices we have observed only *Ta-rich filament* (no metallic phase/filament were detected) as the TEM images and EDS analysis show. So, we do not have observed (or claimed) metallic filament(s). However, after careful checking the main manuscript,

we realized that Fig. 4c may lead to confusion that the filament is metallic (in the sense of the comment of the reviewer). The figure has been modified in the revised version of the manuscript.

In addition, we would like to respectfully disagree that the figure cited by the reviewer (<https://www.sciencedirect.com/science/article/pii/B978008102782000022>), is very similar to our I - V curves (Fig. 1b). In our plot, the RESET is more abrupt and different than the RESET shown in Fig. 2.3 of the article.

Q2.2: However, the HRS is not analysed further. If the filament is ruptured in the HRS, what would the authors call the ‘blocking’ part in this state? It is probably a part of oxide or interface without the metallic filament. Whether at the interface or slightly apart from it, there will be an energy barrier for electrons between the filament and the filament-free oxide part. The junction between a metallic part (filament) and an insulating part is effectively something like a Schottky barrier, so I don’t really understand the authors’ claim to a Schottky-barrier-free new mechanism.

R2.2: We understand the concerns of the reviewer. We’ll do our best to explain in more detail the new mechanism.

The new mechanism we propose is based on an electrochemical passivation/oxidation of the filament. The filament is composed of Ta-rich TaO_x and the change of the resistance state is not due to the change of the Schottky barrier height at the interface with the oxide or somewhere else, but because of the increasing the resistance of the filament itself. This resistance increase is caused by 2 factors:

1. Increasing of the oxidation state of the metal ions.
2. Decreasing the length of the filament.

These 2 factors can (but not necessarily) occur at same time.

When we apply a positive voltage to the filament we oxidize/passivate the TaO_x filament. This means, Ta-ions in the filament of lower oxidation state (e.g. 4+ and/or 3+ and/or 2+ and/or 1+) are oxidized to higher oxidation state e.g. Ta^{5+} , accompanied by additional incorporation of oxygen ions and the oxide transits to its stoichiometric form. This oxidation will affect not only the first layers of the filament tip, but also expand deeper, reducing the effective length of the (TaO_x) filament. During this process, the filament is not completely dissolved (transformed to Ta_2O_5), as evidenced from the experimental results (TEM, electrical measurements showing HRS is independent on electrode area). This oxidation behavior is typical for electrochemical passivation of metals and conductive oxides.

To our best knowledge this mechanism has not been proposed by any study before and is fundamentally different than the wide-spread explanation of the VCM mechanism, explaining the switching by modulation of the Schottky barrier. It is also very different of the ECM mechanism, where the atoms of the filament after oxidation diffuse away and the filament is completely dissolved.

At every metal/oxide interface there is some Schottky barrier (even if very low). In the conventional VCM devices e.g. Ta/ Ta_2O_5 /Pt (or Ti/ TiO_2 /Pt; Hf/ HfO_2 /Pt etc.) the interface Me/MeO is considered/denoted as Ohmic one and the electrode Me is called Ohmic electrode. This is not because this interface is absolutely barrier-free!

In our devices the resistance of the HRS state *is not* determined by the Schottky barrier, but by the filament resistance.

Q2.3: The SI itself indicates there is an expected band offset of about 1 eV between the ohmic electrodes and the tantalum oxide film. One simple way to analyse the HRS further is fitting an electronic conduction model to it, possibly Schottky emission ... In broader terms, I am not convinced whether there is a new mechanism at play here, or whether it is ‘just’ a filamentary switching mechanism with low work function electrodes. The latter is surely a great achievement in itself, but would significantly alter the claim of the manuscript.

R2.3: We understand the reviewer’s point. In this study, only when high-work-function metals such as Pt and Cu are in contact with Ta₂O₅ layers Schottky barriers are formed, thus dominating the performance. However, in the case of ohmic memristive devices, the energy barriers at metal-oxide interfaces are significantly minimized not only due to the lower work function of ohmic electrodes, but also their electrochemical properties. Ohmic electrodes such as Ta, Hf, Zr, have lower work functions compared to Pt and Cu, thus they form much lower intrinsic energy barrier (electrons) with Ta₂O₅. More importantly, ohmic electrodes have very high oxygen affinity hence can easily scavenge oxygen ions from resistive switching oxide, generating nanoscale size, substoichiometric, conductive oxide at the interface; in the meanwhile, the metal oxides closer to ohmic electrodes get partially reduced (more conductive) due to the loss of oxygen ions. Thus, for VCM devices, the ohmic electrodes always form ohmic-like contact with the resistive switching oxides, in both high resistive state and high resistive state (Advances in Physics 2021, 70, 155). Please, keep in mind that for Ta/Ta₂O₅ interface one also calculates about 1 eV Schottky barrier (according to the literature values for the work functions of Ta and Ta₂O₅). However, because the effects described above this is considered as ohmic interface. Similar discussion applies to other Me/MeO interfaces.

Furthermore, we observe clearly rectifying characteristics for classical Ta/Ta₂O₅/Pt device (see Figure R1.1a), which originates from the existence of Schottky barrier. In contrast, for ohmic memristive systems, we didn’t observe any rectifying effects. All pristine ohmic memristive devices show nearly symmetric *I-V* curves, with high currents under forward and reverse biased conditions, as shown in Figure R1.1b from 30-nm Ta₂O₅ device and Figure R1.1c 8-nm Ta₂O₅ device. These results exclude the possible existence of Schottky barrier as resistance determining factor.

To comply with the reviewer’s comment, we have analysed in more detail the HRS. As it can be seen in Figure R1.2a,b the currents at HRS cannot fit to thermionic emission and Frenkel-Poole conduction processes, and the currents at LRS fit ohmic conduction. These observations further confirm our conclusion that Schottky effect is not involved in resistive switching mechanism.

Figure R1.1: I - V characteristics of devices with and without Schottky barriers. The data were collected from (a) pristine 30-nm thick Ta₂O₅ device with Schottky barrier, (b) pristine 30-nm thick Ta/Ta₂O₅/Hf ohmic device and (c) pristine 8-nm thick Ta₂O₅ devices. The smaller figure in each panel shows currents in linear scale.

Figure R1.2: fitting of HRS current densities with thermionic emission conduction model (a), and Frenkel-Poole conduction model. The nonlinear dependence observed at panels a and b implies the HRS conduction process is not related to Schottky effect.

Q3: Related to this, have the authors checked whether their devices can be operated simply as unipolar devices? In addition to the comment above, the two halves of the IV curve look very much like unipolar switching curves, cp., e.g., Fig. 4 in

<https://www.tandfonline.com/doi/full/10.1080/00018732.2022.2084006>

It would be helpful to have this in the SI, as it would support the later mechanism discussion if the devices do not work as unipolar.

R3: To comply with the reviewer's comment, we have checked and confirmed that the resistive switching in ohmic memristive devices does not exhibit unipolar switching characteristics. For unipolar switching, as shown in Figure R1.3a (Advances in Physics 2021, 70, 155), SET and RESET events occur in the same voltage polarity (can be positive or negative), RESET currents are higher than SET currents, the RESET process is very abrupt, and the RESET voltage is smaller than SET voltage, typically because of Joule heating effect.

In addition, we have performed experiments to confirm the bipolar operation of the ohmic memristive device, as shown in Figure R1.3b. The device does not show unipolar switching characteristics, i.e. the device cannot be RESET to HRS under the application of same voltage polarity (arrow 2) as SET operation (arrow 1), only a reverse voltage (arrow 3) was needed to switch the device to HRS (bipolar resistive switching). The SET voltage is lower than the RESET voltage and the RESET current is not higher than the SET current.

Figure R1.3: (a) Schematic of I - V diagram of unipolar resistive switching. This figure was reproduced from Figure 4 of reference: *Advances in Physics* 2021, 70, 155. (b) I - V characteristics of bipolar ohmic memristive device.

Q4: Also related, on p. 10, “The robust device characteristics suggest that the occurrence of resistive switching in the ohmic memristors does not rely on Schottky barrier height modulation.” This is not a self-evident conclusion. Very robust resistive switching has been attributed to Schottky barrier control, e.g., recently <https://www.science.org/doi/full/10.1126/sciadv.adg1946>

R4: We thank the reviewer for this constructive comment. As requested by the reviewer, we have carefully read the provided reference and found that electrical performance of our ohmic memristive devices are superior to the cited interface-type resistive switching devices in endurance, state retention and device-to-device variation (see Figure R1.4). It can be seen that the interface-type switching devices (top panels) show higher cycle-to-cycle and device-to-device variation, lower endurance and distinct drift of conductance levels under constant read voltage.

Figure R1.4: Comparison of I - V curves, endurance, state retention, high temperature stability of ohmic memristive devices with interface-type resistive switching devices. The data of interface-type resistive switching devices (shown in top panels) are from Sci. Adv. 9, eadg1946 (2023).

In our manuscript we aimed to describe that the resistive switching in our ohmic memristive devices do not rely on Schottky-barrier modulation (see our response in R2) and explain the advantages following from this new mechanism. For interface-type switching the resistance change is based on modulation of charge carrier density and hence barrier height at metal/oxide interface, in this case Schottky-like energy barrier is necessary.

In principle, we have to say that it is difficult to directly compare the performance of memristive devices prepared in different groups and tested under different conditions. There are many factors such as electrode and electrolyte sizes and thicknesses, capping layer materials and thicknesses, surrounding atmosphere, pulse length and heights, pulse schemes, as well many others, that vary from group to group. Even the stoichiometry, density and as well impurity levels in the switching oxide vary strongly depending on deposition and operation conditions (also from research group to research group), leading to difference switching performance of apparently “same” oxide. Because of this reason we have focused on the comparison of the performance of samples, produced and tested by us in the same way and same conditions, using same materials, of same purity etc. Nevertheless, the unavoidable comparison with device from the literature show that our ohmic devices perform better or at least comparable (considering all the characterization aspects and criteria for evaluation).

Q5: If the devices are filamentary switching with low work function electrodes rather than a truly different switching mechanism, the authors should present more data on device to device variability than SI Fig. S4. The main challenge of filamentary switching is reproducibility, so, reported improvement on such devices should demonstrate how they are better than existing filamentary devices.

R5: As responded in R1 and R2.1 we have experimentally shown a new switching mechanism in which Schottky barrier is not needed to modulate the resistance of the memristive devices. It is based on filament composition changes (oxidation/reduction) and in that sense it is a filamentary switching, but the mechanism is entirely new. The mechanism has not been proposed or evidenced in the literature. Thus, we are convinced we report on entirely new mechanism.

Ohmic devices have several important advantages. The mitigation of excessive voltage drop at Schottky interface has made ohmic memristive devices:

- exhibit lower forming voltages (Figure R1.5a,b);
- eliminate forming current overshoot (Figure R1.5a);
- overcome RESET failure compared to conventional VCM (Supplementary Fig. 8);
- exhibit higher HRS and LRS resistance ratio (Figure R1.5c,d and Figure R1.6-1.8);
- high endurance (Figure R1.8).

The ohmic memristive device has HRS resistance ~ 300 k Ω and HRS and LRS resistance ratio higher than 100 ($V_{\text{read}} = -0.2$ V), the endurance was measured to be higher than one million cycles (Figure R1.6e,f).

As requested by the reviewer, we have additionally measured more devices, and all the results show the high HRS and LRS resistance ratio, good device-to-device uniformity, as shown in Figure R1.7. Figure

R1.8 depicts direct comparison of endurance between ohmic memristive device and conventional VCM devices. Despite the endurance of conventional VCM has been reported higher than 10^6 , the HRS and LRS resistance (R_{HRS}/R_{LRS}) is typically lower than 10 and the overall device currents are high (high I_{LRS} and I_{HRS}), which results in high power consumption. In contrast, ohmic memristive devices show high endurance with high R_{HRS}/R_{LRS} and lower I_{HRS} (higher R_{HRS}).

Figure R1.5: Comparison of forming. (a) Exemplary forming and (c) resistive switching (20 cycles) characteristics of Pt/Ta₂O₅/Ta device. (b) Exemplary forming and (d) resistive switching (20 cycles) of Hf/Ta₂O₅/Ta device. The inset plots in panels c and d show corresponding resistance change as a function of applied voltages.

Figure R1.6: HRS and LRS resistance ratio (R_{HRS}/R_{LRS}) comparison between conventional VCM devices and ohmic memristive devices.

Figure R1.7: Additional data showing high HRS and LRS resistance ratio and low device-to-device variability of ohmic memristive devices.

Figure R1.8: Comparison of endurance, HRS and LRS resistance ratio, overall device current levels with reported VCM devices. Ohmic memristive devices show higher HRS/LRS resistance ratio with good endurance cycle. Moreover, the overall current levels in ohmic memristive devices are lower than other devices, indicating lower power consumption in ohmic memristive devices.

Based on above results, the ohmic memristive devices exhibit more controllable forming processes (no current overshoot, low forming voltages), higher HRS and LRS resistance ratio (>100x), low device-to-device variation, lower R_{HRS} (lower power consumption), and good endurance (> 10^6). These performances can hardly be achieved simultaneously in conventional VCM devices (Figure R1.8).

In the revised version of Supplementary information, we have added the forming IV curve (Figure R1.5a), device-to-device variation and endurance characteristics (Figure R1.6a-c) of conventional VCM in Supplementary Fig. 1. New data about device variation in ohmic memristive device (Figure R1.7) has been added as Supplementary Fig. 7. Figure R1.8 has been added as Supplementary Fig. 9.

Q6: Also related to the mechanism, the analysis and discussion around Fig. 2 and SI Fig. S12 and S13 do not convince me of the ‘new’ mechanism either, and they do not really fit neatly with the electrical data. It seems like there is the electrical resistive switching data, supported by the TEM analysis, and as a second part there are the CV studies, but the connection is missing.

R6: Our manuscript contains three main parts, that are inherently related. The first part is the realization for first time of reliable ohmic memristive devices and measuring their performance such as $I-V$ and

pulse characteristics, endurance and retention. The second part is focused on revealing the mechanism and this part includes electrical measurements, combined with XPS and TEM studies and a thorough discussion (and measurements) on the interface dynamics and the interplay between Schottky and redox barriers and their importance for the memristors in general. *In our view this is a very novel point and significant contribution to the fundamental understanding of memristive devices.* It provides the conditions and explains why our memristive devices are properly working and why their characteristics are advantageous. It also provides guidelines for the behaviour that have to be expected, depending to the ration between Schottky and redox barriers. Understanding the fundamental discussion shown in Fig. 2 and Fig. S12 and S13 (now Supplementary Fig. 16 and 17) is essential to understand the device functionalities and properties and the advantages compared to other devices.

The measured electrical characteristics presented in Fig. S12 and S13 (now Supplementary Fig. 16 and 17) support the discussion and the schema, provided in Fig. 2, showing the rectification effect for high Schottky barrier (classical VCM) and no rectification for ohmic devices. Please, refer also to Fig. R1.1 (response **R2.3**). From the CVs it is evident that electrochemical peaks are always present, demonstrating that high Schottky barriers cannot suppress electrochemical processes, just in opposite, make the ionic currents comparable on magnitude to electronic currents. This leads to change in the performance of the device.

We also would like to explain in more detail the difference between CVs and I-Vs. Cyclic voltammograms are electrochemical measurements, providing information about the redox behaviour of the system. CVs are performed in a voltage window where no switching takes place. The information we receive from CVs is much different (but complimentary) to the data received form the I-V curves where the resistive switching is in focus. In I-V sweeps the change of the current is generally because of the change of the electronic resistance of the system (e.g due to short circuit due to filament formation). Faraday currents are in the general case much lower and in most cased not detectable in I-Vs.

CV measurements show what happens (electrochemically) before the switching event. Both measurements are strongly complementary.

Thus, we show what happens before the switching event (the processes preceding and initiating the filament formation), during switching (SET/RESET and the mechanism) and after it (stability of states etc.).

TEM measurements complement electrical experiments by showing different types of filaments, depending on the experimental conditions/operation and helps with the interpretation of the electrical measurements. XPS is also contributing for building a complete picture, by verifying the chemical states of different films within the stack.

In that sense fig.2 is showing and discussing why we have new switching mechanism and how/why it happens. Supplementary Fig. 16 and 17 are supporting this discussion.

We hope we succeed better to explain the inherent relation of the different results shown in the manuscript.

Q7: Partly, the authors may be able to remedy this by clarifying their explanations. Partly, there are more fundamental concerns around Fig. 2a and SI Fig. S12:

The insets of the CV in Fig. 2a and SI Fig. S12 are confusing. They cannot be just zoom-ins of the main panels, as there are features visible or different, which are missing in the main figures. For example, in Fig. 2a, in the main panel, the current is just zero at all negative voltages, but in the inset it is clearly different. Are they maybe forward and backward sweeps instead of zoom-ins? Please clarify in the text and figure captions. (Also, rather “zoom-out”, as the voltage axis is larger in the insets?)

R7: We are sorry for the confusion with the inset figures. The inset CV figures are shown with different current resolution (compared to main one) to make visible redox peaks during the negative sweep (the current densities are indicated on the y-axes). To record the redox peaks the sensitivity must be increased as the redox currents are much lower than the electronic currents. There is always a trade-off between the current resolution and maximum current that the Source Meter can measure. To observe overall picture of current densities we set the source meter with a high current range, thus can detect higher currents, as the case of Fig. 2a and Supplementary Fig. 12 (Supplementary Fig. 16 in the revised version). To measure ionic currents, we conducted measurements with lower current range (high sensitivity) to detect the redox current density peaks, as shown as inset in the figures.

In the revised version of manuscript, we have clarified this by replacing “zoom-in” with “high-resolution CV sweeps”

Q8 Please also explain the difference between IV measurements and cyclic voltammetry.

R8: We thank the reviewer for the comment. *I-V* sweeps are used to study the switching event. They are performed in a voltage window, where resistive switching takes place, and the current increase/decrease is predominantly caused by change of the electronic conductivity (filament formation/rupture). Our *I-V* sweep measurements were conducted using Keithley 2636A Source Meter with a sweep rate around 1 V/s. The current resolution ranges from 1 nA to 5 mA.

Cyclic voltammetry is used to study the electrochemical processes, preceding (until begin of) filament formation. The measurements are performed within more narrow voltage window, where the filament formation has still not begun (or the very initial stage of it) and with higher current resolution. For the cyclic voltammetry, the measurements were conducted using Keithley 6430 Sub-Femtoamp Remote SourceMeter, which allows detecting currents down to fA range with sweep rate of up to 130 mV/s.

Both experiments are complementary to complete the physical picture of redox-based resistive switching.

Q9 It is not entirely clear, which currents the authors ascribe to the electronic and which to the ionic processes. Is the ionic current only the little bump? What about the strong current increase after the little bump in the first cycles? Surely these are not ascribed to ionic processes, but electronic conduction?

R9: CV and *I-V* measurements in solid state systems are typically showing mixed ionic-electronic currents. This means that the current signal is the sum of the electronic + ionic currents. Moreover, ionic processes may lead to an increase of the electronic currents (e.g. due to the electrochemical donor or acceptor incorporation). The peaks (or as the reviewer denoted it - bumps) are indicative for electrochemical redox reactions. The exact ionic transference number cannot be determined, but to be able to see the peak means that the ionic current is of same magnitude. The current increase after the bump can be either following redox reaction (e.g. from moisture) and/or increase of the electronic current (due to donor/acceptor incorporation effect, including leakage currents etc.). We would also like to

mention that the absolute currents shown in the inset of Fig. 2a, are as low as 0.37 nA (bump) and 0.9 nA at the end of the sweep (for the violet curve).

Q9a: Firstly, the currents are much too high for ionic currents.

R9a: We respectfully disagree with the statement that the current in the inset are much too high. *The absolute current value on the max of the peak is 0.37 nA.* The current densities are also comparable to what has been previously reported for other systems e.g. Cu/SiO₂, Cu/Ta₂O₅ etc. and in the literature (Adv. Funct. Mater., 25, 6374, 2015; Nanoscale. 8(29):13828, 2016). This is not saying that the current is entirely Faradaic, but that it contains a redox current that can modify the electronic current.

The reviewer should also consider the voltage scale.

Q9b: Secondly, the strong increase in the reverse direction is also visible in the Pt devices. With eq. 1, however, the electronic current in the reverse direction would be zero. This is in line with the comment that eq. 1 is not appropriate for the reverse direction.

R9b: The device we used has the structure Pt/Ta₂O₅/Ta electrode. We have already shown in previous studies, that Ta is electrochemically active and can be dissolved and participate in the switching process, of course, not as actively as Cu electrode but this is also reflected in the currents, recorded in CVs. Moreover, oxygen (and moisture) reactions are also proceeding in parallel. These electrochemical phenomena are well-documented in thin oxide films. Please, consider as well that alone due to redox reactions the electronic conductivity of switching oxide significantly increases due to donor incorporation effects (see *Adv. Electron. Mater.* 4, 1700458, 2017). In our view, Eq 1. is suitable for representing the thermionic emission current in these systems, of course it is not accounting for changes due to the electrochemical incorporation of donor or acceptors.

This underlines the vital importance of considering the Schottky and redox barrier together!

Q10 Eq. 1 is for a forward-biased metal-semiconductor diode. The authors already cite the book by Sze, which discusses this in detail in ch. 3. While for a forward-biased metal-insulator junction as in the presented devices, this is appropriate, it is not really for the reverse direction, which is also discussed in the book by Sze in ch. 4. (Essentially, charge injection in a reverse-biased Schottky contact will lead to image force lowering of the barrier and result in a dependence J proportional to $\exp(\sqrt{V})$). The rapid current increase in the reverse direction would support this.) With this, many of the authors' assumptions in the derivations of the expressions around Fig. 2 and SI Fig. S12 are not valid anymore.

R10: We thank the reviewer for the comment. Indeed, the effective barrier height would be lowered due to image-force barrier lowering, and it could lead to a voltage dependence of the reverse current in the case of high voltages. In Figure 2a and Supplementary Fig. 12 (Supplementary Fig. 16 in the revised version), it is evident that the reverse currents ($V: 0 \rightarrow -7V \rightarrow 0V$) are governed by the saturation currents of Schottky-contact and also ionic currents (oxidation of Ta electrode, and possible reactions of oxygen and/or moisture). Only when the reverse bias exceeds $\sim -7V$, the Schottky effect was observed. Note that in memristive systems the operation voltages are much lower (see Figure 1d in the main text) as the resistive switching oxide is much thinner (8 nm in our case). Moreover, operating the devices using such high voltage ($\sim -7V$) should always be avoided to prevent irreversible breakdown of resistive switching layer, which is typically below 10-nm thick. For this reason, we didn't consider image-force barrier

lowering in our conduction models as the operation voltages for memristive devices are much smaller (Figure 1). Thus, we believe our derivations are valid for memristive systems.

In the revised version of manuscript, we added the conditions to our models that: in the case that the reverse bias current is not affected by image-force barrier lowering. We thank the reviewer for pointing out this issue.

Q11: If the ionic current is only the little bump, the overall conclusion may still be valid, but the explanation needs to be updated.

R11: Indeed, the redox peak / bump is representing the electrochemical reaction. We cannot quantitatively claim what is the ionic transference number. Redox reactions (including of moisture) can also increase the electronic current (see *Adv. Electron. Mater.* **2017**, 4, 1700458, 2017). Please, consider as well our responses **R9,a,b** and **R10**.

Q12: In SI Fig. S12, for the first cycle, the authors should add arrows to indicate the orientation of the considerable hysteresis. It does not seem self-evident whether the current actually saturates or whether the hysteresis is ‘the other way around’.

R12: We thank the reviewer for this constructive comment. In the revised version of Supplementary Information, we have added arrows in Supplementary Figs. 12-14 (Supplementary Figs. 16-18 in the revised version) to indicate the orientation of current density loops.

Q13: If in the analysis of the ionic currents, Fig 2a and SI Fig. S12, the strong current decrease of the second and third cycle, with respect to the first, is due to Ta oxidation of the bottom electrode, surely this would also happen in the actual devices with thinner oxide layers? In that case they would not be as stable and good performance as demonstrated in Fig. 1, so this explanation of Ta oxidation seems inconsistent. The authors argue that indeed, the same happens in the Ta/Hf electrode devices, so why does this not affect the device performance in Fig. 1? It can't be due to the capping, because the authors argue that it is metal oxidation between the electrode and the oxide film.

R13: This is an important question. We would like to point out the differences in the thickness of the Ta₂O₅ film for the devices, used for CV (30 nm Ta₂O₅) and the devices used for switching performance (8-nm Ta₂O₅ ohmic memristive devices), where the metal thickness is the same. For devices with 8 nm thin oxide the metal was only partially oxidized at metal/oxide interface due to oxygen exchange, the ohmic electrodes remain metallic as evidenced by the XPS analysis (Supplementary Fig. 15). The ohmic-like interface in ohmic memristive devices also prevents voltage drop across the metal/oxide interfaces, thus the passivation effect was significantly suppressed, and we achieved reliable resistive switching as shown in Figure 1. The devices studied in Figure 2a and Supplementary Fig. 16 consist of high Schottky barrier heights, the operation voltages during cyclic voltammetry were also high which in turn severely passivated Ta electrode. This explains the decrease of current density in the second cyclic voltammogram (Supplementary Fig. 16).

For 30-nm thick Ta₂O₅ devices the passivation effect was the main reason for the current decrease. In the 30-nm oxide there is more oxygen that can interact with the metal electrode, causing passivation effects. Please note, that even with these devices we do not have observed complete passivation (complete blocking), but a decrease of the current due passivation. Operation of the devices there is still possible.

Please, consider as well our recent work (*Adv. Mater.* **2022**, *34*, 2105022), where we demonstrate the importance and effects of the thickness of the individual layers within the device stack. The results there evidenced that the thickness ration (metal electrode/switching oxide) is determining the behaviour.

Q14: I have a hard time following the network simulations. It may simply be outside of my expertise, so that other reviewers may be more suitable to evaluate this. For reference, I would claim that I have a solid understanding of the fundamentals of resistive switching networks and their training and operation, but I am not an expert in the various advanced modes of operation. From this perspective, there is too little explanation in the manuscript to describe the complexity of the simulations and their results. It may be informative to experts on neural networks, but I am afraid the findings will be lost to a broader audience and that is a shame, given the hard work that the authors put in. As a suggestion, maybe the authors could present the main findings in the main manuscript in a more simplified and stringent way, but then include a more detailed discussion in the SI. This could be a way to make the results accessible to a broader audience. Either way, Fig. 5 is very busy and some of the schematics do not really help my understanding.

R14: We thank the reviewer for this constructive comment. To make the results more accessible to the broader audiences, we have reorganized the content of Fig. 5 and added detailed discussions in the main text (highlighted by yellow color in the section of “Neural network application” in the revised manuscript) and in the Supplementary Note 2. We hope it can help the readers understand the details of the simulation. The rearranged Fig. 5 can be seen as following, in which we have deleted some redundant figures to make the results clearer. The deleted figures which are related to the neural network performances for devices’ fail ratios are moved to Supplementary Fig. 33.

Fig. 5. Overcome catastrophic forgetting in deep neural networks. **a**, Schematics of catastrophic forgetting problems when neural networks are trained by different tasks sequentially. **b**, The testing accuracies for three different tasks during the sequential training of the neural networks, whose weights were set as float point or binary values respectively. **c**, Schematics of the hidden states within synapses, which can influence the plasticity of each synapse. **d**, Diagrams of the neural-networks implementation inspired by the higher order synaptic plasticity. **e,f**, Final testing accuracies of all the trained tasks concerning the quantized bounds and bit precision. **g**, Neural network performances for different devices' fail ratios during the continual training process. **h**, The impacts on the continual learning capacity of the neural network model under the different multilevel programming errors.

Supplementary Fig. 33. The schematic of impact on cell updating when device is stuck at LRS. Neural network performances for different epochs in training each task under the same failure ratio at 0.2%.

Q15: Possibly based on my limited understanding, is the specific switching mechanism of the presented devices a necessary property to provide the simulated results? Or can the same functionality be achieved by devices with other switching mechanisms, e.g., any other multi-level resistive switching?

R15: The well-defined binary and analog switching characteristics induced/caused by the specific new switching mechanism is much more adequate for the continual learning algorithm in this work. However, it is not absolutely necessary property to provide the simulated results for the specific switching mechanism of the presented devices. The same functionality can be achieved by devices with the similar switching mechanisms or the binary and multi-level resistive switching. To demonstrate the advantages of the ohmic memristors, the low failure ratios and large HRS and LRS resistance ratio of the presented devices compared with conventional VCM are explored.

As for the low failure ratios, as shown in Fig. 5g, in the frequent training, the high reliability can keep the neural network learning new tasks continually. About the large HRS and LRS resistance ratio of the ohmic memristors, as shown in Figure R1.9c. Compared with ohmic memristors, the overlap between different states of the conventional VCM is more severe. The continual learning capacity of neural network can be seen in Figure R1.9d-f, in which neural network implemented by the ohmic memristor is superior to the conventional VCM.

Figure R1.9 has been added as Supplementary Fig. 34 in the revised Supplementary Information.

Figure R1.9: The LRS and HRS distribution of conventional VCM (a) and Ohmic memristor (b), in which the R_{HRS}/R_{LRS} ratio of Ohmic memristor is larger than that of conventional VCM. (c) The schematic of the overlap among multi-level conductance states in the conventional VCM and ohmic memristor devices. (d,e) Neural network performances for the conventional VCM and ohmic memristor under the same programming errors. (f) The comparison of the final accuracies of different tasks for the conventional VCM and ohmic memristor.

Some minor comments:

Q16: The pulses used for endurance measurements were fairly long (milliseconds). The manuscript and the presented devices would be strengthened if the authors included measurements with shorter pulse widths, e.g., trying the limits of switching speed. The equipment listed in the methods part is certainly capable of such measurements.

R16: We thank the reviewer for this constructive comment. In our work we demonstrated switching using different pulse widths (please, refer to Fig. 4a of the main manuscript). The ohmic memristive devices can be operated at sub-microsecond range and reached switching speed to 100 ns, as shown in the SET kinetics measurement in Figure 4a.

Q17: On p. 9, the statement of the metallic phase by EDS does not seem convincing. In SI Fig. S10, there is oxygen detected in the hafnium layer. So there could still be oxidation. Maybe the authors meant to reference XPS in SI Fig. S11?

R17: EDS has limited atomic resolution in oxygen mapping, thus we may “see” small signals of oxygen in metal layers such as Hf, Ta and even Pt. This does not mean that the metal layers have been oxidized. The XPS results also support the metallic phase of ohmic Ta and Hf electrodes. In the revised version of manuscript, we have amended this sentence by: “as the energy-dispersive spectroscopy (EDS) and X-ray photoelectron spectroscopy (XPS) analysis reveal the presence of the metallic phase of ohmic electrodes (Supplementary Figs. 14,15)”.

Q18: Introduction, technically, the memristor was introduced as an element linking charge and flux, not resistance and charge.

R18: Indeed, in the very first paper of Leon Chua, the link is between flux and charge, having the dimension of *resistance*. It has been presented as a current-voltage plot in 2008 by the S. Williams, representing the relation between charge and resistance. The argumentation there is well accepted in the society and Leon Chua himself.

Q19: In the introduction, the authors write that their system is reduced in complexity, but follow up immediately with pointing out rich physicochemical dynamics, this is not quite consistent.

R19: Both systems have rich physicochemical dynamics. The system we offer is simpler compared to other VCM systems as the influence of the Schottky barrier is eliminated. In that sense, we believe our statements is correct and hope the reviewer would agree with our argumentation.

Q20: Already at the end of the introduction, it would be helpful to point out whether the network implementations are experimental or simulations.

R20: We thank the review for this comment. We have included in simulation information in at the end of introduction as “*we further conduct neural networks simulation demonstrating the ohmic memristive device can be implemented in artificial neural networks model and achieve high pattern recognition accuracies on multiple image datasets.*”

Q21: I recommend not putting the SI tables at the end, but include them with the rest of the document in order of appearance in the main text. This will make it easier to follow the SI along with the main manuscript.

R21: We thank the review for this comment. We have moved Supplementary Table 1 and Table 2 according to their appearance in the revised manuscript.

Q22: The 3D illustration of the device structure in Fig. 1a is a bit misleading, as it does not represent the actual physical layout. Yet it might be understood to imply that there is some 3D device structure.

R22: In Figure 1a we aimed to show the layers information of fabricated crossbar devices. A scanning electron microscopy image of the device is shown in Supplementary Fig. 14a. In the revised version of manuscript, we have further explained the device structure in the caption of Figure 1.

Q23: I recommend adding numbered arrows to all IV curves (linear, log, main manuscript, SI) for increased figure legibility.

R23: We thank the reviewer for this constructive comment. In the revised version of manuscript and supplementary information, we have added arrows to all IV curves to show the sequence of IV loops.

Q24: The figure caption of SI Fig. S4 is not very explanatory. Some more information would clarify.

R24: Thanks for your kind suggestion, which would help us to clarify the contents of Supplementary Fig. 4 (Supplementary Fig. 6 in the revised version). To clarify the information, we have added necessary explanations about the differential conductance and data acquisition.

Supplementary Fig. 6. The differential conductance distribution for different devices with 1000 endurance cycles, HRS and LRS are read at -0.2 V, Set and Reset operation are conducted at -1.2 V/30 ms and 3.2 V/2 ms, respectively. The differential conductance is defined as the difference between device conductance at LRS and HRS, and Gaussian distribution function is employed to fit the differential conductance distribution for each device, mean value and standard deviation are both reflected in each subplot.

Q25: On p. 8, I don't understand why the authors compare their devices with SI Fig. S6, which is measured with larger voltages. SI Fig. S1 already showed comparison devices with Pt top electrodes, why not compare with this instead?

R25: We thank the reviewer for this constructive comment. The purpose of comparing ohmic memristive device to Supplementary Fig. 6a (Supplementary Fig. 8a in the revised version) is to show that the new device can sustain higher RESET sweep voltages thus overcome RESET failure which is a typical failure reason in conventional VCM devices (Supplementary Fig. 8a). For both types of devices, the maximum V_{RESET} was kept to be 3V for comparison. In conventional VCM device the resistive switching only lasted for a few cycles and show device failure. In comparison, under the same condition the ohmic memristive show more reliable switching (see Figure 1d). We have provided more detailed comparison between conventional VCM and ohmic memristive devices, please refer to our responses in R5.

Q26: In SI Fig. S7, shouldn't the decision "Loop > Max_loop" have Y and N exchanged? I.e., end the script when max_loop is exceeded?

R26: Thanks for your detailed review, after carefully checking the flowchart of the write-verify programming algorithm in Supplementary Fig. 7 (Supplementary Fig. 11 in the revised version), we indeed made a mistake in the decision about "Loop > Max_loop", the script will end when the loop exceeded the max_loop.

To address this issue, we have corrected the flowchart and redrawn it in Supplementary Fig. 11.

Supplementary Fig. 11. Flowchart of the write-verify programming algorithm. It mainly includes two modules as SET and RESET, both work in negative feedback principle.

Q27: In summary, I think the authors have presented a promising manuscript, but in its current state there seem to be two main parts, which are not linked very smoothly. These are the very promising electrical resistive switching data, backed by a very good TEM-based analysis, and separate from this the CV-based analysis, which needs clarification. The network simulations seem too complex to include with the very short provided text. If the authors can explain more convincingly that indeed, they found a new kind of switching mechanism, and can integrate the two parts better, I think a revised manuscript can be suitable for Nature Communications. If the mechanism, however, is 'only' filamentary with low work function electrodes, a different journal would be more suitable. Please note that by 'only', I do not mean to diminish the authors' work. I think the measurements and results are very impressive and promising, and 'only' is just meant to differentiate between a possible new mechanism and a different iteration of similar existing mechanisms. The results should definitely be published. But at this point they still need refinement of what the main findings actually are.

R27: We thank again the reviewer for the comments and suggestions. We hope, we were able to convince him/her, that we report on a new switching mechanism. We also have discussed in our response **R6** that the work includes three main parts relating fundamental science (electrochemical and switching mechanism), with devices performance and network simulations. The conclusion and discussion are supported by electrical (CV, IV, pulse) measurements, XPS, and TEM/EDS analysis. In our view the results are inherently related and the other reviewers have also not raised concerns about the integrity of

the text and discussion. We hope very much the reviewer will accept the responses and the revisions/amendments of the manuscript, based on a significant amount of additional analysis and experimental work.

Reviewer #2 (Remarks to the Author):

General comment: First, this paper applies low work function metals to both top and bottom electrodes to demonstrate an Ohmic memristive device and provides a reasonably detailed description of the resistive switching mechanism. I found no logical inconsistencies in the various approaches to describing the mechanism, and overall, it is well-written. Additionally, it seems to be a suitable manuscript for the Nature Communications journal.

Our response: We appreciate the reviewer for the positive evaluation of the manuscript.

However, to enhance its completeness before publication, I suggest the following points.

Q28: The authors claim in the manuscript to present a new switching mechanism. Upon reading this manuscript, a fundamental question arises: there seems no reason why switching characteristics would not be observed when both electrodes are made of low work function metals, even when compared to the reported filament type mechanisms. One reason for using a high work function metal (or inert metal) like Pt at one electrode is to ensure reliability by preventing undesired reactions at one interface and inducing switching exclusively at the other. Using low work function materials on both sides seems likely to decrease reliability, and I would appreciate the authors' comments on this.

R28: We thank the reviewer for this very important comment. Indeed, as the reviewer pointed out there is no theoretical reason, why switching characteristics of devices with two low work function metals should not be observed. Nevertheless, there were no reports on this in the literature, and several unsuccessful attempts making VCM devices with two low function electrode metals. We are the first that succeeded to demonstrate this. Many reasons may apply, why this effect has not been reported before but in our manuscript, we would prefer not to speculate on the reason.

We agree fully with the reviewer, that high work function electrodes are used to avoid undesired reactions at this interface. The electrodes in the stack are (electro)chemically asymmetric and also in terms of their work function. Many groups have reported “conventional” VCM with excellent performance. However, our work and analysis on the metal/oxide interface (in terms of interplay between Schottky and redox barriers) shows disadvantages having too high Schottky barrier. For example, having high Schottky barrier height may cause high voltage drops across the interface and making the forming voltage and SET voltages rather high and can easily cause irreversible breakdown of resistive switching oxide (see Supplementary Fig. 8a). It may also cause formation of too strong filament where the RESET process can be inhibited/ partially suppressed leading to limited endurance or limiting using higher voltages (to achieve faster switching) compared to devices with ohmic interfaces on both sides. In addition, we would like to mention, that “conventional” VCM have been improved over a decade, whereas our devices demonstrate comparable performance from the first attempt. In that sense, we focus on the advantages our system is providing. We are convinced their performance can be improved a lot by further work.

In case of our ohmic device we succeeded to stabilize our stack system avoiding undesired reactions without having a Schottky barrier. This was possible due to the materials combination (metal electrodes, thicknesses and switching oxide). The driving force for the switching is the reversible oxidation/reduction of the filament. Here the filament is not completely dissolved (in contrast to ECM), but is partially oxidized (passivated), due to change of the oxidation state of the metal ions (and related oxygen content, leading to highly insulating stoichiometric oxide) and change of the length of the filament. OFF state is independent on the Schottky barrier but solely governed by the

composition/thickness modulation of the filament. This also mitigates voltage drop across the interfaces thus improving the reliability and stability.

The most critical point was to stabilize the filament in OFF state. This was achieved by using ohmic material (in our case Hf) that was not able to migrate into the TaOx filament (as our TEM studies have shown). We believe this was the most essential point.

Q29: In Supplementary Fig. 11, XPS data measured at the center of each layer would naturally observe the metallic state. Oxidation of the electrode and reduction of Ta₂O₅ at the interfaces are expected, which might lead to Ohmic contact (reference [1]). The redox at the both interfaces, rather than the electrode's low work function, seems to be the primary cause of Ohmic contact. I would appreciate the authors' comments on this.

R29: We thank the reviewer for this comment. Indeed, the oxygen interaction of at metal/oxide interface is a very important factor that leads to ohmic contact (we have cited reference [1] in the revised manuscript (ref.37)). If one considers the e.g. Ta/Ta₂O₅ contact theoretically and calculates the work function difference one will estimate a Schottky barrier of about 1 eV. However, ohmic electrodes such as Ta, Hf, Zr, Ti, have high oxygen affinity that can scavenge oxygen from the surrounding oxide layer, resulting in partial reduction of the oxide (due to the loss the O²⁻) and partial oxidation of ohmic electrode at metal/oxide interfaces, formatting substoichiometric, conductive metal oxides. The modified interface is significantly reducing the effective Schottky barrier height.

Q30: The paper states that tantalum oxide used is in the amorphous phase. As the authors know, TaO can have various oxidation states depending on the fabrication method, and both filament and interface types have been observed depending on the oxidation state. The authors already have XPS data, so I recommend deriving the composition from XPS.

R30: We thank the reviewer for this constructive comment. From the binding energies of and the XPS spectra obtained from XPS depth profiling experiments, we have derived the composition of the oxide film. As can be seen from Figure R2.1, the main peaks appear at binding energies of 26.6 eV and 28.4 eV, which correspond to the binding energies of Ta 4f_{7/2} and Ta 4f_{5/2} of Ta⁵⁺(Ta₂O₅). The results show that switching oxide is mainly comprised of stoichiometric Ta₂O₅. We also found that part of the oxide layer was reduced to Ta₂O₃ (Ta³⁺) and TaO (Ta²⁺), evidenced by the Ta 4f_{7/2} peaks at 24.3 eV and 22.8 eV, and Ta 4f_{5/2} peaks at 26.1 eV and 24.7 eV, respectively.

We have amended this information in Supplementary Figure 15.

Figure R2.1: XPS analysis showing the composition of sputtered tantalum oxide film.

Q31: I am curious about the stability of the retention characteristics. The author shows measurements for 1200s, but this duration is too short. It would be better to demonstrate retention characteristics for at least two states over 24 hours.

R31: We thank the reviewer for this constructive comment. We have performed additional experiments to verify this. For the retention of 1200s that we have shown in the manuscript, the devices were biased under a constant voltage stress of 200 mV. We didn't observe any distinct drift of the state. We also didn't observe degradation of the conductance levels in the measurement after two days (Figure R2.2). This figure has been added in Supplementary Information (Supplementary Fig. 4).

Figure R2.2: retention of HRS and LRS resistance for over two days.

Q32: Figure 1g shows multi-bit characteristics. It would be beneficial to also show DC curves within the Vread, as in the attached reference [2]. Implementing multi-bit in an Ohmic regime would be advantageous when configuring a vector matrix multiplier of using analog inputs.

R32: We thank the reviewer for this constructive comment. The results in Fig. 1g shows multi-bit results obtained by write-verify programming scheme. We have also added the multi-bit results measured in DC voltage sweeps in Supplementary Information (Supplementary Fig. 10). The reference [2] has also been cited in the revised manuscript (ref.32).

Figure R2.3: DC IV curves showing multi-bit switching in ohmic memristive devices.

Q33: Minor things: (1) Supplementary Figs. 19, 20, 27, and 29 are not mentioned in the main body. All Supplementary Figures should be described. (2) Some Supplementary Figures are out of order. The numbering should match the order mentioned in the text.

R33: We thank the reviewer for pointing out this mistake. We have corrected it in the revised version of manuscript.

References

- [1] Yang, J., Pickett, M., Li, X. et al. Memristive switching mechanism for metal/oxide/metal nanodevices. *Nature Nanotech* 3, 429–433 (2008). <https://doi.org/10.1038/nnano.2008.160>
- [2] Rao, M., Tang, H., Wu, J. et al. Thousands of conductance levels in memristors integrated on CMOS. *Nature* 615, 823–829 (2023). <https://doi.org/10.1038/s41586-023-05759-5>

Reviewer #3 (Remarks to the Author):

The paper reports a new structure design of memristors, with low-work-function electrodes to form a two-terminal Ohmic memristor. By mitigating or eradicating the Schottky contact, the new device shows improvement in reliability and stability. Detailed experimental and theoretical analysis of its working principle and dynamics are given. Possible application of the proposed device in ANN training for continuous learning is also demonstrated. The manuscript is well-structured and written. The topic of exploring a new switching mechanism for improving device performance is interesting and important, and worth further exploring for the community. Still, there are some concerns that need to be addressed to improve the manuscript.

Our response: We thank the reviewer for the positive evaluation of our manuscript and for pointing out the importance of exploring switching mechanism for improving device performance. Enclosed please find our point-by-point response to your comments.

Q34: First, the paper lacks a more straightforward and quantitative statistical comparison between their new devices and the traditional VCM system (e.g., their fabricated Pt/Ta₂O₅/Ta; Hf/Ta₂O₅/Ta memristors).

R34: We thank the reviewer for the constructive comments. Following your advice, we have provided more thorough comparisons between traditional VCM devices and ohmic memristive devices in electrical characteristics (forming, current levels, endurance, HRS and LRS resistance ratio) and neural networks performance: forming (Figure R3.2a,b), current levels (Figure R3.2,c,d), endurance and HRS/LRS resistance ratio (Figures R3.3 and R3.4), and neural networks performance (Figure R3.5). The data is also added in the supplementary information of the revised manuscript.

Q35: It is suggested to mention whether Ta is the bottom electrode or top electrode in those conventional VCM for comparison.

R35: Regarding Pt/Ta₂O₅/Ta VCM devices. The data used for comparing with ohmic memristive devices were from Ta(Te)-based devices. In the revised version of manuscript, we have provided this information in the Method section: “*For the comparison resistive switching characteristics, conventional VCM devices with Pt/Ta₂O₅/Ta(Hf) structure were fabricated, where Ta and Hf electrodes were used as top electrodes, respectively.*” In fact, we have fabricated conventional VCM devices using Ta as top electrode (TE) and bottom electrode (BE), respectively, and did not observe significant difference in electrical properties, as can be seen from Figure R3.1. Both types of devices do not differ in resistive switching phenomenon, i.e., they show reproducible bipolar resistive switching with identical SET and RESET voltages (Figure R3.1a) and nearly the same current levels at HRS and LRS states (Figure R3.1b).

Figure R3.1: Direct comparison of Pt/Ta₂O₅/Ta devices where Ta was used as bottom electrode (a) and top electrode (b), respectively (10 cycles each). The insets in panel b shows the biased conditions (voltages applied to Pt electrodes).

Q36: This comparison should cover aspects such as forming voltage, retention, read precision, cycle-to-cycle stability, device-to-device uniformity, and endurance. While they did mention that the conventional VCM can fail in LRS during the RESET process, it is recommended to provide detailed statistics, such as the number of cycles before failure and the number of cycles/devices tested on Ohmic memristors. It is worth noting that the write endurance of conventional VCM is reported to be at least 10⁶. Without quantitative data and a direct comparison, claiming the new device has better performance is less convincing.

R36: We thank the reviewer for the constructive comments. Regarding the forming comparison, as shown in Figure R3.2a,b, conventional VCM device has a forming voltage of ~-4V. For the ohmic memristive device, as can be seen in Figure R3.2, the forming voltage was reduced (~-1.8V) due to the removal of Schottky barrier (Figure R3.2b, also Supplementary Fig. 3a). Moreover, as shown in Figure R3.2 that in conventional VCM device, the RESET current after forming process is at milliamperage range. The high forming voltages and high RESET currents in conventional VCM devices make overall devices conductance very high ($R_{HRS} \sim 10$ kΩ, see Figure R3.2c), due to the massive generation of oxygen vacancy defects. As a result, the devices typically show small HRS and LRS resistance ratio (~10, $V_{read} = -0.2$ V) under endurance test (Figure R3.3a, Supplementary Fig. 1e,f). Note that the cycle-to-cycle uniformity, endurance, state retention in conventional VCM can be very good under high currents levels (the conductive filament consists of higher concentration of oxygen vacancy). Figure R3.3a-c shows device-to-device and cycle-to-cycle uniformity of conventional VCM devices (Ta/Ta₂O₅/Pt), which exhibit narrow distribution of R_{HRS} and R_{LRS} values, i.e., high cycling uniformity. However, the resistive switching R_{HRS}/R_{LRS} ratio is usually ~10. However, to achieve higher R_{HRS}/R_{LRS} ratio, one must increase the RESET voltage thus the R_{HRS} is lower (barrier higher between oxide/Pt interface is higher). On the other hand, the device typically suffers from early operation failure under high RESET voltage, as shown in Supplementary Fig. 8a, where the device failed in the 17th cycle (we have provided the failure information in the Supplementary Fig. 8a).

Figure R3.2: Comparison of forming. (a) Exemplary forming and (c) resistive switching (20 cycles) characteristics of Pt/Ta₂O₅/Ta device. (b) Exemplary forming and (d) resistive switching (20 cycles) of Hf/Ta₂O₅/Ta device. The inset plots in panels c and d show corresponding resistance change as a function of applied voltages.

In comparison, there's no current overshoot after forming in ohmic memristive device (Figure R3.2b), i.e., the reset current is at the same range of set current. The device has HRS state resistance ~ 300 k Ω and HRS and LRS resistance ratio higher than 100 ($V_{\text{read}} = -0.2\text{V}$), the endurance was measured to be higher than one million cycles (Figure R3.3e-f). Figure R3.4 depicts direct comparison of endurance between ohmic memristive device and conventional VCM devices. Despite the endurance of VCM has been reported higher than 10^6 , the $R_{\text{FF}}/R_{\text{ON}}$ is typically lower than 10 and the overall device currents are high, which results in high power consumption.

Figure R3.3: Comparison of ohmic memristive devices with conventional VCM devices.

Figure R3.4: Comparison of endurance, HRS and LRS resistance ratio, high resistive state current levels with reported VCM devices. Ohmic memristive devices show higher HRS/LRS resistance ratio with good endurance cycle. Moreover, the overall current levels in ohmic memristive devices are lower than other devices, indicating lower power consumption in ohmic memristive devices.

Based on above results, we are convinced that the ohmic memristive devices exhibit more controllable forming processes (no current overshoot, low forming voltages), higher HRS and LRS resistance ratio (>100x), lower R_{HRS} (lower power consumption), and good endurance (>10⁶). These performances can hardly be achieved simultaneously in conventional VCM devices (Figure R3.4). We understand the reviewer’s concern on the performance of ohmic memristive devices, studying the ultra-limits on endurance, multibit switching is the focus of our future work.

In the revised version of Supplementary information, we have added the forming IV curve (Figure R3.2a) and device-to-device variation and endurance characteristics (Figure R3.3a-c) of conventional VCM in Supplementary Fig. 1. Device failure information has been provided in the figure caption of Supplementary Fig. 8. Figure R3.4 has been added as Supplementary Fig. 9. The number of tested ohmic memristors has been provided in the Method section of the revised manuscript. The number of tested cycles is provided in the captions of the figures that show IV characteristics.

Q37: Second, it is not very clear what ‘unique properties’ of the ohmic memristor enable continual learning. The authors mentioned “The binary and analog switching capability in the ohmic memristive device,” but conventional VCM can do the same. It is suggested to provide a comparison of network performance from different types of devices to substantiate the claim, regardless of this application or any other application that is better suited for this device.

R37: We thank the reviewer for this important comment. We have added the comparison in neural networks performance between conventional VCM and ohmic memristors. The ‘unique properties’ of ohmic memristor such as well-defined binary and analog switching characteristics, large HRS and LRS resistance ratio compared with conventional VCM, no reset failure, can make the ohmic memristor more favorable for continual learning application. The advantages of low failure ratios of the ohmic memristor has been explored in Fig. 5g and Supplementary Fig. 33. Moreover, the bigger margin of the ohmic memristor can help the neural network tolerate greater noise amplitudes. The comparison between ohmic

memristor and conventional VCM is shown in Figure R3.5. To reduce the latency and costs in the analog programming, we set the relatively large programming error in the write-verify programming scheme, specifically, we set the error as $2.5\mu\text{S}$ and number of analog states as 4-bit. As shown in Figure R3.5c, the overlap between different resistance states in conventional VCM is more severe than that of the ohmic memristor, due to the small HRS and LRS resistance ratio. The continual learning capacity of neural network is shown in Figure R3.5d-f, in which neural network implemented by the ohmic memristor is superior to the conventional VCM in pattern recognition accuracy. The comparison of neural networks performance has been added in Supplementary Fig. 34.

Figure R3.5: The LRS and HRS distribution of conventional VCM (a) and ohmic memristor (b), in which the $R_{\text{HRS}}/R_{\text{LRS}}$ ratio of ohmic memristor is larger than that of conventional VCM. (c) The schematic of the overlap among multi-level conductance states in conventional VCM and ohmic memristor devices. (d,e) Neural network performances for the conventional VCM and ohmic memristor under the same programming errors. (f) The comparison of the final accuracies of different tasks.

Q38: In Figure 1g & h, which illustrates the device's precision and uniformity, it is challenging to compare the device's performance with other VCM devices due to the conductance ranging across several mS. It would be more informative to discuss if their unique device structure enables this mS-level conductance range. For instance, this could be caused by the removal of the Schottky contact or the large cross-size. This raises another concern: are devices with high mS-level conductance favorable in applications? Such high conductance would lead to high current and a more significant IR drop issue in an array. Therefore, it is recommended that the authors address this concern in their discussion.

R38: We thank the reviewer for this comment. As discussed in R1 ohmic memristors have higher HRS and LRS resistance ratio and can enable device conductance levels range from μS (Figure R3.6 and R3.7) to mS (Figure 1h). Figure R3.7 has been added to Supplementary Fig. 9 in the revised Supplementary Information. As discussed at Supplementary Fig. 35, in the feedback write-verify method, the final weight values have included line resistances, so we didn't consider the effect of the line resistances into the actual hardware implementation. However, to make the devices applicable to the large-scale arrays and reduce energy costs, the multi-level conductance states are expected to be reduced around μS , as indicated in Figure R3.6 and Figure R3.7. In the binary weight switching, which does not include any

feedback mechanisms, the IR-drop induced by the line resistances has been considered, whose effects can be seen in Supplementary Fig. 35, and the conductance states are adopted at the small-size devices. As shown in the results, owing to the relatively small neural network structure, the hardware design can tolerate a large range of line resistances.

Figure R3.6: The μS -level conductance states of smaller-size devices ranging from $1 \mu\text{S}$ to $70 \mu\text{S}$, which are programmed under the write-verify scheme.

Figure R3.7: retention characteristics of conductance states at μS -level.

Q39: In the "Methods" section, they mention "the active area of the fabricated devices ranges from $4 \mu\text{m}$ to $2500 \mu\text{m}$ ". But they don't specify the exact lateral size for the device performance data they report earlier.

R39: According to reviewer's comment, we have specified the sizes of measured devices in the "Methods" section of revised version of manuscript.

Q40: The discussion in at the end of page 4 ("Despite significant progress has been reported and the efforts made to achieve high-performance and reliable electrical properties... making the device operation rather complicated." is confusing. As from my understanding, the devices proposed by the authors also works in "small volumes of charge enrichment/depletion", which will also suffer from related "non-idealities in state retention, endurance, and uniformity". Please clarify the logic here over the main topic of this paper.

R40: We thank the reviewer for the comment. Indeed, as mentioned by the reviewer the text paragraph applies to all memristive devices. Our ohmic memristive devices are not excluded from this discussion. All they suffer from these effects. The whole paragraph is aiming to show that despite significant progress have been made in the field there are still several limiting factors and difficulties. We also have not written/claimed that we have eliminated any of these factors by using ohmic memristors.

To comply with reviewer comment we have added following text to the paragraph: “...a continues fundamental research in this field is essential in order to improve the control over the nanoscale processes, reduce the level of complexity of the systems, exploring new materials and switching mechanisms.”

Q41: There seem to be some drawing errors in Supplementary Fig. 4 (device 9), where the bar chart looks narrow while the fitted curve looks much wider.

R41: Thanks for your detailed review, after carefully checking the fitting results of differential distribution of device 9 in Supplementary Fig. 4 (Supplementary Fig. 6 in the revised version), we made mistakes at limiting the fitting scope, after refitting the distribution of device 9, we also added more descriptions about the Supplementary Fig. 6, which is illustrated in the following:

Supplementary Fig. 6. The differential conductance distribution for different devices with 1000 endurance cycles, HRS and LRS are read at -0.2 V, Set and Reset operation are conducted at -1.2 V/30 ms and 3.2 V/2 ms, respectively. The differential conductance is defined as the difference between device conductance at LRS and HRS, and Gaussian distribution function is employed to fit the differential conductance distribution for each device, mean value and standard deviation are both reflected in each subplot.

Reference:

- [1] Prakash, A., Jana, D., Samanta, S. & Maikap, S. Self-compliance-improved resistive switching using Ir/TaOx/W cross-point memory. *Nanoscale Res Lett* 8, 527 (2013).
- [2] Akbari, M., Kim, M.-K., Kim, D. & Lee, J.-S. Reproducible and reliable resistive switching behaviors of AlO_x/HfO_x bilayer structures with Al electrode by atomic layer deposition. *RSC Advances* 7, 16704–16708 (2017).
- [3] Berthaud, F. et al. In-Depth Analysis of Transistor Influence on OxRAM Performance in Memory Bitcell, With Technology Scaling Perspectives. *IEEE Trans. Electron Devices* 71, 2721–2728 (2024).
- [4] Shin, D. H. et al. Multiphase Reset Induced Reliable Dual-Mode Resistance Switching of the Ta/HfO₂/RuO₂ Memristor. *ACS Appl. Mater. Interfaces* 16, 13, 16462–16473 (2024).
- [5] Chen, Z. et al. High-performance HfO_x/AlO_y-based resistive switching memory cross-point array fabricated by atomic layer deposition. *Nanoscale Res Lett* 10, 70 (2015).
- [6] Munjal, S. & Khare, N. Valence Change Bipolar Resistive Switching Accompanied With Magnetization Switching in CoFe₂O₄ Thin Film. *Sci. Rep.* 7, 12427 (2017).
- [7] Beckmann, K., Holt, J., Manem, H., Van Nostrand, J. & Cady, N. C. Nanoscale Hafnium Oxide RRAM Devices Exhibit Pulse Dependent Behavior and Multi-level Resistance Capability. *MRS Adv.* 1, 3355–3360 (2016).
- [8] Chen, Y.-S. et al. Good Endurance and Memory Window for Ti/HfO_x Pillar RRAM at 50-nm Scale by Optimal Encapsulation Layer. *IEEE Electron Device Lett.* 32, 390–392 (2011).
- [9] Chakrabarti, B., Galatage, R. V. & Vogel, E. M. Multilevel Switching in Forming-Free Resistive Memory Devices With Atomic Layer Deposited HfTiO_x Nanolaminate. *IEEE Electron Device Lett.* 34, 867–869 (2013).
- [10] Mahata, C., Kang, M. & Kim, S. Multi-Level Analog Resistive Switching Characteristics in Tri-Layer HfO₂/Al₂O₃/HfO₂ Based Memristor on ITO Electrode. *Nanomaterials* 10, 2069 (2020).
- [11] Kim, W. et al. Forming-free metal-oxide ReRAM by oxygen ion implantation process. in 2016 IEEE International Electron Devices Meeting (IEDM) 4.4.1-4.4.4 (2016).
- [12] Lee, M.-J. et al. A fast, high-endurance and scalable non-volatile memory device made from asymmetric Ta₂O_{5-x}/TaO_{2-x} bilayer structures. *Nature Mater* 10, 625–630 (2011).
- [13] Ismail, M., Mahata, C. & Kim, S. Forming-free Pt/Al₂O₃/HfO₂/HfAlO_x/TiN memristor with controllable multilevel resistive switching and neuromorphic characteristics for artificial synapse. *Journal of Alloys and Compounds* 892, 162141 (2022).
- [14] González, M. B. et al. Synaptic devices based on HfO₂ memristors. in *Mem-elements for Neuromorphic Circuits with Artificial Intelligence Applications* (eds. Volos, C. & Pham, V.-T.) 383–426 (Academic Press, 2021). doi:10.1016/B978-0-12-821184-7.00028-1.
- [15] Ahn, M. et al. Memristors Based on (Zr, Hf, Nb, Ta, Mo, W) High-Entropy Oxides. *Adv. Electron. Mater.* 7, 2001258 (2021).
- [16] Ismail, M. et al. Improved Endurance and Resistive Switching Stability in Ceria Thin Films Due to Charge Transfer Ability of Al Dopant. *ACS Appl. Mater. Interfaces* 8, 6127–6136 (2016).

- [17] Mikhaylov, A. N. et al. Bipolar resistive switching and charge transport in silicon oxide memristor. *Materials Science and Engineering: B* 194, 48–54 (2015).
- [18] von Witzleben, M. et al. Study of the SET switching event of VCM-based memories on a picosecond timescale. *Journal of Applied Physics* 127, 204501 (2020).
- [19] Zaffora, A. et al. Electrochemical Tantalum Oxide for Resistive Switching Memories. *Advanced Materials* 29, 1703357 (2017).
- [20] Park, J. & Kim, S. Improving endurance and reliability by optimizing the alternating voltage in Pt/ZnO/TiN RRAM. *Results in Physics* 39, 105731 (2022).
- [21] Yang, J. J. et al. High switching endurance in TaOx memristive devices. *Applied Physics Letters* 97, 232102 (2010).

Response letter to the reviewers

We would like to thank the reviewers for reading our manuscript and for the (in general) positive assessment.

We realized that one of the major concerns is using the term “new mechanism” for our ohmic memristive devices. Complying with the concerns of the reviewers, we have carefully rethought this aspect and agree that indeed the term “new” is not precise. After careful consideration and analysis of our results and the literature we decided to remove it from the text and have introduced a term that most correctly describes our findings. Thus, we introduce in the manuscript the term “Filament conductivity Change Mechanism” (FCM).

We have summarized in Table R1 all the criteria that are used to characterize ECM and VCM mechanisms and compared it to the specifics of our FCM. From this comparison it becomes obvious that the switching mechanism we observed can be neither assigned to ECM, nor to VCM, because major criteria of either ECM or VCM are not fulfilled. Therefore, it was necessary to define the “Filament conductivity Change Mechanism” that captures the differences and specifics of the observed type of switching.

We provide in the point-by-point responses a full comparative analysis of the criteria of the switching mechanisms (ECM, VCM and FCM) and hope the reviewers will agree with our argumentation.

We also hope that the reviewers will find the responses and the corresponding revisions satisfactory. If any additional information is required, we will be more than happy to provide it.

Kind regards,

Shaochuan Chen, Yuchao Yang and Ilia Valov

Point-by-point response to the reviewers' comments

Reviewer #1 (Remarks to the Author):

Q1: Thank you to the authors for their efforts in replying to my comments. I appreciate and respect greatly the work which they invested. They have addressed several of my comments satisfactorily. However, the main point of the manuscript, the explanation of a new mechanism, is still not convincing to me. After clarification in the review comments, based on the provided explanations and experimental evidence, the mechanism looks like ECM filamentary switching to me with low work function electrodes. The authors even write this themselves in the discussion, where they use established ECM/VCM models to analyze their data. (For example l. 392: “The results fit well with the model presented previously for ECM devices”, l. 407 they conclude that the calculated values are “comparable to that determined for other ECM systems”, or in Fig. 4, where they use the plug/disc model.)

The manuscript can still be suitable for Nature Communications, even if there is not a discovery of a new mechanism.

R1: We thank the reviewer for reading and commenting on our revised manuscript. We have carefully considered the reviewer's comment and to address the concerns, have analyzed all criteria used to define ECM and VCM and compared it to our mechanism. In table R1 and related discussion the reviewer will see that the observed mechanism cannot be assigned to ECM and also not to VCM, as major criteria are not fulfilled. Therefore, we defined the Filament conductivity Change Mechanism (FCM), capturing the specifics of the type of switching.

To better demonstrate the differences in the switching mechanisms, here, we compare the key features and explain the fundamentals and differences on the switching mechanisms of ECM, VCM, and ohmic memristive devices. As shown in Table R1 and Fig. R1, the switching mechanism of ohmic memristive device cannot be assigned to ECM or to VCM mechanisms. Table R1 has been added as Supplementary Table 3.

Table R1: Comparison of working mechanisms between ECM, VCM and ohmic memristive devices. Specifics of the switching processes in ECM, VCM and FCM are shown in Fig. R1. (Abbreviation: AE, active electrode; IE, inert electrode; OE, ohmic electrode; SE, Schottky electrode)

Device type	Structure	Mobile specie(s)	Filament type	Resistance change	LRS	HRS
ECM	AE/Oxide/IE (e.g. Cu/SiO2/Pt)	Metal cation	Metallic	Metallic filament formation/dissolution	Metallic filament	Filament dissolved completely

VCM	OE/Oxide/SE (e.g. Ta/Ta ₂ O ₅ /Pt)	Oxygen anion (oxygen vacancies)	Reduced oxide Consists of Plug and Disc	Shift of oxygen vacancies in/out of the Disc and related Schottky barrier height change	reduced interfacial barrier height	increased interfacial barrier height
FCM	OE/Oxide/OE (Ta/Ta ₂ O ₅ /Hf)	Metal cation, oxygen anion	Reduced oxide	Change of filament conductivity due oxidation and length change and thus, the resistance	Filament of reduced oxide	Filament (partially) oxidized and with reduced length

1) **ECM mechanism**

As shown in Figure R1a, ECM stack is comprised of metal oxide layer sandwiched by active electrode (Ag or Cu) and inert electrode (Pt). By applying a positive voltage to the active metal electrode e.g. Cu, the SET process is triggered by the oxidation of active electrode, introducing cations at the metal/oxide interface, diffusion of these cations into the oxide, nucleation of Cu at counter electrode (Pt) and further growth of the Cu metallic filament by electrochemical reduction (processes 2-4). The formed Cu filament is bridging the top and bottom electrode, changing the device from HRS to LRS. Applying a reverse voltage will cause dissolution of the Cu filament, and the device recovers to high resistive state condition, changing the resistance from LRS to HRS (RESET process). Note that after the RESET process, the Cu metallic filament is completely dissolved.

2) **VCM mechanism,**

The switching film of metal oxide is sandwiched between one ohmic electrode that has high oxygen affinity, such as Ta, Ti, Hf, Zr, and one inert electrode that has high work function, such as Pt. The FORM process and subsequent RESET lead to formation of so-called Plug and Disc as shown in Fig. R1b (detailed description can be found in e.g. *Advances in Physics* 70(2), 155-349, 2021). Here the main mobile species are oxygen anions (oxygen vacancies). The inert electrode forms a Schottky-like interface with the tip of the filament (Disc). The subsequent SET/RESET cycles are driven by shift of the oxygen vacancies (mobile donors) within this Disc. Thus, the resistive switching in VCM devices is based on the redistribution/migration of oxygen vacancies within the DISC, which modulate the electronic barrier height at Schottky interface, as shown in Figure R1b.

Figure R1: Working principle and electrical characteristics of (a) ECM device, (b) VCM and (c) ohmic memristive devices. (a) Schematics of ECM switching processes. The central panel shows an exemplary I-V curve of a Cu/SiO₂/Pt ECM device. (1) – (5) shows different resistive state during one RS cycle. (1) Virgin state, the device is highly resistive. (2) Anodic oxidation of Cu enabled by the applied positive bias. The generated metal cations migrate toward CE (Pt) driven by the electric field. (3) Nucleation of cation at CE interface. Further accumulation of nucleus leads to the creation of metallic filament and SET process is achieved when the applied volage exceeds V_{SET}. (4) LRS state, the metallic filament bridges AE and CE and act as conduction channel for charge transport. (5) Dissolution of conduction channel by applying a reversed bias, the metallic filament is re-oxidized to cation and dissolved completely in RESET process. Adapted with permission from reference: Nanoelectronics and Information Technology, Wiley-VCH, 2012.

(b) Schematics of VCM switching processes. (A) HRS state, high concentration mobile donors, here oxygen vacancies (V_o^{••}, green spheres) make up the filament tip, called Plug region. (B) Under electrical bias, the SET process is achieved by the movement of oxygen vacancies from Plug to Disc. At Disc region, the oxide is gradually reduced (the valence of metal cations which are represented by purple spheres, is reduced) and the effective Schottky barrier height at Pt/oxide interface is lowered with the migration of oxygen vacancies. (C) LRS, increased concentration of V_o^{••} at Disc and decrease of effective barrier height cause high electronic conductivity. (D) RESET process, V_o^{••} are repelled from Pt electrode, resulting in the oxidation of metal oxide at Disc and the increase of energy barrier height at Pt/oxide interface. Reproduced with permission from reference: Advances in Physics 70(2), 155-349, 2021.

(c) Working principle and electrical characteristics of ohmic memristive device. By applying negative voltage to Hf (positive voltage to Ta), the Ta electrode is ionized, and Ta cations diffuse into the oxide, and along with oxygen ions (vacancies) form at the Hf/oxide interface a filament growing towards Ta electrode. After SET (LRS state) the filament is composed of reduced TaOx (Fig. 3a in the manuscript), not metallic. In RESET process, positive voltage is applied to Hf, the TaOx filament is oxidized (Ta valence is increased at filament tip), Ta ions are removed from filament region to deposit at the Ta electrode and in same time oxygen ions can move in the filament region, decreasing the filament conductance and thus device resistance. Depending on the amplitude of voltage bias, the filament can be re-oxidized to different valence state, resulting multiple resistance states (inset in the right plot in panel c).

3) Filament conduction Change Mechanism (FCM),

The metal oxide layer is sandwiched between two ohmic electrodes with low work functions (Fig. R1c). As a result, the electronic energy barriers at metal/oxide interfaces are substantially reduced. Thus, the device **differs from VCM devices where Schottky-barrier height modulation is the case of resistive switching, but also differs from ECM devices.** Both cations and anions can move/diffuse and contribute to the formation and oxidation of the filament. Indeed metal cations (in this example Ta-cations) also can move as evidenced cross-section transmission electron microscopy (Figure 3a,b in manuscript) and previous work. However, from the EDS elemental analysis, it is evident that the filament is not metallic, but made by substoichiometric TaOx. **This is completely different with ECM, where the filaments are fully metallic.** Moreover, in devices with different HRS resistances we observed partially formed (or partially dissolved) filaments of different lengths, which is much **different from VCM** (Plug is supposed not to vary in length) and **different from ECM** (filament completely dissolved in HRS).

Filament composition and the way that resistance change in ohmic memristive devices is different from VCM and ECM devices.

ECM criteria that are not fulfilled:

- Device structure/stack: active electrode/oxide/inert electrode
- Metallic filament
- Completely dissolved filament in HRS
- Cations as main mobile species

VCM criteria that are not fulfilled:

- Device structure/stack: Ohmic electrode/oxide/Schottky electrode
- Oxygen (vacancies) as main mobile species
- OFF state defined by Schottky barrier
- Switching occurs in the DISC by shift of oxygen vacancies (as mobile donors)
- PLUG length remains not significantly changed

Thus, no single criterion is completely fulfilled to allow an assignment of the switching mechanism to ECM or VCM. Neither the device structure nor the mobile ions, nor the type of resistance change.

The mobile species are both cations and anions and do not allow simple assignment to one of both types. The resistance change is not ECM (filament is neither metallic nor completely dissolving), but as well not VCM (no Schottky barrier change, and the length of the filament is changing).

Therefore, most proper description of the mechanism is in our view Filament conductivity change mechanism (FCM).

To summarize, the switching mechanism in ohmic memristive devices, i.e., filament conductivity change mechanism (FCM), is different from the switching mechanisms of ECM and VCM devices. In the revised version of manuscript, we removed the claim “new” mechanism.

Q2: My high-level recommendations are twofold: 1) To drop the focus on a new mechanism and just describe the mechanism as derived from the experimental support. The achievement of low work function (or combination of ECM with oxygen instead of metal) can still be highlighted! 2) Irrespective of the mechanism being new or not, to make the explanation clearer.

R2: We thank the reviewer for the suggestion. As recommended, we have removed the claim for “new” mechanism, and we focused on the mechanism as derived from the experimental support. We have also tried our best to explain more clearly the switching mechanism.

Some more detailed points about this:

Q3: As discussed by the authors in the mechanism discussion sections, and illustrated by Fig. 4, it looks like the mechanism is essentially the tip of the filament near the electrode being oxidized and reduced during switching, which leads to a longer or shorter filament and thus different resistance.

R3: The picture is a bit more complicated. The conductivity of the filament changes due oxidation and/or change of length. This results in a change of the resistance (during SET or RESET). It is important to explain that the resistance change in our devices is not happening only at the tip of the filament but can proceed much further (however, not to complete oxidation). We agree it may be confusing, showing in Fig. 4 that the filament is oxidized only at the tip. The oxidation proceeds much more along the filament (however, not completely). In the revised version of the manuscript, we have improved Fig. 4 to be more clear that a much longer part of the filament can be oxidized. Also, a short paragraph explaining this is added in the revised version of the manuscript.

Q4: This process happens by the movement of oxygen (vacancies) to and from the changing area (the tip of the filament) and this is common understanding for changing filaments in resistive switching. I understand that the authors want to highlight that the resistance is not due to a VCM Schottky mechanism where the height of a barrier changes at the tip of the filament. But the change of the filament still is very similar to an ECM mechanism, just oxygen moves instead of metal. The authors even write themselves that the switching kinetics can be explained with existing ECM models and they use the disc model (side comment, the disc is introduced in the text without explanation at first) which was also developed for ECM/VCM devices. So this looks something like an ECM mechanism where oxygen moves instead of metal ions.

R4: We respectfully disagree with the reviewer. At first, we would like to say, that one of the main criteria for distinguishing ECM from VCM is the type of main mobile species. One cannot simply say – this is an ECM device, but oxygen moves, instead of cations, as this is contradicting the classification criteria. Moreover, in our system both cations and anion are mobile.

As a second point we need to say that much longer part of the filament is involved (not only the tip, as discussed in **R3** of our responses). The switching kinetics (using ECM model) indicates only that we have a process that is similar to nucleation. However, one should not forget that all electrochemical processes with charge transfer rate limiting step (irrespective cations or anions) obey same kinetics, given by the Butler-Volmer equation. So, similarity in the charge transfer kinetics is not surprising.

We would also like to kindly remind that the Disc/Plug model is developed and applies solely to VCM! Disc model has never been developed for ECM and cannot apply to it as the filament in ECM/CBRAM devices are supposed to dissolve completely in stable HRS.

The filament conductance change mechanism (FCM) in ohmic memristors is very different with ECM and VCM mechanism (see **R1**).

Here we would like to explain again the mechanism (introduced in R2.2 in previous response letter):

The filament is composed of Ta-rich TaO_x and the change of the resistance state is not due to the change of the Schottky barrier height at the interface with the oxide or somewhere else, but because of the decreasing/increasing the conductance of the filament itself. This resistance increase is caused by 2 factors:

1. Increasing of the oxidation state of the metal ions (changing the stoichiometry).
2. Increasing/Decreasing the length of the filament.

These 2 factors can (but not necessarily) occur at same time.

When we apply a positive voltage to the filament we oxidize/passivate the TaO_x filament. This means, Ta-ions in the filament of lower oxidation state (e.g. 4+ and/or 3+ and/or 2+ and/or 1+) are oxidized to higher oxidation state e.g. Ta⁵⁺, accompanied by additional incorporation of oxygen ions and the oxide transits to its stoichiometric form. This oxidation will affect not only the first layers of the filament tip, but also expand deeper, reducing the effective length of the (TaO_x) filament. During this process, the filament is not completely dissolved (transformed to Ta₂O₅), as evidenced from the experimental results (TEM, electrical measurements showing HRS is independent on electrode area). This oxidation behavior is typical for electrochemical passivation of metals and conductive oxides.

This mechanism is fundamentally different compared to the wide-spread explanation of the VCM mechanism, explaining the switching by modulation of the Schottky barrier. It is also very different of the ECM mechanism, as discussed in our responses in R1.

Q5: So this looks something like an ECM mechanism where oxygen moves instead of metal ions. It seems to make sense to explain it like that, and can even be highlighted, but it is not a completely new mechanism.

R5: For this point we respectfully disagree with the reviewer. ECM and VCM are defined initially by the type of mobile species in the matrix. So, saying our mechanism is ECM with oxygen ions (anions) instead of cations is contradicting the definition for ECM and VCM. Moreover, we have both cations and anions as mobile species.

We also feel the need to discuss and clarify when one reaction or process mechanism is different than another:

One can try to unify all RRAMs speaking on a global level, saying all switching mechanisms are based on resistance change and for this reasons ECM, VCM, MRAM, FeRAM, STT-RAM etc. are not different, because all of them change their resistance.

One could also speculate on a lower level of generalization, saying ECM and VCM are not different, because they both are based on electrochemical reactions and ion transport. Such an argumentation can be made for several different mechanisms.

However, such a generalization is failing in describing the difference in the physics behind, which is essential for understanding and control of the processes and thus, functionalities and performance of the devices.

Therefore, mechanisms are distinguished based on criteria including, origin for resistance change, type of the filament, mobile species, reactions, how LRS and HRS are defined and composition of the stack.

Here we would refer to a physicochemical point of view how different mechanisms are justified and provide a very simple example: The reaction of hydrogen reduction/formation from water splitting (providing H₂) seems very simple. Nevertheless, in the electrochemistry there are three (3) different mechanisms of this simple reaction, namely:

1. $2\text{H}^+ + 2\text{e}^- = \text{H}_2$
2. $\text{H}^+ + \text{e}^- = \text{H}$ (atomic hydrogen)
 $\text{H} + \text{H} = \text{H}_2$
3. $\text{H} + \text{H}^+ + \text{e}^- = \text{H}_2$

In the physical chemistry these are 3 well recognised and accepted mechanisms. No one has argued that because these 3 reactions have same ions and atoms/molecules as educts and products they should be treated as same mechanism. Same applies for all (electro)chemical reactions having different pathway and/or different rate limiting step.

In the case of our ohmic memristors we clearly distinguish different mechanism, that cannot be assigned to ECM and as well not to VCM. Therefore, we feel scientifically obligatory to define precisely the mechanism and we propose the Filament conduction Change mechanism.

We hope very much the reviewer can follow our argumentation and agree with us.

Q6: There also remains part of my initial comment about the HRS. In the reviewer replies the authors write that the filament does not dissolve completely, but in the manuscript, Fig. 4, the disc model discusses and illustrates the opposite, that the filament dissolves completely.

R6: We are a bit confused by this comment. In **Q#3** the reviewer comments that from Fig.4 it looks like the oxidation/reduction of the filament occurs only at the tip, and here the reviewer comments again on Fig. 4, saying it looks like filament dissolves completely. Both comments appear to us contradictive.

Neither in the manuscript, nor in Fig. 4 we say or show the filament is dissolved completely. Just in opposite – we always say and show that the filament is never completely dissolved!

Q7: With the added figure R1.2 (which would be good to show in the SI!), the authors argue that HRS is not controlled by Schottky or Frenkel-Poole, because the plotted areas are not linear vs $E^{1/2}$. But the HRS currents are not linear ohmic either, plus they increase as temperature increases (more than LRS, Suppl. Fig. 19b). So they do not seem to be determined purely by the filament resistance, as the review reply states. So what is the conduction in HRS? (The part between 200 and 400 $(V/cm)^{1/2}$ in R1.2a actually looks more linear than Suppl. Fig. 19b.)

R7: We understand the concern of the reviewer about the conduction mechanism at HRS. Here we need to clarify that the measurements we have shown on the conductance in HRS and LRS were made in a broad voltage window (R1.2. from previous responses) of about 380 mV (-190 mV to +190 mV). However, in this range the applied voltage is sufficient to initiate electrochemical reactions as well, resulting in non-linear current response. Now we present the analysis in a narrower voltage window (-50 mV to 50 mV) which strongly minimizes the effects of polarization due electrode reactions, as shown in Fig. R2 (Supplementary Fig. 19). Thus, using a more appropriate conditions for analysis of HRS currents it is confirmed that HRS currents are linearly related to applied voltage, evidencing that no Schottky barrier effect is present. At the same time Fig. R2c clearly showed non-linear dependence, evidencing we do not have a resistance dominated by a Schottky barrier.

Fig. R2 (Supplementary Fig. 19): Temperature-dependent electrical characteristics in ohmic memristive devices. a, LRS currents show linear dependence on the voltage under various temperatures. **b,** HRS currents show linear dependence on the voltage under various temperatures. **c,** Fitting of HRS current using Schottky-emission model, the nonlinear curves indicate the current transport does not follow Schottky-emission mechanism.

Q8: Maybe as the other alternative, the authors mean by new mechanism that this happens with a low work function electrode. This is a very good achievement and should be highlighted! But it does not constitute a new mechanism, as it is still electrochemical as observed before, resulting in a change in length of filament. (The authors point out that the resistance of the remainder of the filament could affect the resistance further, but there does not seem sufficient evidence to conclude this. More in comment 3.)

R8: Thank you for this comment. We not only show resistance switching in devices with low work function electrodes but also demonstrate a different switching mechanism (see **R1**). We have observed partially dissolved TaOx filament (Supplementary Fig. 20a) in the TEM inspection, which supports the conclusion that filament composition change cause resistance change and the filament is not completely dissolved in RESET operation.

Q9: About clarification, the explanation in the author replies to the reviewer comments are much clearer about the mechanism than the manuscript itself. R2.2., for example, is a very clear statement, whereas in the manuscript, the explanation is not as clear and straightforward. For example, on p. 17 it is not clear whether the discussion is about the electroforming step or the switching. And the key aspects of the mechanism are not mentioned very explicitly. I recommend making the manuscript discussion clearer and more explicit on the central points.

R9: We thank the reviewer for this comment/recommendation. We have improved the explanation about switching mechanism in the manuscript (changes in the revised manuscript are highlighted in pages 20-21).

With more clarity from the review replies, there are still open questions, especially about the CV part:

Q10: It is mentioned a few times in the review replies and manuscript that it is not only the length of the filament, but also its resistance along the entire filament. However, there does not seem to be sufficient experimental evidence to support this as a novel finding.

R10: The resistance of one material (in this case our filament) is given by the formula $R = (1/\sigma)(L/A)$, where σ is the conductivity, L is the length and A is the surface area. Assuming the diameter of the filament is not substantially changed, the resistance of the filament is a function of two factors – the oxidation state (σ) and the thickness/length (L). In that sense as longer the filament, as higher the resistance.

As explained in **R4** in this response, there are two factors that influence the filament resistance – 1) the oxidation state of the ions and 2) the filament length. Both factors can be involved individually or together in the resistance change.

In addition, we have observed filaments of different lengths in our TEM images, corresponding to different resistances. So, our discussion is supported by electrochemical fundamental knowledge and experimental results.

Q11: It can be concluded from previous publications where the oxygen concentration was measured directly (e.g. <https://onlinelibrary.wiley.com/doi/full/10.1002/adma.201700212>), but the experiments in the present manuscript do not reveal this conclusion directly. With the added understanding from the review replies, I think the CV measurements are meant to provide evidence for the switching mechanisms, but as the authors write, they only reveal information about the interface reactions, and thus only about switching close at the interface with the electrode, but not about the rest of the filament.

R11: We are confused by the comment. We have shown in the manuscript (Fig. 3) and as well in the Supplementary information (Supplementary Figs. 20, 23 and 24) TEM analysis coupled to EDS profiles, clearly showing the change in the oxygen vacancy concentration and as well Ta-stoichiometry. It is also evident that filaments of different lengths were formed in different devices (in different HRS). Therefore, we provide direct experimental evidence with concentration profiles.

Indeed, the CV is providing only indirect evidence about the composition and length of the filament, but our conclusions were directly supported by TEM/EDS.

Q12: In my initial assessment, I commented that the CV discussion seemed detached from the rest of the manuscript and its purpose was not clear, so there were no comments to make really. From the review comments, I understand that the CV is meant to support the switching mechanism. This is still not

explained very clearly in the main text. (It says “We have studied in more details and comparatively analyzed the electrochemical behavior in both Schottky and Ohmic systems.” – but not why this matters for the manuscript. The CV discussion needs a clear motivation before delving into the details.) With this motivation now clarified, either way, there are some open questions in the CV discussion. (Minor comment in Fig. 2d caption, region II is in panel a.)

R12: We appreciate the reviewer comment. As we explained in previous response letter (**R6**), CV recorded the electrode kinetics and interfacial redox reactions about the systems. The electrochemical current peaks in CV provides information about electrochemical processes which are important to the explanation of filament formation processes and the passivation. Please refer to references **R1-R2** where we have more thorough discussion on the use and importance and CV in memristive systems.

In the revised version of manuscript, we have improved the explanation and motivation (in page 11) for using CV measurement. We have also corrected the figure caption.

[**R1**] Valov, I. & D. Lu, W. Nanoscale electrochemistry using dielectric thin films as solid electrolytes. *Nanoscale* 8, 13828–13837 (2016).

[**R2**] Lübben, M. & Valov, I. Active Electrode Redox Reactions and Device Behavior in ECM Type Resistive Switching Memories. *Advanced Electronic Materials* 5, 1800933 (2019).

Q13: a. Why does the current in CV saturate for the Pt devices, but not for the Cu devices? What is the difference and why does it matter for the manuscript? Is it just the different electronic barrier height before the electronic current takes over?

R13: Thank you for this comment. Indeed, the work functions of Cu and Pt are different (Pt has much higher work function) and thus, form different Schottky barriers. In both devices the currents reach a plateau/saturate (in the case of Cu devices ~ -4 V - -6 V). After the saturation region, the current in Cu device is increased due to the lower Schottky barrier Cu/Ta₂O₅ (1.45 eV), compared to that of Pt/Ta₂O₅ (2.45 eV) and possibly due other electrode process providing additional number of donors. This result shows that high Schottky barrier does not block electrode redox processes, but cause more severe electrode passivation due oxidation of Ta (compare Supplementary Fig. 16b,d). The latter can cause reliability problems in electrical performance of the memristive device. In contrast, in ohmic memristive system the passivation effect is significantly inhibited.

Q14: b. Centrally, about the H⁺ and H₂O conclusion, l. 247, which peaks does this refer to in Fig. 2a? The first small peak before saturation in the negative voltage region? It seems like it, because this peak occurs both for Pt and for Cu, and the authors write that the Pt should not undergo any reaction, whereas the Cu should. So the common feature in both measurements would not be due the different reactivity of the metals. If the little peak is indeed H⁺ or H₂O and not metal/electrode reaction, however, this implies that the shoulders in Fig. 2b could also be from the H⁺ or H₂O reaction, and not from metal/electrode reaction, and then there is no support for the electrode oxidation conclusion from CV. (Either way, the H⁺ or H₂O mention needs a reference.)

R14: Our statement in the text was: “We attributed the current density peaks to reduction reactions of metal ions and H⁺ or H₂O”. So we think the reviewer has overlooked the possibility for reduction of the metal ions Please, keep in mind, that as we used Ta electrode, still oxidation/reduction of Ta is probable.

The evidence in the CV that we indeed observe (partial) passivation of the Ta electrode and that it is more pronounced for devices with Pt electrode (higher Schottky barrier) than for Cu (lower Schottky barrier) is the decrease of the current of the whole CV with increasing number of cycles as shown in Supplementary Fig.16. There it is evident that the current is decreasing with increasing number of cycles and the decrease is much more pronounced for devices with Pt electrode.

Due to the complexity of the system i.e. electrode redox reactions (Ta redox reaction) and parallel H^+/H_2O reaction, and the fact we work with two electrode system, we cannot unequivocally assign the H^+ or H_2O reaction (or Ta reaction) to a specific peak. Electrode reactions may also proceed together (in parallel), and their redox peaks may merge resulting for example in a plateau. Nevertheless, the broader peak in Fig. 2b could be the oxidation of the Ta electrode as the current starts to saturate afterwards.

We would like to highlight that if there's only H^+ or H_2O reaction then the current in the system will be not reduced after the CV sweep. Instead, the currents will be increased due to the incorporation of H^+ and OH^- [R3], as the metals such as Pt catalyze the reactions [R4].

Reference R3 has been added (ref.36) in the revised manuscript.

[R3] Lübben, M. *et al.* Graphene-Modified Interface Controls Transition from VCM to ECM Switching Modes in Ta/TaOx Based Memristive Devices. *Advanced Materials* 27, 6202–6207 (2015).

[R4] Li, X. *et al.* Single-Atom Pt as Co-Catalyst for Enhanced Photocatalytic H_2 Evolution. *Advanced Materials* 28, 2427–2431 (2016).

Q15: c. The authors write that the current decrease in subsequent CV cycles for Pt and Cu is due to oxidation of the opposing Ta electrode. But there is a control missing for this conclusion with an inert opposing electrode. Could the authors make a device with Pt electrodes on either side? If the decrease in CV current is from oxidation of opposite electrode, then with Pt on both sides, the current should not decrease.

R15: Thank you for this comment. We have fabricated Pt/Ta₂O₅/Pt device and studied CV characteristics in our previous work [R3]. It can be seen from Fig. 1a in ref. [R3], the CV currents do not decrease with increasing CV sweeps in Pt/Ta₂O₅/Pt device. Just in opposite they increase due to incorporation of new ionic species serving as donors.

[R3] Lübben, M. *et al.* Graphene-Modified Interface Controls Transition from VCM to ECM Switching Modes in Ta/TaOx Based Memristive Devices. *Advanced Materials* 27, 6202–6207 (2015).

Q16: d. The authors write that the eq. 3-6 agree well with the experimental characteristics, but do not show it, like a fit to the data. Is this instead meant simply as qualitative statement that in different voltage regimes, the electronic current dominates and in others, where the electronic current is suppressed, the ionic contribution becomes visible? I think it should be stated like this if this is the case. Also, it is not really true for eq. 6, as in Fig. 2b, the electronic current is very pronounced both in forward and reverse, but in eq. 6, the electronic contribution is constant and small. This goes back to my initial comment about the electronic current in a reverse Schottky barrier.

R16: We thank the reviewer for the comment. Due to the complexity of the system (multiple redox processes) the conduction model is not able to quantitatively fit the whole CV sweep range. Moreover, due to redox reactions (and related charge injections) Schottky barrier height may also change with time. In that sense the equations cannot be simply fitted to the CV data. Nevertheless, the dominating factor that influences the device conductivity can be determined through the conduction equations. Please note that in Fig. 2b, one observes both electronic currents and redox currents (peaks), where electronic currents in the negative voltage range are much higher compared to same region in Fig. 2a. To distinguish the difference with the conduction in region II, it is justified to add the contribution of electronic current in equation 6.

Q17: e. The discussion after these equations, is it hypothesis as to what the effect might be? There is no experimental evidence that any of the Faradaic reactions indeed are more efficient or less efficient in the different situations. Or is this just a description of what is visible in Fig. 2? If yes, this would go with comment d that the equations give a qualitative description rather than direct confirmation. It should be stated clearly whether something is hypothesis or description of observation.

R17: We respectfully disagree with the reviewer. The discussions are based on the experimental results of CV characteristics. We clearly observed the differences in CVs with high and low Schottky barriers and distinguish the electronic and Faraday contributions in the different cases. The observed current density peaks indicative for redox reactions demonstrates where electrochemical redox processes dominate (reverse biased region in Fig. 2a, Suppl. Fig. 16, and Suppl. Figs. 17-18). Please refer to R10 and R16 on the role of CV characteristics in conduction mechanism in the system. In that sense the discussion and the provided equations are based on the conclusions drawn on the basis of experimental results.

Q18: f. Finally, in terms of contributing to the mechanism explanation, the CV explores effects at the interface to electrodes, but does not provide support for the changing resistance of the remainder of the filament. As mentioned earlier, the experimental support is only there for interface effects.

R18: The CV results are indicative only for the redox processes of Ta metal and Ta cations, suggesting Ta cation in the filament (TaOx) can also undergo such redox process. Cyclic voltammetry is not suitable method for determining the resistance. To evidence resistance change we provide TEM results, EDS and XPS element analysis to support the filament conductivity change mechanism. We have also discussed that the resistance change can be due oxidation or change of the length or both. And we also clearly have stated, we cannot distinguish between these 2 factors. We would provide again the equation for the resistance $R = (1/\sigma)(L/A)$ and the TEM results (main text and Suppl. Figures) to justify our claims.

Q19: Another note on clarification, p. 17, l. 344, “Switching at positive voltages (to Ta bottom electrode)” – this is confusing. At the start of the manuscript, it was written that bias was applied to the top electrode. There was no mention of changing this during the CV discussion, but this is very important, as the CV measurements were carried out with very asymmetric devices (Pt and Cu at top), so where was the bias applied? The schematics indicate bias application still at the top, but in l. 344 now suddenly it says at the bottom

R19: Thank you for this comment. We simply wanted to mention that the voltage was applied to Ta electrode. To prevent confusion, we have changed the description to “*Applying voltage bias to Ta electrode results in ...*” in the revised manuscript. We also updated the color of legend text Fig. 2a,b to highlight that the Pt, Cu, Ta, Hf, Ta, Zr were under biased, respectively during CV sweeps.

Q20: Finally a side note, Q4 from the previous comments maybe my initial comment was not clear. It was not my intention to compare the authors' work with this reference. Just the sentence "The robust device characteristics suggest that the occurrence of resistive switching in the ohmic memristors does not rely on Schottky barrier height modulation." On p. 11, l. 220, is not correct. Just because the device characteristics are robust, it does not mean that they do not rely on Schottky height barrier modulation. It is not a logically valid conclusion.

R20: We agree with the reviewer and have modified the text paragraph. In our work, we experimentally verify and discuss that Schottky barrier is not necessary/essential to have resistive switching in the memristive systems. We also experimentally demonstrated the advantages of ohmic memristive devices over conventional VCM devices, see response R5 in previous response letter, Supplementary Figs. 8,9 and related discussion in the main text.

Q21: Apologies for the load of text. I do not mean to drag out the review process. But I want to engage in a proper scientific discussion which the manuscript deserves. With the recommended clarifications, I believe the manuscript can still be suitable for Nature Communications. It does not need a 'revolutionary thing' like a new mechanism. The demonstration of excellent resistive switching with low work function electrodes is a strong enough finding! But it is very important to present the data and conclusions accurately and clearly.

R21: As responded in R1, we have changed the claim about switching mechanism to better demonstrate switching processes. We hope the reviewer will be satisfied with the changes.

A few minor comments:

Q22: For the sputtering processes, the gas pressure, gas flow, substrate temperature, substrate bias, and target power are missing.

R22: Thank you for this comment. We have added the details of layers deposition conditions (see Table R2) in Supplementary Table 4.

Table R2 (Supplementary Table 4): Summary of electrode and metal oxide deposition parameters.
All deposition processes were conducted at room temperatures.

RF Magnetron Sputtering					
Target (material)	Sputter power (W)	Pressure (μbar)	Ar flux (sccm)	O ₂ flux (sccm)	Rate (nm min ⁻¹)
Hf	60	5	15	–	~20
HfO ₂	13	10	15	–	~0.95
Ir (IrO ₂)	30	3	20	2	~18.3
Pt	80	5	15	–	~30
Ta	25	4	15	–	~5.5
Ta (Ta ₂ O ₅)	100	40	12	8	~1.8
Zr	60	5	15	–	~15

DC Magnetron Sputtering					
Target (material)	Sputter power (W)	Pressure (μbar)	Ar flux (sccm)	N ₂ flux (sccm)	Rate (nm min ⁻¹)
Pt	200	5.5	30	–	~69
Ti (TiN)	200	1.1	27	3	~7.4

Electron beam evaporation				
Material	Voltage (kV)	Current (A)	Pressure (μbar)	Rate (nm s ⁻¹)
Cu	~8.43	~0.2	1.2	0.01

Q23: As the simulations are a central part of the manuscript, I believe there should be a Code Availability statement. Ideally, the code would be available in a repository.

R23: Thank you for this comment. We have added the Code availability statement in the manuscript. The codes are available with the publication of this manuscript.

Q24: The difference between Fig. 3 f and h is not pointed out in the caption.

R24: Thank you for this comment. We have clarified this by adding the description in the figure caption “*h*, EDS line-scan analysis (see circled area in panel j) of the conduction channel region, suggesting larger Ta-rich filament is formed in ohmic memristive devices when operated in δw polarity (as shown in panel g).”

Q25: I tried to understand better the network simulations. The key things that made it difficult are:

R25: We have tried our best to explain the neural network simulations. We hope that the reviewer will have a clearer understanding on the simulation part in the revised manuscript.

Q26: The structure of the simulated network is not explained very clearly, the information is scattered

in different methods and figures parts. I think a clearer dedicated description of the network architecture will help, for example by expanding on Suppl. Fig. 28 and 29 and writing clearly in the main text that the structure is explained there.

R26: Thank you for this comment. The structure of the simulated neural networks has been explained in detail in the Method section “*Metaplasticity inspired continual learning.*” We further added the description of neural network structure in the Figure caption of Supplementary Fig. 28.

Q27: It would be helpful to have the propagation and update equations in the supplementary information. For example, the text says in l. 510: “in which the hidden parameters can decide the degree of the synaptic weights’ change.” That sounds like the hidden value is a weight parameter in some equation. But what is the equation?

R27: Thank you for the important suggestion, the equation about the hidden value can be found in Supplementary Fig. 27, in which the meta-function is:

$$f_{meta} = 1 - \tanh(m \times W_h)$$

where m is the hyperparameters in the neural network training. And we added this equation during the weight updates into the caption of Supplementary Fig. 27.

Q28: L517, “the weights between two neurons are hard to keep ‘memory’ for previous training experience”, there seems to be something wrong with this sentence.

R28: We thank the reviewer for pointing out this detail question, after revising again this sentence, to avoid unnecessary confusion, we have corrected this sentence (in page 25) as ‘the pre-trained weights between two neurons are hard to keep ‘memory’ for previous training experience after being trained by the new tasks’

Q29: Different effects of noise and failure rate are discussed, but without assessment, how ‘good’ it is. Given that the experimental resistance values were mapped to the network, is there something like a digital baseline, with which to compare? Or how do the simulations compare with other networks based on experimental values?

R29: Thank you for this comment. In the discussion about the network performances with respect to the experimental parameters of the ohmic memristors, such as failure ratios, resistances and variations, we mainly focus on the comparison among the different levels of these parameters, the assessment for evaluating the neural network performances is the final accuracy of different testing tasks, which can be found intuitively in the manuscript or the supplementary information. The digital baseline performances are added into Supplementary Fig. 28b, in which all the weights are implemented at the software platform, which can be the benchmark, with the tiny accuracy loss when adopting the ohmic memristors, the continual learning capacity can be better kept in the memristive hardware platform.

Reviewer #2 (Remarks to the Author):

The authors have provided adequate explanations regarding the issues I previously raised. My concerns about the interpretation of the mechanism proposed by the authors, compared to the conventional filament-type switching mechanism, have not been fully resolved. However, there are no logical inconsistencies, and the manuscript offers readers a meaningful and interesting perspective. Therefore, I recommend publishing the current version of the manuscript in Nature Communications.

Our response: Thank you for the positive evaluation on our revised manuscript. We have tried our best to explain in more details the switching mechanism in the general responses to all reviewers at the beginning and as well in detailed answers to Reviewer #1.

Reviewer #3 (Remarks to the Author):

The authors have addressed most of my comments. Regarding the 'unique properties' for continual learning, the authors rightly pointed out that the application demands low programming error. While there have been other efforts to reduce programming errors through testing methods and algorithms, the material effort is orthogonal. The authors also provided endurance performance comparisons with other device technologies, but it would be even better if the authors could discuss whether the current endurance performance can support continual learning applications.

Our response: We thank the reviewer for the positive evaluation on our revised manuscript. In the revised version of manuscript, we added the discussion on the impact of endurance characteristics on continual learning application in the Method section: “Endurance requirement for continual learning”.

Endurance requirement for continual learning

For continual learning applications, the weights need to be updated continually along with the network optimization, hence the high endurance of devices are crucial for the online training. And the endurance is associated with the resistance switching window, therefore the endurance for the binary-switching devices is the bottleneck in this application. Generally, the maximum number of the binary resistance switching can be derived as:

$$N_{max} = \sum_{i=1}^T \frac{S_{total,i}}{S_{batch,i}} \times E_{train,i}$$

Where $S_{total,i}$ is the total number of samples in the i_{th} trained task, $S_{batch,i}$ is the batch number of samples in the i_{th} trained task, $E_{train,i}$ denotes the training epochs for the i_{th} trained task, hence the endurance of the device should be more than the maximum binary-switching number N_{max} during training all the tasks from the first to the T_{th} . By taking the experimental values in this work into the equation and assuming that the parameters of each dataset are consistent, the endurance of more than 10^6 can be trained for more than 83 tasks, which can meet most needs in the edge platform, and the maximum number of trained tasks can be further increased by adjusting related parameters.

Response letter to the reviewers

We would like to thank the reviewers for the positive assessment of our revised manuscript.

In accordance with the reviewer's comments, we have amended and revised the text, provided the voltage range information for the data used in conduction model fitting. We have also updated the discussion and explanations on the part considering the interfacial properties and related CV measurements.

The manuscript and supplementary information were carefully read and edited.

We hope that the reviewers will find the responses and the corresponding revisions satisfactory. If any additional information is required, we will be more than happy to provide it.

Kind regards,

Shaochuan Chen, Yuchao Yang and Ilia Valov

Point-by-point response to the reviewers' comments

Reviewer #1 (Remarks to the Author):

Q1: Thank you to the authors for their continued effort and patience to improve the manuscript. The new changes and explanations are very good and especially the naming of their observation and comparison with ECM and VCM is an elegant solution. With this comparison and the additional explanations, the experimental observations can be explained consistently.

Apologies for the things I had overlooked in the previous version - this is a very comprehensive work! – or where my formulations weren't clear (e.g. between Q3 and Q6 – I meant complete rupture of the tip of the filament, not complete dissolution of the complete filament, obviously not clear enough wording).

R1: We appreciate that the reviewer found our revised manuscript satisfactory.

Q2: My main concerns have been addressed and the integration of the CV experiments is much clearer now, as well as the explicit explanation of the observed switching. As part of this, the linear IV in HRS in SI Fig. 19 looks convincing. For easier comparison, I suggest adding the measured voltage range in the figure caption for SI Fig. 19c, so it is clear that the displayed voltage range is comparable to a&b.

R2: Thank you for this comment. In this Figure caption of Supplementary Fig. 19c, we have added the voltage range information: “c, Fitting of HRS current using Schottky-emission model, the nonlinear curves indicate the current transport does not follow Schottky-emission mechanism. The voltage sweep range is between -50 mV to 50 mV, same as that in panels a and b.”

Supplementary Fig. 19. Temperature-dependent electrical characteristics in ohmic memristive devices. a, LRS currents show linear dependence on the voltage under various temperatures. b, HRS currents show linear dependence on the voltage under various temperatures. c, Fitting of HRS current using Schottky-emission model, the nonlinear curves indicate the current transport does not follow Schottky-emission mechanism. The voltage sweep range is between -50 mV to 50 mV, same as that in panels a and b.

Q3: There is one inconsistency left, which I think should be addressed.

Q&R16 there remains the inconsistency between the equation (6) and the measured CV curve in Fig. 2b plus the explanation beginning in l. 331. Fig. 2b and the explanation clearly state that there is similar dominant exponential electronic current in forward and reverse bias, but eq. (6) does not include this electronic contribution. It only includes the constant contribution ($-A \cdot T^2$) if the argument of the

Schottky exponential function is zero. The $-A \cdot T^2$ term provides a constant contribution, which does not reflect the measurement and explanation of Fig. 2b.

R3: We thank the reviewer for this important comment. After reading the text carefully, we found our discussion at this part was not clear in respect to the qualitative description of the interfacial barriers and the experimental results we use to support it. In the revised version of the manuscript, we rearranged the text/figure and improved the discussion and explanations.

Eq. 1-6 describe a steady-state situation. Here time plays no role in formalism. Examples of these equations are the Butler-Volmer equation and as well the Schottky emission equation. Time is not appearing in these equations as a variable. This part aims to explain qualitatively what happens on fundamental level *at the interfaces* at the four boundary conditions.

Cyclic voltammetry is a potentiodynamic experimental technique that is time dependent. The formalism here is different. The formalism from steady-state experiments is not appropriate for describing CV dependencies. This part supports the discussion and conclusions, made based on considerations using eq. 1-6.

In our manuscript we have discussed the importance of considering both Schottky and redox interfacial barriers at the metal/oxide interface. We also provided the steady-state equations that describe this interplay and qualitatively analyzed the four boundary conditions at this interface (Fig. 2a,b,c,d). This is especially important to understand why high Schottky barriers are not advantageous for reliable resistive switching performance. Thus, eq.1-6 serve to critically consider the interface interplay between Schottky and redox barriers. They are not supposed to describe the I - V dependencies (that include also the transport terms).

To support the discussion, we provided CV experiments (e.g. Fig. 2e,f and others in SI) to demonstrate the effects of Schottky barriers on the dynamic system behavior. Thus, we confirm that there is an obvious difference when using materials with different (high) Schottky barriers, namely Cu and Pt (Fig. 2e). Moreover, these experiments show that in both cases the electronic current is substantially blocked but the Faraday currents are not. In Fig. 2f the CVs show that for ohmic devices both electronic and ionic currents can pass the surface energy barriers, and the electronic current of the devices is not blocked by the Schottky interface barrier (in contrast to devices with interfaces with high Schottky barrier). The CV experiments thus support our discussion.

Please consider as well, that the incorporated ions are also serving as mobile donors, thus increasing electronic conductivity. Thus, if the number of incorporated ions increases exponentially (due to a Faraday reaction), the number of compensating charges (electrons or holes) will also increase exponentially, thus, contributing to the total current. The phenomenon of increasing the total current without having filament formation is observed and reported by us in Lübben, M. *et al.* Graphene-Modified Interface Controls Transition from VCM to ECM Switching Modes in Ta/TaOx Based Memristive Devices. *Advanced Materials* 27, 6202–6207 (2015). Within the total current we cannot exactly determine the ionic and electronic contributions (the transference numbers)

Equation 6 says that the electronic current due to Schottky emission at the interface is low and constant and that the ionic reaction current is supposed to increase exponentially. As mentioned before, in the dynamic systems the incorporation of ions is also increasing the electronic conductivity.

It is important to say, (as we responded) in the previous review round, equations (1-6) are describing qualitatively the interplay between the Schottky barrier and the redox reaction barrier at the metal/oxide interface. They apply to the qualitative description of the current, if controlled by the interface exchange of mass and charge at that interface. These equations are supposed to qualitatively support the discussion on the effect of the Schottky barrier and the importance to be considered together with the redox barrier. The purpose of providing these equations is to discuss the differences in the physicochemical behavior of the electrochemical systems in the cases of the four different boundary conditions.

Q4: With some further thought, I think that for the OE/oxide/OE in Fig. 2b, the starting point of adding eq. (1) and (2) is not appropriate anymore in the same way as for the case of high Schottky contact. It should be ok as a simplification for the high Schottky contact, where the electronic current is mostly blocked in the reverse bias direction, and the forward direction is then controlled by the forward-biased Schottky barrier, if the counter-electrode is ohmic.

R4: Thank you for the comment. In accordance with the discussion in **R3**, equations (1) and (2) apply to discuss the situation of interfacial processes at the metal/oxide interface. They do not account for the dynamic situation of reversible incorporation of ions (donors) and transport.

By achieving nearly ohmic-like interface we mitigate not only the passivation of metal electrode but also high voltage-drop across metal/oxide interface which could result in irreversible breakdown of the oxide. This model is important to analyze the interplay between Schottky barrier and redox barrier to determine the physics of filament formation processes and is applicable to other metal/oxide/metal memristive systems.

Q5: But for the OE/oxide/OE, the main point of the manuscript is that the electronic resistance is not controlled by a Schottky barrier, isn't it? So then in the expression $j = j_{TE} + j_{ION}$, for the OE/oxide/OE case, the j_{TE} should not be a Schottky expression, right? Otherwise this would be inconsistent with the mechanism explanation. This then also extends to eq. (5): Here, while the exponential term from the Schottky expression qualitatively describes the current, the Schottky expression does not seem appropriate physically, as the explanation of the mechanism is that the current is not controlled by a Schottky barrier.

R5: Thank you for the comment. Here applies the explanation/discussion provided in **R3**. Eq.5 considers the situation at the particular metal/oxide interface and does not aim to account for the dynamic change due ion/donor reversible incorporation at potentiodynamic conditions.

We show that the resistive switching in OE/oxide/OE is not based on Schottky barrier height modulation, but filament conductivity change. The latter could result in more controllable switching processes and device performance (see **R5** in our first response letter). In equation (5) the parameters are not influenced by Schottky barrier height (ϕ_B). Moreover, the ionic current (right side of equation (5)) can also contribute to the exponential increase of currents and as well related increase of electronic charges. Note that the CV measurement was conducted on devices prior to forming, to use the model and analyze the conduction is useful to study the physics/chemistry of the filament formation and to compare the switching process in SE/oxide/OE and OE/Oxide/OE systems.

Q6: Since the injection from the OE into the oxide is near-ohmic, maybe an expression in the space-charge-limited conduction framework can provide an appropriate expression for the electronic

contribution. I know SCLC is often misused in resistive switching work, but in this case, some of the boundary conditions should actually be fulfilled. See e.g.

<https://journals.aps.org/prapplied/abstract/10.1103/PhysRevApplied.9.044017> for an example of a similar band structure configuration as for the present manuscript. (I know it's for a completely different materials system, but the theoretical treatment might be transferable.) Of course, the explanation doesn't have to be based on SCLC, it is just an idea for a possible starting point.

R6: Thank you for this constructive comment. Indeed, when we start the next project on the theoretical framework for modelling the system, we'll definitely consider the SCLC. For the case of our present manuscript we refer again to our discussion provided in **R3**.

Q7: For the explanation of the measured added ionic and electronic current, in all contact cases, please can the authors also add a very brief explanation, why the ionic contribution appears as peaks, i.e. reduces after this peak? From the provided expression for the ionic contribution, I think there should not be a peak, but a constant increase with voltage. Are some of the parameters in eq. (2) possibly voltage-dependent? Just one sentence would clarify this.

R7: The current density peaks observed in the CV correspond to the oxidation/reduction processes at the electrode. The reason for observing current peaks is the dynamic situation at the interface. When the voltages increase the oxidation/reduction reactions start leading to exponential increase of the current. However, at certain moment (depends on the sweep rate), the concentration of reactants depletes, as the reaction rate is fast, and the supply of new reactants (ions) is too slow. Of this reason, the current decreases, thus, forming a peak. Both the concentration depletion at the interface and slow transport of reactants leads to the formation of current density peaks. Details on CV theory can be found in A. Bard and L. Faulkner, *Electrochemical Methods: Fundamentals and Applications* (Wiley, New York, 2001).

The mathematical formalism of the cyclic voltammetry is different than that of the steady state processes (Butler–Volmer equation). For example, in B-V equation the ionic currents (j_{ION}) are voltage-dependent, but not time dependent.

Q8: On the same discussion, while the authors disagreed to my comment Q17, they also confirmed that the provided equations are qualitative descriptions of the observed measurements, rather than exact analytic expressions which. While a small detail, and while qualitative explanations are often appropriate, for scientific accuracy, I think it is important to distinguish between qualitative description and quantitative/analytical analysis. (This is obviously a minor comment.)

R8: Thank you for this comment. We fully agree that qualitative expressions are different than analytical solutions (allowing for quantitative predictions and modelling). As explained in **R3** (and in previous review round) due to the complexity of the system (multiple redox processes, donor/acceptor incorporation etc.) the models are not able to quantitatively fit the whole voltage sweep ranges. In the revised manuscript we have explained this in the text (page 17): *“Due to the complexity of the system (multiple redox processes) the clear assignment of the redox peaks is challenging. Possible electrode half-cell reactions are summarized in Supplementary Table 2.”*

Q9: Finally, I recommend special care in proof-reading the hopefully final version of the manuscript, as there persist some small grammar mistakes here and there. Again, this is obviously a minor comment.

R9: Thank you for this comment. We have read the manuscript carefully and have corrected the grammar mistakes and typos.

Reviewer #3 (Remarks to the Author):

The authors have addressed my comments, and I agree that this is ready for publication in Nature Communications.

Our response: We thank the reviewer for the positive evaluation on our manuscript.

Reviewer #1 (Remarks to the Author):

We thank the reviewer for reading and commenting on our manuscript. In the current point-by-point responses we hope we have now addressed the reviewer's concerns. The changes are included in the revised version of the manuscript and highlighted in yellow.

Q1: Thanks again to the authors for their efforts to improve the manuscript. The smaller comments have been answered sufficiently. However, I am sorry to say that my main comments Q3-5 have not been answered. The authors provided a discussion about ion exchange, but this does not address my comments.

R1: We thank the reviewer for the comment and apology for not have been properly understood the concerns related to comments Q3-Q5 from the previous review round. We have now reassessed our responses and believe now we have correctly identified and addressed the criticism of the reviewer in an appropriate way.

Q2: In a CV measurement, an electronic signal is measured. Just from the measured signal, it is not possible to say whether this is an ionic or electronic response. (Frequency- or temperature-dependent measurements could possibly distinguish.)

R2: We fully agree with the reviewer. All signals measured by potentiostats and current/source meters are electronic. CV measurements cannot be used to distinguish between electronic and ionic currents contributions.

Q3: However, in the absence of an ion reservoir (like liquid metal electrodes in the early days of ionic measurements), and thus at least partially ion-blocking electrodes, and at the same time low electronic barriers, as in the given case of near-ohmic electrodes, the electronic contribution is typically much larger than the ionic contribution.

R3: We understand the concern and agree that in these systems the electronic contribution typically dominates over the ionic one. The fact that we are able to distinguish redox peaks is evidencing that at least at lower voltages both contributions are of same order, but at higher voltage ranges it is not possible to distinguish. Taking into account the low Schottky barriers, we assume (as also the reviewer pointed) the electronic currents are dominating.

Q4: Without proof to the opposite, it thus stands to assume that the exponential current increase in Fig. 2f (and similar figures) between -3 and -5 V and between 2 and 4 V is electronic and not ionic.

R4: We fully agree with the reviewer that the exponential increase after the redox peak is most probably dominated by electronic currents.

However, we cannot exclude a) direct and/or b) indirect ionic (redox) contribution to the high voltage increase of the current.

- a) The direct contribution is due to redox reactions. In our system we have a Ta metal electrode, Ta ions, lattice oxygen, and in same time H⁺ and H₂O (in total 5) that are sources of ions and

parallel electrochemical redox reactions. In CVs in liquid electrolytes, typically more redox peaks are observed and an exponential increase at high overvoltages. In most cases this exponential increase is due to water splitting (either O₂ or H₂ evolution, depending on the polarity). Thus, one can speculate (although we do not assume this) that the observed in Fig 2 peak is due to Ta oxidation and the exponential increase at higher voltages due water reaction.

- b) The indirect contribution is due to incorporation of ions into the oxide matrix (after the redox reaction), or oxidation of O²⁻ ions to O₂. In the lattice remain oxygen vacancies and electrons (as compensating defects). These ions serve as donors and can significantly increase the electronic currents.

In our manuscript Lübben *et al. Advanced Materials* 27, 6202–6207 (2015) we have already observed effects of H₂O reaction on the CVs and related increase of the current due to redox reactions and incorporation of water molecules and/or protons in TaO_x matrix. The current increase starts at about 2.5 V (at positive voltages) and -2.5 V at negative voltages. Therefore, ionic current contribution cannot be ruled out.

In the present manuscript we assume that especially at high voltages, the electronic current is dominating over the ionic one, but contributions of ionic/redox of direct or indirect contributions cannot be excluded.

Q5: So the explanation that this exponential current increase is due to ionic exchange is not convincing and not supported by evidence. The author reply "Equation 6 says that the electronic current due to Schottky emission at the interface is low and constant and that the ionic reaction current is supposed to increase exponentially." is not proof for this either, as it is only an interpretation of the results.

R5: We fully agree with the reviewer and apology for the mistake. In the revised version of the manuscript, we have modified equations 5 and 6 accordingly. Deriving eq. 5 and 6 we have assumed that the Schottky barrier is approx. 0 eV. However, we rethought this assumption and find it is not correct, as the value of the interface barrier is low but never zero, resulting in different equations. Therefore, we modified equations 5 and 6 in the revised version of the manuscript.

Q6: In fact, this is exactly what my main comments Q3-6 are about, namely the concern that this interpretation is not correct and the authors have not convincingly argued otherwise, as detailed just above. In fact, in R7 the authors write that the reason why the ionic contribution causes a peak, which decreases afterwards, is that "at certain moment (depends on the sweep rate), the concentration of reactants depletes, as the reaction rate is fast, and the supply of new reactants (ions) is too slow. Of this reason, the current decreases, thus, forming a peak. Both the concentration depletion at the interface and slow transport of reactants leads to the formation of current density peaks." This supports further the interpretation that the exponential current increase after the peaks is not ionic anymore.

R6: We agree with the reviewer, that the exponential current after the redox peak is dominated by electronic currents. Although direct and/or indirect ionic contribution (**R4** to **Q4**) cannot be excluded, the low Schottky barriers lead to high electronic currents.

Q7: In the previous manuscript version, the authors had written that there is similar exponential electronic current increase in the forward and negative bias, but in the new version, this was just replaced by writing that there are similar ionic contributions. So the authors just changed their interpretation from before.

R7: We apology for the mistake. The correct text is as in the initial version of the manuscript. The low Schottky barriers support higher electronic currents. We have not changed our minds but have changed wrongly the type of current contributions.

Q8: To clarify my point, I do not assume that the electronic contribution would arise from changing dopant concentrations due to changing ion concentrations. It simply arises because the low, near-ohmic, electronic barriers between the oxide and the electrodes make it very easy to inject electrons from the electrodes into the oxide. This is not a doping effect, but a carrier injection effect. (Hence my suggestion in Q7 that maybe SCLC could play a role here.)

R8: We fully agree that the electronic currents are dominating. As explained in R4, direct and/or indirect contributions cannot be excluded, but electronic currents are dominating. The modified equations 5 and 6 now, include an exponential term for the electronic current term at the interface barrier.

As the reviewer suggested we have checked about SCLC conduction influence. In the general case the current density versus voltage characteristics exhibit three regions. When increasing the voltage, the injection of charge carriers become stronger, changing the current from linear region to trap-filled limited conduction region, and space-charge-limited (trap-free) conduction region. The slope of current-voltage curve keeps increasing until reaching SCLC conduction region. This phenomenon is different with that observed in CV curves presented in our work. In the CV curves (Supplementary Fig. 18.) the slope of the J-V curve is decreasing at higher voltages.

So we think the revised equation 5 and 6 are capable for providing a qualitative description of the observed CVs.

Q9: In brief, the exponential current increase in Fig. 2f and similar figures looks to be electronic, not ionic, and equations 5 & 6 are not adequate to describe this, as it ascribes the exponential increase to an ionic contribution and does not include the exponential electronic contribution. My previous comments Q3-5 provide further details. To my understanding, the authors either need to provide proof that the exponential increase is indeed ionic, or they need to adapt the equations for the interpretation of the results.

R9: We fully agree with the reviewer. The main contribution of the current increase is electronic. We have accordingly modified equations 5 and 6 in the revised version of the manuscript.

Q10: As a side comment arising from this discussion, the CV measurements are actually not always explained clearly. What were the sweep rates in the figures? The experimental part provides a large range of sweep rates. But which ones were used in the figures? And were different sweep rates compared to investigate reactivity?

R10: For the shown CVs it was fixed to 130 mV/s. We have added this information in the Method section.

We have performed some CV measurements at different sweep rates and found that 130 mV/s is most appropriate for presenting and comparing all the different samples.

Using the sweep rates, typically serves to determine the type of control of the electrochemical reaction (charge transfer or transport limited). From the change of the peak current (and/or) peak voltage as a function of the sweep rate one determines the type of control. This type of measurements is mostly relevant for the linear sweep voltammetry (LSV)

In case of cyclic voltammetry, one typically is using a different criterion – the difference of the peak positions for oxidation and reduction reaction. This criterion provides same information. In our CVs we have quite a large distance between the oxidation and reduction peak, indicating charge transfer (Buttler-Volmer) control.

Point-by-point response to the reviewers' comments

Reviewer #1 (Remarks to the Author):

Q1: Thanks to the authors for these final corrections. I am glad we agree on the science of the discussed measurements and equations and think it inspires confidence in the results. The updated discussion reflects the complexity of the system well and I think it will be instructive to a wide audience. Thanks to the authors for their patience and the engaging review discussions! I think the article can be published in Nature Communications now and will have good impact. Congratulations on these demonstrations and good luck for all future work!

R1: We thank the reviewer for the positive recommendation on our manuscript.

General comment:

First, this paper applies low work function metals to both top and bottom electrodes to demonstrate an Ohmic memristive device and provides a reasonably detailed description of the resistive switching mechanism. I found no logical inconsistencies in the various approaches to describing the mechanism, and overall, it is well-written. Additionally, it seems to be a suitable manuscript for the *Nature Communications* journal. However, to enhance its completeness before publication, I suggest the following points.

1. The authors claim in the manuscript to present a new switching mechanism. Upon reading this manuscript, a fundamental question arises: there seems no reason why switching characteristics would not be observed when both electrodes are made of low work function metals, even when compared to the reported filament type mechanisms. One reason for using a high work function metal (or inert metal) like Pt at one electrode is to ensure reliability by preventing undesired reactions at one interface and inducing switching exclusively at the other. Using low work function materials on both sides seems likely to decrease reliability, and I would appreciate the authors' comments on this.
2. In Supplementary Fig. 11, XPS data measured at the center of each layer would naturally observe the metallic state. Oxidation of the electrode and reduction of Ta₂O₅ at the interfaces are expected, which might lead to Ohmic contact (reference 1). The redox at the both interfaces, rather than the electrode's low work function, seems to be the primary cause of Ohmic contact. I would appreciate the authors' comments on this.
3. The paper states that tantalum oxide used is in the amorphous phase. As the authors know, TaO can have various oxidation states depending on the fabrication method, and both filament and interface types have been observed depending on the oxidation state. The authors already have XPS data, so I recommend deriving the composition from XPS.
4. I am curious about the stability of the retention characteristics. The author shows measurements for 1200s, but this duration is too short. It would be better to demonstrate retention characteristics for at least two states over 24 hours.

5. Figure 1g shows multi-bit characteristics. It would be beneficial to also show DC curves within the V_{read} , as in the attached reference 2. Implementing multi-bit in an Ohmic regime would be advantageous when configuring a vector matrix multiplier of using analog inputs.

Minor things:

- (1) Supplementary Figs. 19, 20, 27, and 29 are not mentioned in the main body. All Supplementary Figures should be described.
- (2) Some Supplementary Figures are out of order. The numbering should match the order mentioned in the text.

References

- [1] Rao, M., Tang, H., Wu, J. *et al.* Thousands of conductance levels in memristors integrated on CMOS. *Nature* **615**, 823–829 (2023). <https://doi.org/10.1038/s41586-023-05759-5>
- [2] Yang, J., Pickett, M., Li, X. *et al.* Memristive switching mechanism for metal/oxide/metal nanodevices. *Nature Nanotech* **3**, 429–433 (2008). <https://doi.org/10.1038/nnano.2008.160>